# FGF7 promotes load-bearing tendon regeneration and suppresses fibrosis

Ruifu Lin[1,2,3,10], Junchao Luo[4,10], Hong Zhang[5,10], Chunmei Fan[2,6], Yue Hu[2], Ruojin Yan[1,2], Zetao Wang[1,2,3], Yang Fei[1,4], Chenqi Tang [4], Tianxi Huang[2], Tianshun Fang[1,2,3], Weiliang Shen [4], Sunbin Ling [7], Hongwei Ouyang [1,2], Xiao Chen [2,3,4,8] ✉ & Zi Yin [1,2,3,9] ✉

Tissue fibrosis is a major cause of organ dysfunction. Preventing fibrosis in tissue pathological condition remains a significant clinical challenge. Here we investigate the role of fibroblast growth factor 7 (FGF7) in mitigating fibrosis and promoting regeneration of load-bearing tendons. Fgf7 knockout mice exhibit impaired motor function and disordered matrix assembly in tendons. Single-cell RNA sequencing reveals an enrichment of a pro-fibrotic cell sub-population in Fgf7-deficient tendons, which is also predominant in human tendinopathy. Using ProTracer technology, we find that FGF7 deficiency drives proliferating cells toward this pro-fibrotic lineage. Furthermore, we find that FGF7 promotes the tenogenic differentiation of tendon stem/progenitor cells while suppressing their fibrotic differentiation. Importantly, a hydrogel loaded with recombinant FGF7 effectively reduces fibrosis and significantly promotes functional tendon regeneration in vivo. These findings elucidate FGF7's dual role in driving tenogenesis and inhibiting fibrosis, suggesting a potential therapeutic strategy.

Fibrosis is a pervasive condition affecting a wide range of tissues and organs and is a hallmark of numerous chronic diseases. The onset of fibrosis significantly impairs the functionality of various organs, including the heart, liver, lungs, and tendons, etc.[1–4]. Fibrosis is a pathological process initiated by the excessive deposition of extracellular matrix components, such as collagen, resulting in the replacement of healthy parenchyma with scar tissue[5]. Tendons, composed almost entirely of collagen, are particularly susceptible to fibrosis due to their dense structure and role in withstanding substantial mechanical loads. Due to their relatively simple collagen composition, tendons provide an ideal model for studying dysregulated collagen deposition in fibrosis versus normal matrix assembly. Such insights would not only advance fundamental tendon biology but also provide valuable perspectives on dysregulated collagen deposition in other collagen-dense tissues, such as skin and cardiac valves. Tendon fibrosis disrupts the normal extracellular matrix, leading to a weak and disorganized scar. Tendon fibrosis is characterized by a loss of parallel collagen alignment with fibrils appearing fine and uniformly

[1]Department of Orthopedic Surgery of Sir Run Run Shaw Hospital, and Liangzhu Laboratory, Zhejiang University School of Medicine, Hangzhou, China. [2]Dr. Li Dak Sum & Yip Yio Chin Center for Stem Cells and Regenerative Medicine, Zhejiang University School of Medicine, Hangzhou, PR China. [3]Key Laboratory of Motor System Disease Research and Precision Therapy of Zhejiang Province, Hangzhou, Zhejiang, PR China. [4]Department of Sports Medicine & Orthopedic Surgery, The Second Affiliated Hospital, Zhejiang University School of Medicine, Hangzhou, Zhejiang, China. [5]Center for Rehabilitation Medicine, Rehabilitation & Sports Medicine Research Institute of Zhejiang Province, Department of Rehabilitation Medicine, Zhejiang Provincial People's Hospital, Affiliated People's Hospital, Hangzhou Medical College, Hangzhou, Zhejiang, China. [6]Key Laboratory of Novel Targets and Drug Study for Neural Repair of Zhejiang Province, Department of Clinical Medicine, School of Medicine, Hangzhou City University, Hangzhou, Zhejiang, PR China. [7]Institute of Translational Medicine, Zhejiang University, Hangzhou, China. [8]State Key Laboratory of Transvascular Implantation Devices, Hangzhou, China. [9]Institute of Cell Biology Zhejiang University, Hangzhou, China. [10]These authors contributed equally: Ruifu Lin, Junchao Luo, Hong Zhang. ✉e-mail: chenxiao-610@zju.edu.cn; yinzi@zju.edu.cn

distributed when observed under transmission electron microscopy, along with aberrant proteoglycan deposition, matrix assembly disorder, an increase in myofibroblasts, elevated type III collagen deposition, and neovascularization. These pathological changes particularly the disorder in tendon matrix assembly directly compromise tendon biomechanics, resulting in chronic pain, joint stiffness, and an elevated risk of re-rupture, which ultimately leads to failed recovery and diminished quality of life for patients. The significant disability associated with these outcomes translates into a substantial clinical and economic burden incurring an annual cost of approximately $7 billion in the United States alone[6]. A deeper understanding of tendon matrix assembly disorder as a critical pathological factor in tendon fibrosis is urgently needed. However, the primary regulator that governs the tendon matrix assembly disorder has yet to be identified.

In this context, the fibroblast growth factor (FGF) family is of major interest, as its crucial functions in tendon development, homeostasis, and repair. Under mechanical signals, FGF4 exhibits tenogenic effects in the undifferentiated cells of chick embryo limbs but shows anti-tenogenic effects in mouse embryos[7]. Furthermore, FGF4 does not induce Scx expression in either mouse limb explants or mouse mesenchymal stem cells[8]. In conditions such as micrognathia, FGF8 secretion alters cell fate from tendon to cartilage, affecting mechanical force transmission and resulting in malformation[9]. In X-linked hypophosphatemia, disorders at the tendon-bone junction are predominantly associated with FGF23's impact on 1,25D production[10]. During the early stages of tendon development, FGF-2 is involved in promoting progenitor cell proliferation[11]. Additionally, electrospun fiber membranes embedded with FGF2 nanoparticles have been found to reduce adhesion in regenerating tendon tissues, thus facilitating tendon repair[12]. FGF2 is known to regulate the expression of genes involved in extracellular matrix (ECM) production, such as type I/III collagen and fibronectin[13]. Overall, current research on the impact of the FGF family on tendon tissues remains fragmented, and the exact functions of FGFs, particularly in normal collagen matrix formation and scar tissue formation, are still unclear. Notably, insights from wound healing in other tissues offer valuable clues. For instance, in skin wound healing, FGF7 is known to promote collagen formation and facilitate repair[14]. Existing studies indicate that the III domain of proteoglycans, which interacts with collagen in the matrix, can bind to FGF7. This binding not only serves as a reservoir for FGF7 but also facilitates its release when needed, thereby promoting cell proliferation and differentiation[15,16]. In a CCL4-induced acute liver injury model, FGF7 has been shown to assist hepatocyte survival, consequently mitigating the extent of fibrosis[17]. Our previous research identified FGF7 as a key factor enhancing the tenogenic capabilities of 3D-cultured human tendon stem/progenitor cells compared to 2D cultures[18]. Other studies suggest that FGF7 may contribute to the rapid repair and regeneration of heart chordae tendineae, which endure repetitive cyclic loads yet rarely experience overuse injuries compared to skeletal tendons[19]. We hypothesize that FGF7 may be a key regulator in normal collagen matrix formation and matrix assembly disorder of tendons.

To investigate the specific regulatory role of FGF7 in tendon morphogenesis and collagen matrix formation, we first explored the Fgf7 knockout (*Fgf7*[KO]) mice, revealing a tendon matrix assembly disorder phenotype in the load-bearing regions of these mice. Single-cell RNA-sequencing and ProTracer lineage tracing[20–22] identified that *Fgf7* deficiency leads to the enrichment of a *Lox*[+]*Uts2r*[+] subpopulation, driven by the activation of the transcription factor *Creb3l1* via the TGF-β pathway. This subpopulation also plays a promotive role in the progression of human tendinopathy. Furthermore, we developed a Gelatin methacrylate (GelMA) hydrogel loaded with recombinant FGF7 (GelMA-*c*-rFGF7), which enhanced both the functional and structural regeneration of tendons while inhibiting pathological fibrosis during the tendon repair process. These findings underscore the pivotal role

of FGF7 in preventing tendon fibrosis and promoting regeneration, offering potential therapeutic strategies for tendon diseases.

## Results

### Impaired motor performance in *Fgf7*[–/–] mice
Tendon tissues transition from a proliferative developmental stage to a mature homeostatic phase postnatally, facilitating mechanical transmission and supporting normal locomotor activities[23]. To investigate the role of FGF7 in this process, we generated systemic *Fgf7* knockout mice. To control for potential confounding effects of body weight and length on motor performance, we conducted comparative analyses of body length and weight (Fig. 1A, B) at critical developmental stages (postnatal days 7, 14, 21, and 10 weeks) between *Fgf7*[–/–] and wildtype mice. Our findings indicate no statistically significant differences in these measures between the two groups. Subsequently, we assessed the motor function of *Fgf7*[–/–] and wildtype mice (Fig. 1C). Grip strength and endurance were evaluated using the hanging wire test at postnatal days 7, 14, 21, and 10 weeks. No significant differences were observed between the groups at days 7 and 14; however, *Fgf7*[–/–] mice exhibited significantly reduced hanging times at days 21 and 10 weeks compared to wildtype mice (Fig. 1D). Additionally, maximal exhaustive treadmill tests were performed at days 21 and 10 weeks to further evaluate motor performance (Fig. 1E). *Fgf7*[–/–] mice displayed inferior performance relative to wildtype controls at both time points, including reduced time to exhaustion, shorter running distance, and lower maximum speed at exhaustion (Fig. 1F–H). These results collectively suggest that FGF7 plays a crucial role in maintaining motor function during the late developmental and homeostatic phases without affecting body weight or length.

### Aberrant macro, micro, and nanostructures of tendons in mechanically stressed regions of *Fgf7*[–/–] mice
To investigate the impact of FGF7 on tendon function, we conducted comparative analyses of tendon tissues from *Fgf7*[–/–] and wildtype mice at macro, micro, and nano scales. Macroscopic observations revealed that hypoplastic and tendon matrix assembly disorder tendons were predominantly observed in high mechanical load regions, including the patellar tendon, flexor digitorum tendon, and Achilles tendon (Fig. 2A). To determine whether the hypoplastic phenotype in high mechanical load tendons arises, we performed histological examinations of load-bearing tendons such as the patellar and Achilles tendons at P14, P21, and 10 weeks (Fig. 2B, D). At P14, no significant differences were found in the thickness of the patellar and Achilles tendons and their peritendinous tissues between *Fgf7*[–/–] and wildtype mice. However, by P21, the thickness of both the patellar and Achilles tendons was significantly reduced in *Fgf7*[–/–] mice compared to wildtype mice, with further reductions observed at 10 weeks (Figs. 2C, E and S1A–D). Additionally, we found that the deficiency phenotype at post-P21 and 10 weeks was not presented in the positional tail tendons (Fig. S1E–G). We further evaluated the development of patellar and Achilles tendons under polarized light at P14, P21, and 10 weeks (Fig. 2F, G). At P14, both wildtype and *Fgf7*[–/–] mice exhibited mature, parallel, golden-yellow collagen fibers. At P21, wildtype mice showed increased mature collagen fibers, whereas *Fgf7*[–/–] mice displayed fewer and discontinuous collagen fibers. By 10 weeks, wildtype mice had dense, mature collagen fibers, while *Fgf7*[–/–] mice exhibited a marked absence of these collagen structures. To further investigate structural abnormalities due to *Fgf7* deficiency, we examined tendons at the mechanically demanding stage of 10 weeks using transmission electron microscopy. Collagen fibril diameters in the patellar tendons of *Fgf7*[–/–] mice were generally smaller compared to wild-type mice (Fig. 2H). Additionally, the expected bimodal distribution of collagen fibril diameters in wildtype mice shifted to a more uniform, unimodal distribution in *Fgf7*[–/–] mice (Fig. 2I). Similar observations were made in the Achilles tendons, where *Fgf7*[–/–] mice exhibited smaller collagen fibril diameters and a uniform

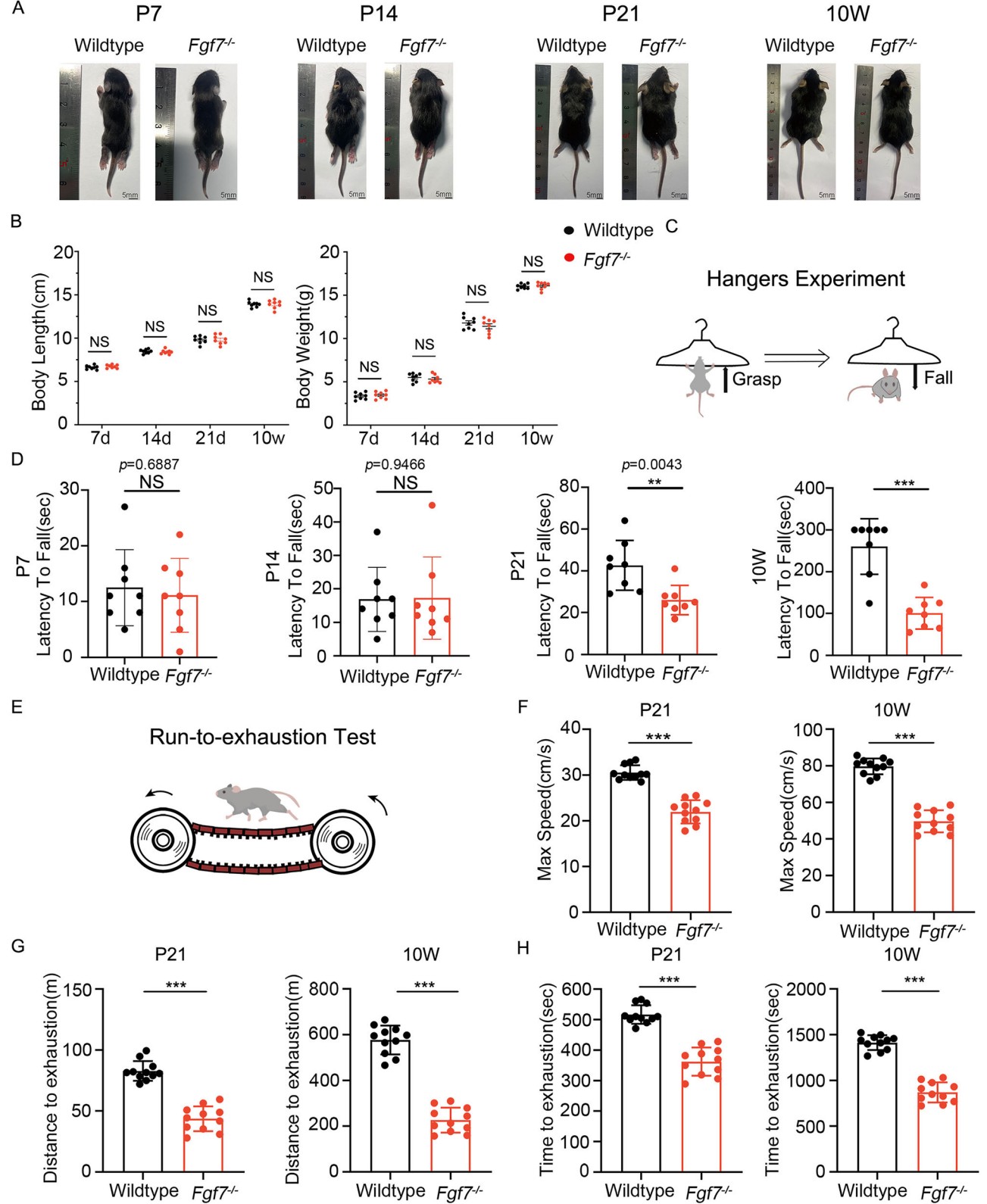

fibril distribution, in contrast to the bimodal distribution seen in wildtype mice (Fig. 2J, K).

In conclusion, these findings suggest that *Fgf7* deficiency leads to structural abnormalities in tendon tissues during late development (P21) and into maturity (10 weeks) as mechanical load increases. The absence of *Fgf7* may heighten sensitivity to mechanical stress, resulting in macro, micro, and nano-structural aberrations and damage in tendon tissues.

## Matrix assembly disorder phenotype in tendon tissues of load-bearing regions in *Fgf7*−/− mice

To further investigate the exact role of *Fgf7* in tendon tissues hypoplastic and matrix assembly disorder, we harvested Achilles tendon tissues from wild-type and *Fgf7*−/− mice at tendon maturation (10 weeks) and performed bulk RNA-sequencing (10 weeks) (Fig. S2A) to investigate differential gene expression. Principal component

**Fig. 1 | Impaired motor function in *Fgf7*[−/−] mice compared to wildtype mice.**
**A** Morphological changes and body length at postnatal days 7, 14, 21, and 10 weeks. Scale bars, 5 mm. **B** Statistical analysis of body length (left) and body weight (right) at postnatal days 7, 14, 21, and 10 weeks for *Fgf7*[−/−] and wildtype mice (*n* = 8 per time point, independent experiments, bars represent mean ± SD; NS: not significant, two-tailed Student's *t* test. body length: 7 days: *t* = 1.075, df = 14; 14 days: *t* = 0.5283, df = 14; 21 days: *t* = 0.7001, df = 3; 10 weeks: *t* = 1.134, df = 3. Body weight: 7 days: *t* = 0.5048, df = 14; 14 days: *t* = 0.1722, df = 12; 21 days: *t* = 1.265, df = 3; 10 weeks: *t* = 0.3482, df = 14.). **C** Schematic representation of the wire hang test protocol. **D** Statistical analysis of hanging time in the wire hang test at postnatal days 7, 14, 21, and 10 weeks for *Fgf7*[−/−] and wildtype mice (*n* = 8 per time point, independent experiments, bars represent mean ± SD; WT wildtype. NS: not significant, *p < 0.05, **p < 0.01, two-tailed Student's *t* test. Hanging time in the wire hang test at postnatal days: 7 days: *t* = 0.409, df = 14; 14 days: *t* = 0.06817, df = 14; 21 days: *t* = 3.399, df = 14; 10 weeks: *t* = 5.913, df = 14.). **E** Schematic representation of the treadmill test.

**F** Statistical analysis of maximum running speed in the treadmill exhaustion test at postnatal day 21 and 10 weeks for *Fgf7*[−/−] and wildtype mice (*n* = 11 per time point, independent experiments, bars represent mean ± SD; WT wildtype. *p < 0.05, ***p < 0.001, two-tailed Student's *t* test. Maximum running speed at postnatal days: 21 days: *t* = 9.478, df = 20; 10 weeks: *t* = 13.35, df = 20). **G** Statistical analysis of maximum running distance in the treadmill exhaustion test at postnatal day 21 and 10 weeks for *Fgf7*[−/−] and wildtype mice (*n* = 11 per time point, independent experiments, bars represent mean ± SD; WT wildtype. *p < 0.05, ***p < 0.001, two-tailed Student's *t* test. Maximum running distance at postnatal days: 21 days: *t* = 10, df = 20; 10 weeks: *t* = 13.92, df = 20). **H** Statistical analysis of maximum running time in the treadmill exhaustion test at postnatal day 21 and 10 weeks for *Fgf7*[−/−] and wildtype mice (*n* = 11 per time point, independent experiments, bars represent mean ± SD; WT wildtype. *p < 0.05, ***p < 0.001, two-tailed Student's *t* test. Maximum running time at postnatal days: 21 days: *t* = 9.223, df = 20; 10 weeks: *t* = 13.23, df = 20). Source data are provided as a Source Data file.

analysis (PCA) indicated distinct clustering between *Fgf7*[−/−] and wild-type groups (Fig. S2B), and the volcano plot highlighted differentially expressed genes between groups (Fig. S2C). Using bulk RNA-sequencing data, Gene Set Enrichment Analysis (GSEA) with tendon ECM, tendon TF datasets and PI3K-AKT pathway gene sets in *Fgf7*[−/−] tendons compared to wildtype tendons (Figs. 3B and S2G–I). RT-qPCR further validated downregulation of tendon-related transcription factors (Egr1, Scx, Mkx) and ECM-related genes (Tnc, Col1a1, Dcn, Bgn) in *Fgf7*[−/−] mice tendon tissues (Fig. 3A). These RNA-level changes correlated with previous observations of hypoplastic tendon structures at macro, micro, and nano scales in *Fgf7*[−/−] mice, suggesting a matrix assembly disorder phenotype in tendons. Western blotting results revealed an upregulation of ACTA2 in the tendon tissues of *Fgf7*[−/−] mice (Fig. 3C, D). Downregulation of collagen type I (COL1) was also observed in the tendons of *Fgf7*[−/−] mice, a finding implicated in the disruption of tendon matrix assembly and organization. Immunohistochemical analyses showed decreased expression of tendon marker proteins (COL1A1, TNMD, EGR1) and increased expression of fibrotic markers (COL3, ACTA2) in *Fgf7*[−/−] mice tendons (Figs. 3E and S2J). Immunofluorescence analyses on *Fgf7*[−/−] mice on SCX-GFP background showed decreased GFP expression in Achilles tendons, indicating abnormal tendon phenotype (Fig. 3F–H). KEGG pathway analysis of bulk RNA-sequencing data from wildtype and *Fgf7*[−/−] mice revealed significant differences in pathways related to stem cell growth regulation, cell body membrane, and cell cycle, cytokine-cytokine receptor interaction, and PI3K-AKT signaling (Fig. S2D, E). Additionally, we found that FGF7 is primarily expressed in the tendon paratenon/epitenon tissue and in close proximity to stem cell markers (sca-1[24]) within tendon tissues (Fig. S2F). Since the paratenon/epitenon is considered to be the niche for tendon stem/progenitor cells, the specific localization of FGF7 and RNA-seq findings suggest that FGF7 appears to facilitate tenogenesis by regulating the biological activities of tendon stem/progenitor cells. Overall, our findings indicate that *Fgf7* knockout mice exhibit a matrix assembly disorder phenotype in tendon tissues, which is likely related to the role of FGF7 in regulating the function of tendon stem/progenitor cells.

## FGF7 regulates the switch between tendon and fibrosis cell fates

To validate our hypothesis that FGF7 may induce tendon morphogenesis and inhibit fibrosis by influencing tendon stem/progenitor cells within the tendon, we isolated tendon stem/progenitor cells (TSPCs) from the tendon tissues of *Fgf7*[−/−] and wild-type mice. These were divided into three groups: TSPCs from *Fgf7*[−/−] mice, TSPCs from wildtype mice in standard culture, and TSPCs from wildtype mice cultured with rFGF7 (Fig. S3A). After 3 days of culture, we observed that rFGF7-treated TSPCs exhibited enhanced proliferation compared to the control group, as evidenced by an increased proportion of Ki67[+] cells and a higher number of EDU-incorporating cells. Conversely, the *Fgf7*[−/−] group showed reduced proliferative capacity, with fewer Ki67[+]

cells (Fig. S3B) and fewer EDU-incorporating cells (Fig. S3C). This indicates that FGF7 promotes the proliferation of mouse TSPCs. To investigate whether FGF7 can enhance tenogenic protein expression and suppress fibrotic protein expression in mouse TSPCs, we performed immunofluorescence staining for relevant markers in rFGF7-treated, control, and *Fgf7*[−/−] groups. The rFGF7-treated group exhibited elevated levels of tenogenic markers COL1A1, MKX, EGR1, and TNMD compared to the control group, while the *Fgf7*[−/−] group showed reduced expression of these markers (Fig. S3E–H, K). Additionally, the rFGF7-treated group displayed significantly lower levels of fibrotic markers ACTA2 and COL3 compared to the control, whereas the *Fgf7* knockout group demonstrated higher expression of these fibrotic proteins (Fig. S3I–K). Furthermore, after inducing tenogenic differentiation in all three groups, Sirius Red and Masson's trichrome staining revealed that rFGF7-treated TSPCs exhibited a stronger tenogenic differentiation capacity compared to the control group (Fig. S3D). In contrast, TSPCs from the FGF7 knockout group showed a diminished tenogenic differentiation capacity compared to the control group.

To ascertain whether FGF7 exerts similar antifibrotic effects in human TSPCs and its potential translational application in clinical tendon regeneration, we isolated human TSPCs and treated them with rFGF7 and the FGFR-specific inhibitor AZD4547 (Fig. 4A). Post a 3-day culture, rFGF7-treated human TSPCs displayed increased Ki67[+] cell proportions (Fig. 4B), higher EdU incorporation (Fig. 4C), and enhanced clonogenic capacity (Fig. 4D) compared to controls, all indicative of augmented cell proliferation. In contrast, AZD4547-treated human TSPCs demonstrated diminished proliferative capacity, with lower *Ki67*[+] and EdU-positive cell proportions (Fig. 4B, C). Furthermore, the RT-qPCR analysis showed the elevated tenogenic gene expression (Col1a1, Mkx, Bgn) in rFGF7-treated TSPCs relative to controls (Fig. 4F). Additionally, after 14 days of tenogenic differentiation, rFGF7-treated human TSPCs demonstrated enhanced tenogenic differentiation, as evidenced by matrix production results through Sirius Red and Masson's trichrome staining, whereas AZD4547-treated TSPCs showed reduced differentiation capacity (Fig. 4E). Immunofluorescence staining of tenogenic markers (COL1A1, MKX, EGR1) and fibrotic markers (ACTA2, COL3) in human TSPCs revealed that rFGF7 treatment increased tenogenic marker expression and decreased fibrotic marker expression compared to controls. Conversely, AZD4547 treatment alone produced the opposite effect. Nevertheless, the addition of AZD4547 did not fully reverse the effects of rFGF7 (Fig. 4G, H). Notably, subsequent co-administration of rFGF2 and rFGF4 following AZD4547 treatment appeared to upregulate the expression of fibrotic markers in human TSPCs (Figs. S4 and S5).

In summary, we confirmed that FGF7 inhibits fibrosis and concomitantly promotes the proliferation and tenogenic differentiation of both mouse and human tendon stem/progenitor cells. These findings

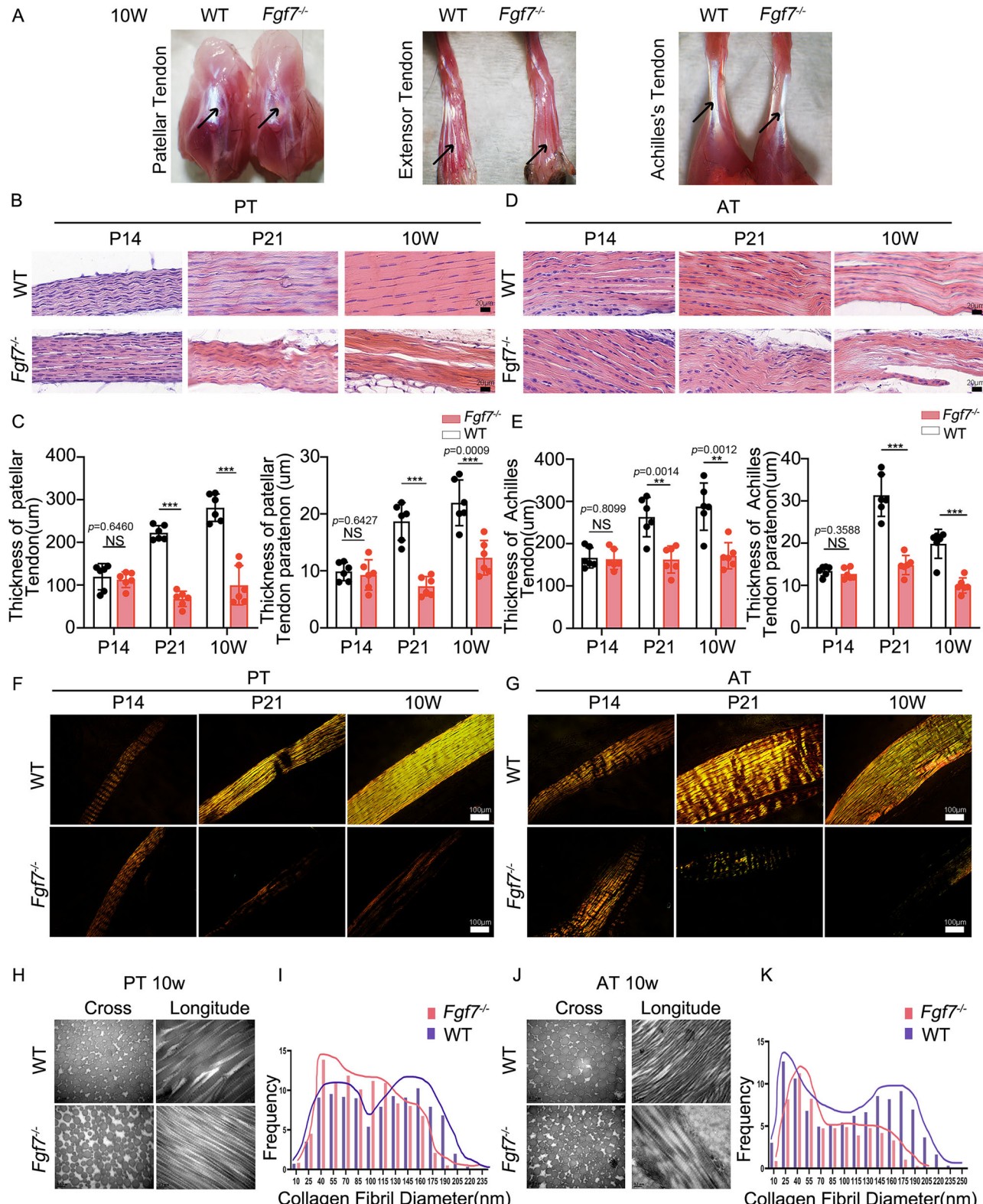

also underscore the potential therapeutic applications of FGF7 in tendon regeneration and anti-fibrotic interventions.

### *Fgf7* deficiency leads to the enrichment of *Lox⁺Uts2r⁺* subcluster in mature tendon tissue

Driven by the observed matrix assembly disorder phenotype in FGF7 knockout (*Fgf7*^KO) tendon tissues and the antifibrotic effects of recombinant FGF7 (rFGF7) at the cellular level, we sought to investigate the underlying mechanisms of tendon matrix assembly disorder due to *Fgf7* deficiency. To achieve this, we conducted single-cell RNA-sequencing (scRNA-seq) on Achilles tendon tissues from *Fgf7* knockout (*Fgf7*^−/−) and wildtype (WT) mice at postnatal day 21 (P21) and at 8 weeks of age. Following quality control and removal of presumed doublets, we obtained 31,756 tendon cells. Unsupervised clustering revealed 17 distinct subpopulations, visualized using UMAP (Figs. S6A and S7A). Based on significantly enriched genes and their expression

**Fig. 2 | Macro-, micro-, and nano-structural abnormalities in load-bearing tendons of *Fgf7*⁻/⁻ mice. A** Gross morphological observation of the patellar tendon, flexor digitorum longus tendon and Achilles tendon in *Fgf7*⁻/⁻ and wild-type mice. Tendons under mechanical load exhibited developmental abnormalities, including the patellar tendon (left), flexor digitorum longus tendon (middle), and Achilles tendon (right). Arrows indicate respective tendons. **B** Hematoxylin and eosin (HE) staining of the patellar tendon at postnatal days 14, 21, and 10 weeks in *Fgf7*⁻/⁻ and wild-type mice. Scale bars, 20 μm. **C** Statistical analysis of patellar tendon thickness and patellar tendon paratenon thickness in HE-stained sections at postnatal days 14, 21, and 10 weeks in *Fgf7*⁻/⁻ and wildtype mice (*n* = 6 per time point, independent experiments, bars represent mean ± SD; WT wildtype. NS: not significant, **\**p* < 0.01, ***\**p* < 0.001, two-tailed Student's *t* test. Patellar tendon thickness at postnatal days: 14 days: *t* = 0.4736, df = 10; 21 days: *t* = 15.31, df = 10; 10 weeks: *t* = 7.932, df = 10. Patellar tendon paratenon thickness at postnatal days: 14 days: *t* = 0.4783, df = 10; 21 days: *t* = 7.496, df = 10; 10 weeks: *t* = 4.689, df = 10.). **D** HE staining of the Achilles tendon at postnatal days 14, 21, and 10 weeks in *Fgf7*⁻/⁻ and wildtype mice. Scale bars, 20 μm. **E** Statistical analysis of Achilles tendon thickness and Achilles tendon paratenon thickness in HE-stained sections at postnatal days 14, 21, and 10 weeks in *Fgf7*⁻/⁻ and wildtype mice (*n* = 6 per time point, independent experiments, bars represent mean ± SD; WT wildtype. NS: not significant, ***\**p* < 0.001, two-tailed Student's *t* test. Achilles tendon thickness at postnatal days: 14 days: *t* = 0.2470, df = 10; 21 days: *t* = 4.356, df = 10; 10 weeks: *t* = 4.443, df = 10. Achilles tendon paratenon thickness at postnatal days: 14 days: *t* = 0.9618, df = 10; 21 days: *t* = 7.398, df = 10; 10 weeks: *t* = 6.257, df = 10.). **F** Polarized light images of the patellar tendon at postnatal days 14, 21, and 10 weeks in *Fgf7*⁻/⁻ and wildtype mice. Scale bars, 100 μm. Representative polarized light images from three independent experiments with consistent results. **G** Polarized light images of the Achilles tendon at postnatal days 14, 21, and 10 weeks in *Fgf7*⁻/⁻ and wildtype mice. Scale bars, 100 μm. Representative polarized light images from three independent experiments with consistent results. Transmission electron microscopy (TEM) images (**H**) showing transverse and longitudinal sections of collagen fibers in the patellar tendon at 10 weeks in *Fgf7*⁻/⁻ and wildtype mice (*n* = 3), scale bars: 200 nm, and their respective collagen diameter distribution (**I**). Scale bars, 0.2 μm. TEM images (**J**) showing transverse and longitudinal sections of collagen fibers in the Achilles tendon at 10 weeks in *Fgf7*⁻/⁻ and wildtype mice (*n* = 3), scale bars: 200 nm, and their respective collagen diameter distribution (**K**). Scale bars, 0.2 μm. Source data are provided as a Source Data file.

profiles (Fig. S6B–D), we identified and named 17 subpopulations: tenocyte subpopulations (Tenocytes 1-3, Enthesis cells, Muscle-tendon junction cells) characterized by markers such as *Postn, Scx, Prg4, Tnmd, Cd34, Tppp3, Thbs4, Has1, Eln*, and *Col2a1*; B and T cell subpopulations marked by *Ighd, Igkc, Cd3d, Cd3e*; Macrophage subpopulation marked by *C1qa, Cd68*; Neutrophils subpopulation marked by *Retnlg, S100a8*; Erythroid cells marked by *Slc4a1, Epor*; Neural cells marked by *Sox10, Mpz*; Endothelial cells marked by *Pecam1, Cdh5*; Lymphocytes cells marked by *Prox1, Lyve1*; Smooth muscle cells marked by *Cnn1, Myh11*; Erythroid cells marked by *Alsa2, Fech*; MuSC and progenitors marked by *Pax7, Myf5*; and Muscle cells marked by *Acta1, Myl1* (Figs. S6C and S7B). To investigate the impact of *Fgf7* knockout on the proportions of cellular subpopulations within tendon tissue, we performed a detailed analysis of cell proportions changes upon *Fgf7* deficiency and wildtype tendon tissue (Fig. S6E). Our analysis both revealed that tendon-specific cellular subpopulations (Tenocytes 1, Tenocytes 2, Tenocytes 3, Enthesis, Muscle Tendon Cells) exhibited the most significant and prominent changes in cell proportions. Consequently, we focused on the tenoc-lineage subpopulations within the tendon tissue (Fig. 5A).

We reclassified these tenocyte subpopulations (Tenocytes 1, Tenocytes 2, Tenocytes 3, Enthesis, Muscle Tendon Cells) into nine subclusters based on gene expression characteristics and visualized using UMAP (Fig. 5B). These were named *Lox⁺Uts2r⁺* cells (Cluster 0), characterized by *Lox, Uts2r, Mfap4*; Tenocytes subcluster (Cluster 1), characterized by *Postn, Apod*; Tenoblasts subcluster (Cluster 2), characterized by *Thbs4, Comp*; *Cd44⁺Tppp3⁺* tendon progenitors (Cluster 3), characterized by *Tppp3, Cd44, Pi16*; Tenogenic (Cluster 4), characterized by *Eln, Cygb*; Enthesoblasts (Cluster 5), characterized by *Col2a1, Cnmd*; *Prg4⁺* tendon progenitors (Cluster 6), characterized by *Prg4, Tspan15*; Muscle-tendon junction cells (Cluster 7), characterized by *Col22a1, Frem2*; and Mineralizing chondrocytes (Cluster 8), characterized by *Runx2, Bglap*. Single-cell RNA sequencing analysis of tendon tissues from 21-day-old and 8-week-old *Fgf7*⁻/⁻ and wild-type mice showed that the *Uts2r⁺Lox⁺* cell population is present in immature tendons at postnatal day 21, but disappears in normally developed wild-type tendons by 8 weeks. In contrast, this population persists in *Fgf7*⁻/⁻ tendons at 8 weeks (Fig. S8). To better illustrate the distribution of the *Lox⁺Uts2r⁺* subpopulation among the overall cells in tendon tissue prior to subclustering, we highlighted the *Lox⁺Uts2r⁺* subpopulation within the gray circle and noted that it is primarily expressed in the Tenocytes 2 subgroup of the initial clustering (Fig. S9A, B). Comparative analysis of cell proportions between 8-week wildtype and *Fgf7*⁻/⁻ samples revealed increases in subclusters 0, 4, and 8, with subcluster 0 showing the largest increase (Fig. 5C, D). Immunofluorescence staining also confirmed a significant increase in this subpopulation within the patellar and Achilles tendons of *Fgf7*⁻/⁻ mice relative to wild-type controls (Figs. 5F, G and S10A, B). Consistent with the observed tendon matrix assembly disorder in the patellar and Achilles tendons of *Fgf7*⁻/⁻ mice, analysis of subcluster 0 revealed a positive correlation with multiple markers in promoting collagen fiber formation and cross-linking compared to other subpopulations, such as *Lox*[25], *Uts2r*[26], *Mfap4*[27], and *Acta2*[28], among others (Figs. 5E and S11). The *Lox⁺Uts2r⁺* subcluster was associated with the matrix assembly disorder phenotype induced by FGF7 knockout. Notably, this *Lox⁺-Uts2r⁺* subcluster was also present during fibrotic tendon repair, where matrix assembly disorder is widespread. Analysis of single-cell RNA sequencing data from both fibrotic tendon repair and healthy mouse tendon samples revealed dynamic changes in the *Lox⁺Uts2r⁺* subcluster during tendon fibrosis repair in wild-type (WT) mice. In the UMAP visualization (Fig. S12), the circled regions highlight cell populations that increased following injury. Specifically, the *Lox⁺Uts2r⁺* subcluster (marked as Cluster0) was significantly expanded during tendon injury compared with normal tendons (Fig. S12A, B).

Collectively, these results show that *Fgf7* deficiency promotes the expansion of the *Lox⁺Uts2r⁺* subcluster in mature tendon tissue.

## ProTracer reveals FGF7 plays a crucial role in directing tendon cell fate in vivo

Given the evidence of cellular turnover within tendon tissue (Galloway et al. reported approximately 5% turnover in tendon cells at P21)[29], and the near absence of the *Lox⁺Uts2r⁺* subcluster in normal mature tendons, we noted its substantial presence in the matrix assembly disorders tendon tissues of *Fgf7*⁻/⁻ mice. This observation led us to hypothesize that, in the absence of *Fgf7*, replenishing cells fail to follow the normal differentiation trajectory toward mature tendon. Instead, they aberrantly differentiate into *Lox⁺Uts2r⁺* cells. We performed pseudotime trajectory analysis on the nine subclusters using StemID (Stem Cell Identification) analysis (Fig. 6A), the ridge plots from Monocle 2 pseudotime analysis (Fig. 6B) and corroborating previous reports that *Tppp3⁺* subclusters primarily occupy early differentiation stages[4], We found that the *Lox⁺Uts2r⁺* subcluster indeed resides at the terminal end of tendon cell differentiation and is almost exclusively present in the matrix assembly disorders tendon tissues of *Fgf7* knockout (*Fgf7*⁻/⁻) mice. Additionally, we also analysis of human tendon samples from healthy (*n* = 11) and tendinopathy samples (*n* = 64) revealed a higher presence of the *Lox⁺Uts2r⁺* subcluster in tendinopathy clinic samples (Fig. 6F–H). Obtaining genuinely healthy rotator cuff tendon samples from human donors presents considerable practical and ethical difficulties, as such tissues are seldom accessible through surgical procedures. Therefore, we utilized normal hamstring

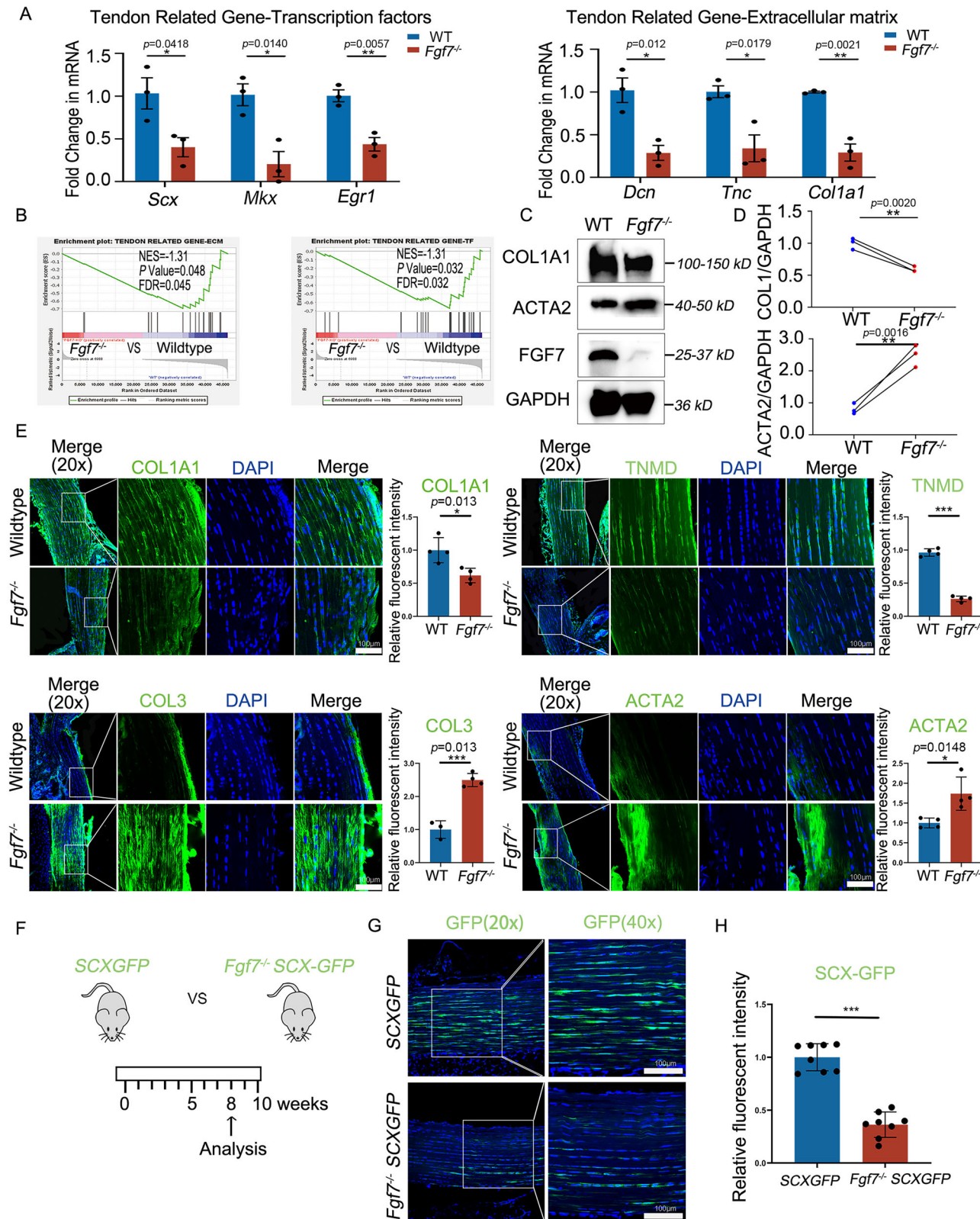

tendons as an alternative model for our investigation. The proportions of tendon cell subpopulations in both normal hamstring tendons and normal rotator cuff tendons were observed to be largely similar (Fig. S13).

To directly track the differentiation pathways of tendon cells aberrantly differentiating into *Lox⁺Uts2r⁺* cells due to *Fgf7* deficiency, we utilized the ProTracer mouse tracing system to continuously record

*Ki67* gene expression, representing proliferating tendon cells. This allowed uninterrupted monitoring of replenishing cell differentiation trajectories within the tissue[21,22] and allowed us to investigate whether the absence of *Fgf7* causes proliferating tendon cells to follow erroneous tendon degeneration pathways. Traditional *Ki67-CreER* lineage tracing techniques require repeated tamoxifen administration to maintain continuous activation of the *Ki67-CreER* system, which can

**Fig. 3 | Tendon matrix assembly disorder phenotype in tendons under mechanical load in *Fgf7*⁻/⁻ mice. A** qRT-PCR analysis of RNA expression levels of *Scx*, *Egr1*, *Mkx*, *Col1a1*, *Dcn* and *Tnc* in mature tendon tissues (*n* = 3 independent experiments, bars represent mean ± SD; WT wildtype. *$p < 0.05$, **$p < 0.01$, two-tailed Student's *t* test). **B** Gene Set Enrichment Analysis (GSEA) evaluation of tendon transcription-related and extracellular matrix-related gene sets in the transcriptomes of tendons from wildtype and *Fgf7*⁻/⁻ mice. GSEA gene set enrichment analysis, FDR false discovery rate, NES normalized enrichment score. **C** Western blot analysis showing protein expression levels of FGF7, ACTA2 and COL1 in tendon tissues of wildtype and *Fgf7*⁻/⁻ mice. Representative western blot of three independent experiments is shown; all yielded reproducible results. **D** Statistical analysis of protein expression levels of FGF7, ACTA2 and COL1 in tendon tissues of wildtype and *Fgf7*⁻/⁻ mice (*n* = 3 independent experiments, bars represent mean ± SD; WT wildtype. *$p < 0.05$, **$p < 0.01$, two-tailed Student's *t* test). **E** Immunofluorescence staining and statistical analysis of tendon-related marker genes COL1A1, TNMD and fibrosis marker genes COL3 (*n* = 3) and ACTA2 in wildtype and *Fgf7*⁻/⁻ mice tendon tissues (*n* = 4 independent experiments, bars represent mean ± SD; WT wildtype. *$p < 0.05$, ***$p < 0.001$, two-tailed Student's *t* test). Scale bars, 100 μm. **F** Experimental plan schematic for the *ScxGFP* transgenic mice model, comparing *Fgf7*⁻/⁻ *ScxGFP* transgenic mice and *ScxGFP* transgenic mice, with tissue collection at tendon maturation (8–10 weeks). **G, H** Confocal fluorescence analysis of GFP protein expression, with statistical data in tendons of *Fgf7*⁻/⁻ *ScxGFP* mice and *ScxGFP* mice (*n* = 8 independent experiments, bars represent mean ± SD; ***$p < 0.001$, two-tailed Student's *t* test). Scale bars, 100 μm. Source data are provided as a Source Data file.

negatively affect animal health and result in inconsistent activation efficiency. In contrast, the ProTracer system uses an initial tamoxifen-induced *DreER-rox* recombination to activate the *Ki67-Cre* allele. Once activated, *Ki67* gene activity is continuously recorded by *Ki67-Cre*, eliminating the need for repeated tamoxifen administration and allowing for uninterrupted recording of *Ki67*-expressing cells, enabling the tracking of their ultimate differentiation fate within the tissue (Fig. 6C). As tendon abnormalities in *Fgf7*⁻/⁻ mice primarily emerge after postnatal day 21, we administered tamoxifen to ProTracer mice and ProTracer-background *Fgf7*⁻/⁻ mice at postnatal day 21 and collected tendon tissues at 5, 6 and 8 weeks (Figs. 6C and S14). Immunofluorescence staining revealed a higher proportion of replenishing GFP⁺ cells in the tendons of ProTracer-background *Fgf7*⁻/⁻ mice differentiating into the *Lox*⁺*Uts2r*⁺ subcluster, whereas only a minimal number of GFP⁺ cells in normal ProTracer mice tendons exhibited this differentiation (Figs. 6D, E and S10A, B). These findings show the critical role of FGF7 in maintaining normal tendon homeostasis and preventing matrix assembly disorder function, offering significant therapeutic potential.

## Activation of the *TGF-β* pathway via *Creb3l1* sustains the *Lox*⁺*Uts2r*⁺ subcluster in *Fgf7* knockout tendon tissue

To elucidate the mechanisms underlying the persistence of the *Lox*⁺*Uts2r*⁺ subcluster in *Fgf7* knockout tendons, we conducted Gene Ontology (GO) pathway enrichment analysis on the differentially expressed genes between the *Lox*⁺*Uts2r*⁺ subcluster and the other eight tendon subclusters (Fig. 7A, B). Our analysis revealed a pronounced downregulation of pathways associated with tissue regeneration and an upregulation of tendon matrix assembly disorders pathways, including extracellular matrix remodeling, epithelial-mesenchymal transition, and dysregulated growth factor responses. Notably, we observed an increased sensitivity to the *TGF-β* signaling pathway within the *Lox*⁺*Uts2r*⁺ subcluster. We found that the *Lox*⁺*Uts2r*⁺ subcluster not only transmits TGF-β signals to other cellular subclusters but also responds to TGF-β signaling autonomously by CellChat analysis (Fig. 7C, D). SCENIC analysis further identified six key transcription factors within the *Lox*⁺*Uts2r*⁺ subcluster (Fig. 7E), with *Creb3l1* being significantly associated with the heightened sensitivity to TGF-β signaling (Fig. 7B). Subsequent analysis of *Creb3l1* expression levels demonstrated a marked upregulation of *Creb3l1* within the *Lox*⁺*Uts2r*⁺ subcluster, with predominant expression in this subpopulation (Fig. 7F, G). Specifically, we implemented inhibition experiments targeting *Creb3l1*. By introducing *Creb3l1* siRNA into tendon stem cells derived from FGF7 knockout mice, we observed a reduction in *Lox*⁺*Uts2r*⁺ double-positive cells. This was accompanied by an increase in the expression of the tendon-specific protein TNMD and a decrease in the expression of ACTA2 and COL3 (Fig. S15). In summary, our findings indicate that the persistence activity of the *Lox*⁺*Uts2r*⁺ subcluster in *Fgf7* knockout tendons is maintained through the regulation of the *Creb3l1* transcription factor by *TGF-β* signaling, highlighting its potential as a therapeutic target for mitigating tendon fibrosis.

## Recombinant fibroblast growth factor 7 (rFGF7) mitigates fibrotic phenotype in ex vivo tendon tissues

We utilized ex vivo tendon tissues from SCX-GFP mice, which undergo fibrotic changes when removed from their in vivo environment. Previous studies have reported that tendon tissue from SCX-GFP mice loses its SCX phenotype immediately after explant culture[30]. The loss of SCX expression in tendon-related genes is also a characteristic of tendon fibrosis. These tissues were cultured for 7 days in media containing or lacking rFGF7. rFGF7 presence in culture medium resulted in sustained GFP expression in SCX-GFP tendon tissues (Fig. 8A). Confocal fluorescence microscopy confirmed that rFGF7-treated tendon tissues exhibited significantly stronger GFP fluorescence compared to controls (Fig. 8C). To further substantiate these observations at the protein level, we performed immunostaining for tenogenic markers (COL1A, TNMD, THBS4, EGR1) and fibrotic markers (COL3, ACTA2) in ex vivo tendon tissues cultured with or without rFGF7 for 7 days. rFGF7-treated tissues showed elevated expression of tenogenic markers and reduced expression of fibrotic markers compared to controls (Fig. 8E–J). These findings were corroborated by RT-qPCR analysis, which revealed that rFGF7-treated tendon tissues had significantly higher expression of tenogenic genes (*Egr1*, *Col1a*, *Thbs4*, *Tnmd*) and lower expression of fibrotic genes (*Acta2*, *Col3*) relative to controls (Fig. 8B). At the ultrastructural level, transmission electron microscopy demonstrated that rFGF7 treatment maintained a bimodal distribution of collagen fibril diameters in the tendon tissues, with a rightward shift indicating larger collagen fibril diameters compared to controls (Fig. 8D). This suggests that rFGF7 effectively preserves the structural integrity of collagen fibers in ex vivo tendon tissues.

Collectively, these data underscore the pivotal role of rFGF7 in mitigating fibrotic changes and promoting the maintenance of tenogenic phenotypes in tendon explant.

## GelMA hydrogel encapsulated rFGF7 (GelMA-*c*-rFGF7) promotes functional tendon regeneration

Because of the rapid inactivation of rFGF7 post-injection and the pain associated with multiple injections, we aimed to develop a localized delivery system to facilitate functional tendon regeneration and treat tendon ruptures. We engineered a GelMA-*c*-rFGF7 system for sustained rFGF7 release. To validate the biocompatibility and non-toxicity of this hydrogel system, we co-cultured rat TSPCs with the GelMA-*c*-rFGF7 system for 7 days. The live/dead staining assay (Fig. S16A, C) and CCK8 proliferation assay (Fig. S16D) indicated no significant differences in cell viability between the GelMA-*c*-rFGF7 system and the control. Additionally, ELISA assays demonstrated that protein release from the GelMA-*c*-rFGF7 hydrogel was significantly slower than from the GelMA/FGF7 mixture, confirming the sustained release capability of the GelMA-*c*-rFGF7 system (Fig. S16E). We have conducted scanning electron microscopy (SEM) on the GelMA-*c*-rFGF7 hydrogel to elucidate its internal microarchitecture. Representative SEM images (Fig. S17A) revealed that the hydrogel exhibited a highly porous and interconnected three-dimensional network.

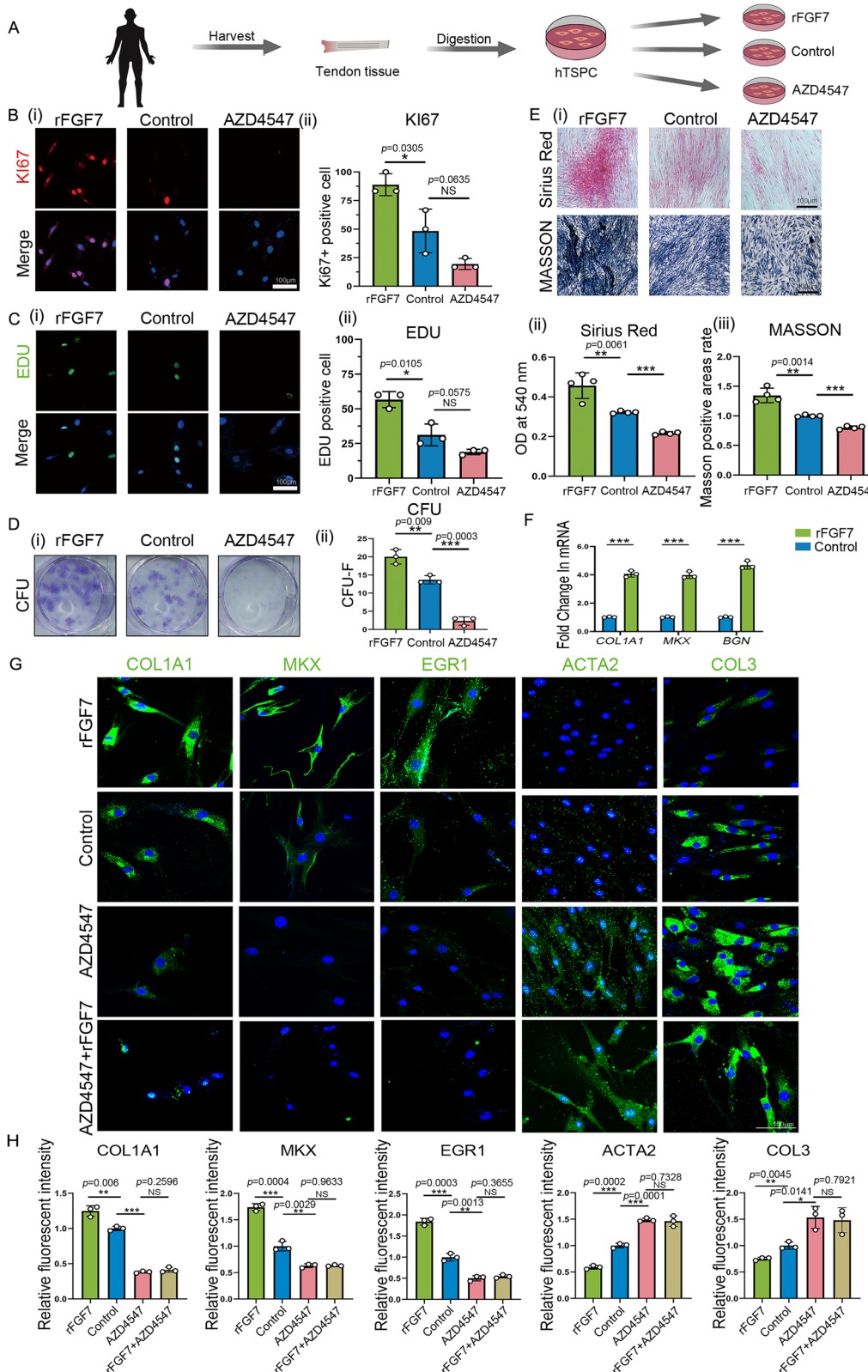

Approximately three-quarters of tendon injury fibrosis occurs within the tendon proper[31]. To evaluate the potential effect of the GelMA-*c*-rFGF7 hydrogel on tendon healing and fibrosis, we created a 4-mm window defect model in rat patellar tendons and implanted the GelMA-*c*-rFGF7 hydrogel into the injury site. The experimental procedure, including the creation of the tendon injury model and in situ adhesion operation, is depicted in Fig. S16F. Schematic overview of the entire experimental design as illustrated in Fig. 9A. At 2 and 4 weeks post-surgery, we conducted motor behavior analysis on rats treated with GelMA-*c*-rFGF7 hydrogel, injury rats, GelMA-treated rats, and age-matched sham operation rats (Sham group). Using the device shown in Fig. 9B, we collected paw prints for gait analysis (Fig. S18A. Spatial parameters (stride length, stride width, and step length) and paw parameters (paw length and toe spread) were recorded at 2 and

**Fig. 4 | FGF7 promotes hTSPC tenogenic differentiation and inhibits fibrotic differentiation. A** Schematic representation of the experimental procedure for promoting tendon lineage differentiation of hTSPCs using FGF7 growth factor. **B** Immunofluorescence staining of KI67 protein in hTSPCs treated with rFGF7, PBS or AZD4547 (i) and semi-quantitative fluorescence analysis (ii) ($n = 3$ independent experiments, bars represent mean ± SD; NS: not significant, *$p < 0.05$, two-tailed Student's $t$ test). Scale bars, 100 μm. **C** Immunofluorescence staining of EDU protein in hTSPCs treated with rFGF7, PBS, or AZD4547 (i) and statistical analysis of EDU positivity rate (ii) ($n = 3$, independent experiments, bars represent mean ± SD; NS: not significant, *$p < 0.05$, two-tailed Student's $t$ test). Scale bars, 100 μm. **D** CFU-F assay results in hTSPCs treated with rFGF7, PBS, or AZD4547 (i) and semi-quantitative analysis (ii) ($n = 3$ independent experiments, bars represent mean ± SD; **$p < 0.01$, ***$p < 0.001$, two-tailed Student's $t$ test). **E** Sirius Red and Masson's Trichrome staining of hTSPCs treated with rFGF7, PBS or AZD4547 after 14 days of tendon lineage culture (i), with Sirius Red optical density (OD) statistical analysis (ii) ($n = 4$ independent experiments, bars represent mean ± SD; **$p < 0.01$, ***$p < 0.001$, two-tailed Student's $t$ test) and statistical analysis of the percentage of Masson's Trichrome positive area (iii) ($n = 4$ independent experiments, bars represent mean ± SD; **$p < 0.01$, ***$p < 0.001$, two-tailed Student's $t$ test). Scale bars, 100 μm. **F** qPCR analysis of RNA expression levels of tendon lineage marker genes Col1a1, Mkx and Bgn in hTSPCs treated with rFGF7 or PBS ($n = 3$ independent experiments, bars represent mean ± SD; ***$p < 0.001$, two-tailed Student's $t$ test). **G, H** Immunofluorescence staining of tendon-related proteins (COL1A1, MKX, EGR1) and fibrotic proteins (ACTA2, COL3) in hTSPCs treated with rFGF7, PBS, AZD4547 or co-treated with rFGF7 and AZD4547 with corresponding semi-quantitative fluorescence analysis ($n = 3$ independent experiments, bars represent mean ± SD; NS: not significant, *$p < 0.05$, **$p < 0.01$, ***$p < 0.001$, two-tailed Student's $t$ test). Scale bars, 100 μm. Source data are provided as a Source Data file.

4 weeks post-surgery (Figs. 9H and S18B). At 2 weeks, GelMA-$c$-rFGF7-treated rats exhibited greater stride length and width and lower step length, paw length, and toe spread asymmetry index compared to the injury group, indicating partial recovery of motor function. By 4 weeks, the GelMA-$c$-rFGF7 group showed motor function comparable to the sham group (Fig. 9C–G). We also performed maximum treadmill exertion tests (Fig. 9I) and found significant improvements in time, distance, and speed for the GelMA-$c$-rFGF7 group compared to the injury and GelMA groups at both 2 and 4 weeks (Fig. 9J, K). Mechanical testing at 4 weeks post-surgery revealed that the GelMA-$c$-rFGF7 group had superior stress at failure, modulus, and failure force compared to the injury and GelMA groups, nearing the mechanical properties of the tendon of the sham group (Fig. 9M, N). Gross morphology of neo-tissues showed smoother and more uniform tendon surfaces in the GelMA-$c$-rFGF7 group (Fig. 9L).

To investigate whether the GelMA-$c$-rFGF7 hydrogel system can enhance tendon tissue healing, we performed histological analysis using HE and Masson staining, as well as immunofluorescence staining for tendon markers (COL1A1, TNMD, THBS4) and fibrosis markers (COL3, ACTA2). HE and Masson staining at 2 weeks revealed that the newly formed tendon tissues in the GelMA-$c$-rFGF7, injury, and GelMA groups all exhibited a high number of cells, with predominantly round nuclei. Notably, the GelMA-$c$-rFGF7 group showed more organized cell and collagen alignment (Fig. 10A, C). At 4 weeks, the newly formed tendon tissues in the GelMA-$c$-rFGF7 group, compared to the injury and GelMA groups, exhibited elongated spindle-shaped nuclei, a significantly reduced cell number, and enhanced alignment of cells and collagen fibers. The increased intensity of polarized light also indicated improved collagen organization and maturation (Fig. 10B, D). These findings suggest that by 4 weeks post-surgery, the newly formed tendon tissue in the GelMA-$c$-rFGF7 group more closely resembled native tendon rather than fibrotic tissue. We found that the GelMA-$c$-rFGF7 group not only exhibited more consistent nuclear orientation (DAPI) but also showed increased expression of tendon markers (COL1A1, TNMD, THBS4) and decreased expression of fibrosis markers (COL3, ACTA2) compared to the injury and GelMA groups (Figs. 10E, F and S19). This indicates that the newly formed tendon tissue in the GelMA-$c$-rFGF7 group exhibited more tendon-like characteristics and reduced fibrosis. We also certificated the expression of $Uts2r^+$ $Lox^+$ cells during the repair process of injured rat patellar tendons treated with GelMA-$c$-rFGF7 hydrogel, GelMA, and the untreated injury group (Fig. S20). The success of tendon tissue repair is critically dependent on the enhancement of collagen diameter. To further investigate the nanoscale characteristics of the newly formed tendon tissue, we utilized transmission electron microscopy. Our findings revealed that the collagen diameter distribution in the GelMA-$c$-rFGF7 group exhibited a rightward shift, indicating larger overall collagen diameters compared to the injury and GelMA groups. Moreover, the GelMA-$c$-rFGF7 group displayed the characteristic bimodal collagen distribution typical of normal tendon tissue (Fig. 10G, H). This likely contributes to the superior mechanical properties observed in the regenerated tendon tissue of the GelMA-$c$-rFGF7 group (Fig. 9M, N).

Overall, these macro-, micro-, and nanoscale characterizations, along with motor behavior and mechanical testing results, confirm that GelMA-$c$-rFGF7 promotes functional tendon regeneration and enhances the biological function of repaired tendon tissues.

## Discussion

The FGF family represents a crucial pathway in tendon development[32,33]. However, it remains unclear which specific FGF subtype plays a significant role in tenogenesis. Some studies have reported that FGF4 effectively regulates the expression of tendon-related proteins, such as TNMD and THBS2, during chick limb development[7,34]. However, its role in mouse limb tendon development has been shown to be ineffective and suppresses the expression of $Scx$ in mouse tendon stem cells[35]. FGF8 has been demonstrated to play an important role in tendon development in chickens[36], yet its expression has not been detected during mouse tendon development[34]. Additionally, studies have indicated that activated FGF8 inhibits the expression of $Scx$, $Tnmd$, and $Tnc$ in the tendon tissue of the mandible[9]. FGF2 has been shown to enhance the production of extracellular matrix in injured tendon tissues[11]. However, the tenogenic effect of FGF2 on stem cells remains controversial[8,37,38]. The specific role of different FGF subtypes in tendon development is still not fully understood. In this study, our results indicate that FGF7 plays an important role in promoting normal tendon development by utilizing FGF7 knockout (FGF7KO) mice.

In vivo experiments, our findings highlight the diminished motor function and identified a $Lox^+Uts2r^+$ subpopulation in $Fgf7$ knockout mice, underscoring the necessity of FGF7 for maintaining tendon integrity and inhibiting degeneration. In vitro experiments demonstrated that FGF7 can promote the tenogenic differentiation of tendon stem cells while inhibiting their fibrotic trajectory. Additionally, we also conducted tendon explant cultures to confirm the phenotype maintenance and anti-fibrotic effects of FGF7. Finally, we employed the GelMA-$c$-rFGF7 system to deliver FGF7 to damaged tendon sites, resulting in the functional regeneration of injured tendons. Overall, our findings provide valuable insights into FGF-specific subtype FGF7 promotes tendon commitment of TSPCs and regulate tendon differentiation during development and pathology. This presents a potential strategy for the treatment of tendon injuries in the future.

In the context of load-bearing tendon regeneration, the characterization of fibrosis transcends mere collagen overproduction, focusing instead on pathological matrix remodeling. Tissues attempt to aberrantly activate processes, such as the increase of α-SMA, and exhibit abnormal cross-linking, as indicated by the rise in $Lox^+$ cells and the increased presence of more easily reshaped yet mechanically weaker collagen type III, in an effort to compensate for mechanical deficiencies. This compensation fails to yield functional tissue, ultimately leading to structural stiffness, increased brittleness, and a decline in functional capacity for tendon formation. The culmination

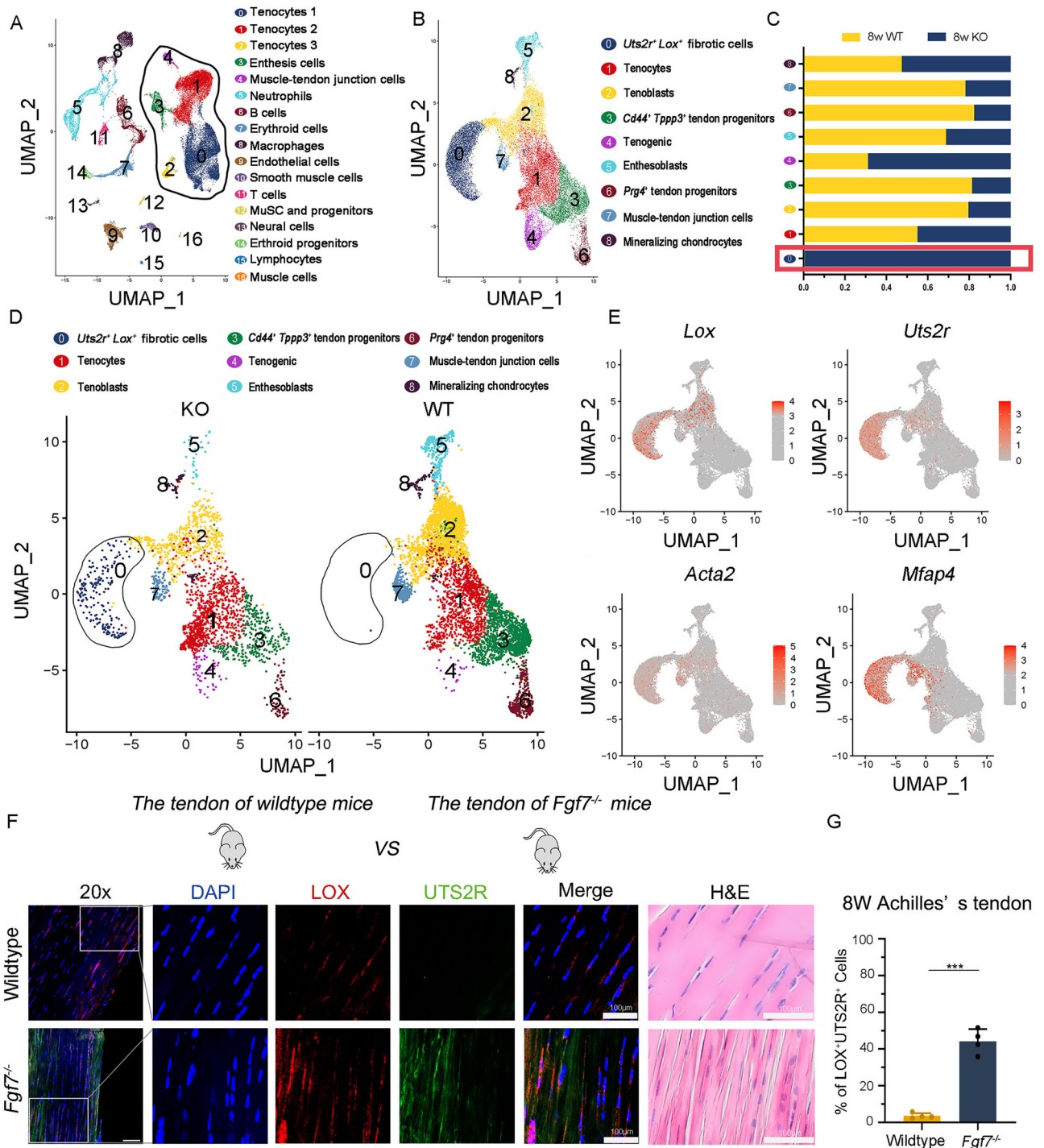

**Fig. 5 | *Fgf7* knockout leads to the emergence of the *Lox⁺Uts2r⁺* subcluster in Achilles tendon tissues.** **A** A Uniform Manifold Approximation and Projection (UMAP) plot is presented, displaying 31,756 cells divided into 17 distinct subpopulations. The circle subpopulation includes subgroups 0, 1, 2, 3, and 4, which exhibit tendon characteristics. **B** UMAP plot displaying 24,583 cells from the Achilles tendon subpopulations, further divided into 9 distinct subclusters. **C** Representative bar plots showing the rate of cells belonging to each subcluster in Achilles tendon tissues from 8-week wildtype (WT) and 8-week *Fgf7⁻/⁻* mice.

**D** UMAP plots showing the distribution of the 9 subclusters from 8-week *Fgf7⁻/⁻* and 8-week wildtype mice Achilles tendon samples. **E** UMAP plots illustrating the expression of matrix assembly marker genes *Lox*, *Uts2r*, *Acta2* and *Mfap4*. **F, G** Confocal fluorescence imaging and showing the co-expression of *Lox* and *Uts2r* in Achilles tendon tissues from 8-week *Fgf7⁻/⁻* and wildtype mice, with corresponding statistical analysis and representative HE-stained images (*n* = 4 independent experiments, bars represent mean ± SD; ***p < 0.001, two-tailed Student's *t* test). Scale bars, 100 μm. Source data are provided as a Source Data file.

of these pathological processes results in tissues that cannot withstand physiological loads, rendering them prone to rupture and healing failure. FGF7, comprising approximately 194 amino acids and weighing about 22.4 kDa, exhibits a tertiary structure composed of multiple β-sheets and α-helices forming a compact globular protein[39]. Our

previous research demonstrated that FGF7 mediates interactions among tendon stem/progenitor cells and enhances tenogenic phenotype within a 3D culture system[18]. In this study, we further elucidate the critical role of FGF7 in promoting regeneration and inhibiting degeneration in mechanically loaded tendons. The absence of *Fgf7* results in

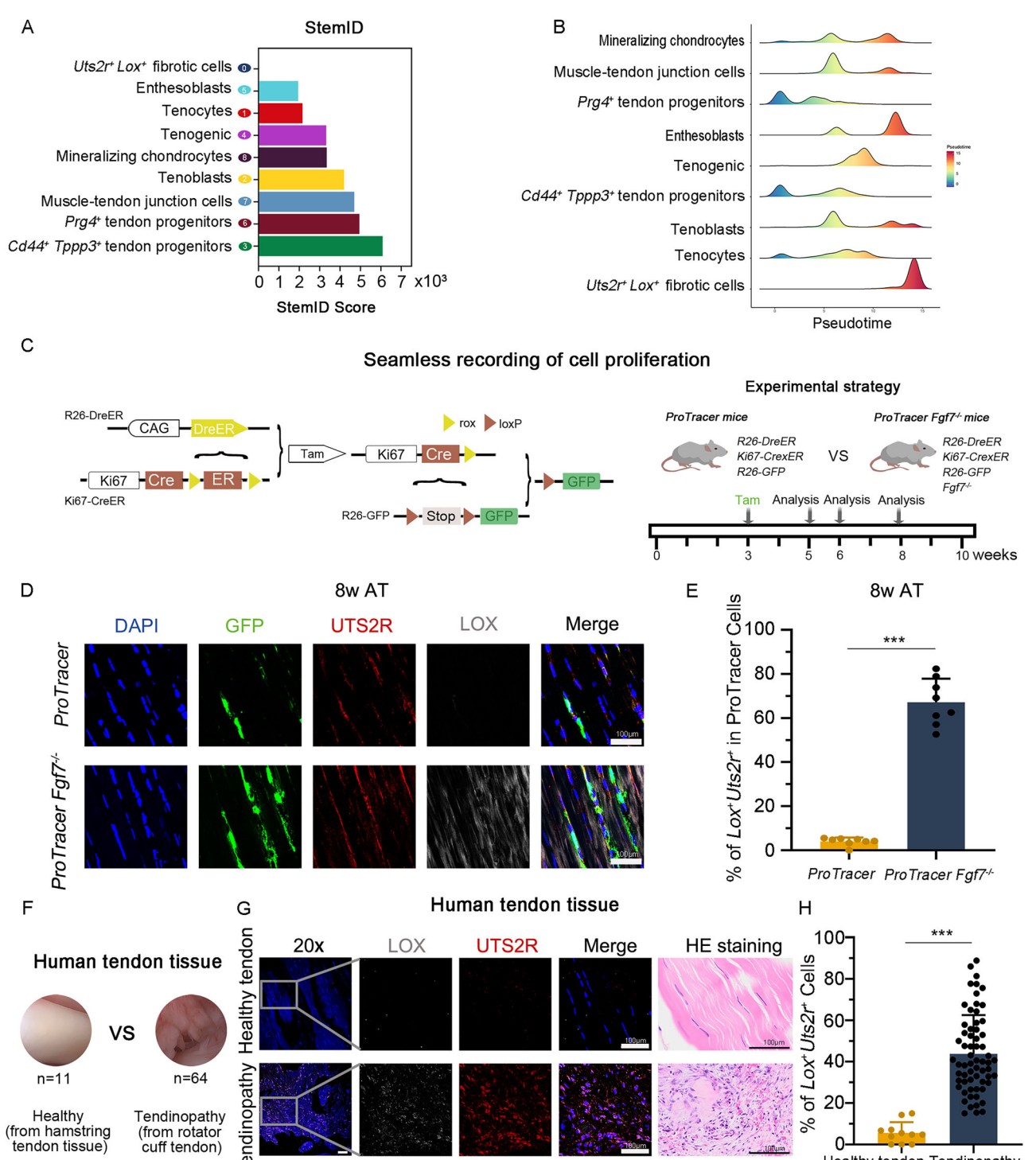

**Fig. 6 | *Fgf7* knockout leads to proliferation and partial differentiation of tenocytes into the *Lox⁺Uts2r⁺* subcluster. A** STEMID score analysis for the 9 subclusters. **B** Ridge plot showing the differentiation hierarchy of cell subclusters based on pseudotime analysis. **C** Strategy for the seamless recording of cell proliferation and the schematic of the experimental design. **D, E** Confocal fluorescence staining showing co-expression of GFP, UTS2R and LOX in tendon tissues from ProTracer mice and *Fgf7⁻/⁻* ProTracer mice, with corresponding statistical analysis of the proportion of ProTracer labeled cells differentiating into *Lox⁺Uts2r⁺* cells (*n* = 8 independent experiments, bars represent mean ± SD; ***p* < 0.001, two-tailed

Student's *t* test). Scale bars, 100 μm. **F** Schematic diagram of clinical human tendon tissue samples from hamstring tendon tissue as human healthy tendon and pathological rotator cuff tendon tissue as human tendinopathy. **G, H** Fluorescence staining showing co-expression of UTS2R and LOX in human normal tendon samples (*n* = 11) and tendinopathy samples (*n* = 64), with corresponding statistical analysis and representative HE-stained images (each dot represents one human sample, bars represent mean ± SD; ***p* < 0.001, two-tailed Student's *t* test). Scale bars, 100 μm. Source data are provided as a Source Data file.

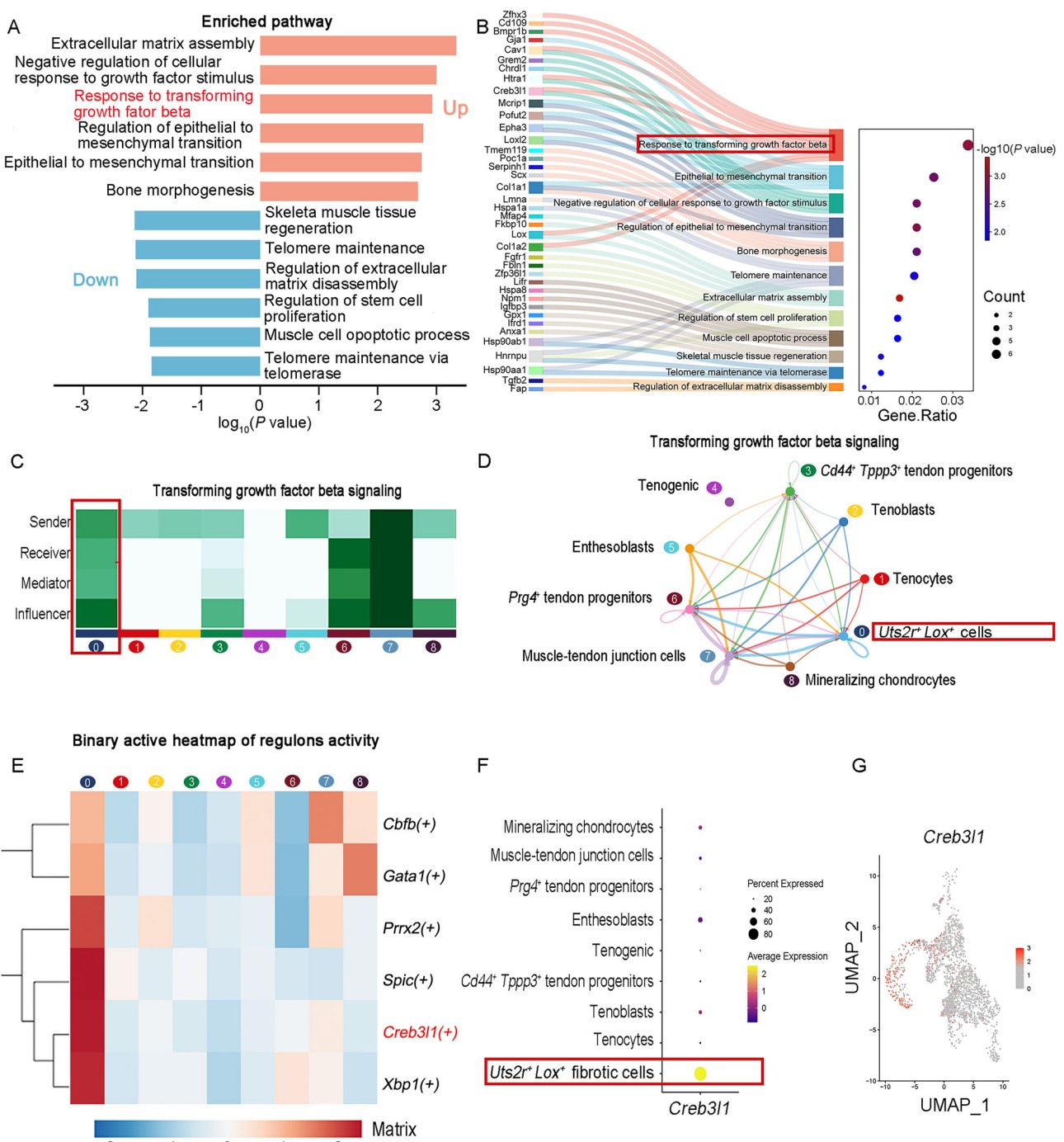

**Fig. 7 | Activation of the TGF-β pathway via *Creb3l1* sustains the *Lox⁺Uts2r⁺* subcluster in *Fgf7* knockout tendon tissue. A** GO pathway enrichment analysis of differentially expressed genes in the *Lox⁺Uts2r⁺* subcluster, show casing representative upregulated pathways (fold-change >1.2; *p* value < 0.05) and down-regulated pathways (fold-change >1.2; *p* value < 0.05). Hypergeometric test, FDR correction. **B** A bubble plot of GO analysis from the upregulated and down-regulated pathways in (**A**). The left column lists genes within the *Lox⁺Uts2r⁺* sub-cluster that contribute to the respective pathways, the middle column indicates pathway names, and the right column shows the proportion of pathways affected. Hypergeometric test, FDR correction. **C** A heatmap demonstrating the roles and relative importance of the nine tendon subclusters in TGF-β signaling networks. **D** CellChat analysis showing the interactions between tendon subclusters in TGF-β signaling. The direction of the arrows indicates the acting subcluster, and the thickness of the lines represents the communication probability within that path-way. **E** SCENIC results for the nine tendon subclusters, highlighting *Creb3l1* genes involved in TGF-β signaling (**B**) in red. **F** A dot plot illustrating the expression levels of the *Creb3l1* transcription factor in the nine tendon subclusters. The red box highlights the highest expression levels in the *Lox⁺Uts2r⁺* subcluster. **G** UMAP plots of the nine tendon subclusters illustrating the concentrated and highest expression of the *Creb3l1* transcription factor in the *Lox⁺Uts2r⁺* subcluster.

morphological alterations in load-bearing tendon tissues such as the patellar and Achilles tendons not the position tendon such as tail by day 21, similar to the effects observed following the knockout of the SPARC protein, which impacts tendons in mechanically loaded regions

of mice after 21 days post-birth[40]. We hypothesize that this is due to the gradual increase in limb movement abilities and frequency in mice after birth, particularly beyond 3 weeks, as they start foraging for food independently. Consequently, the mechanical load on tendons in

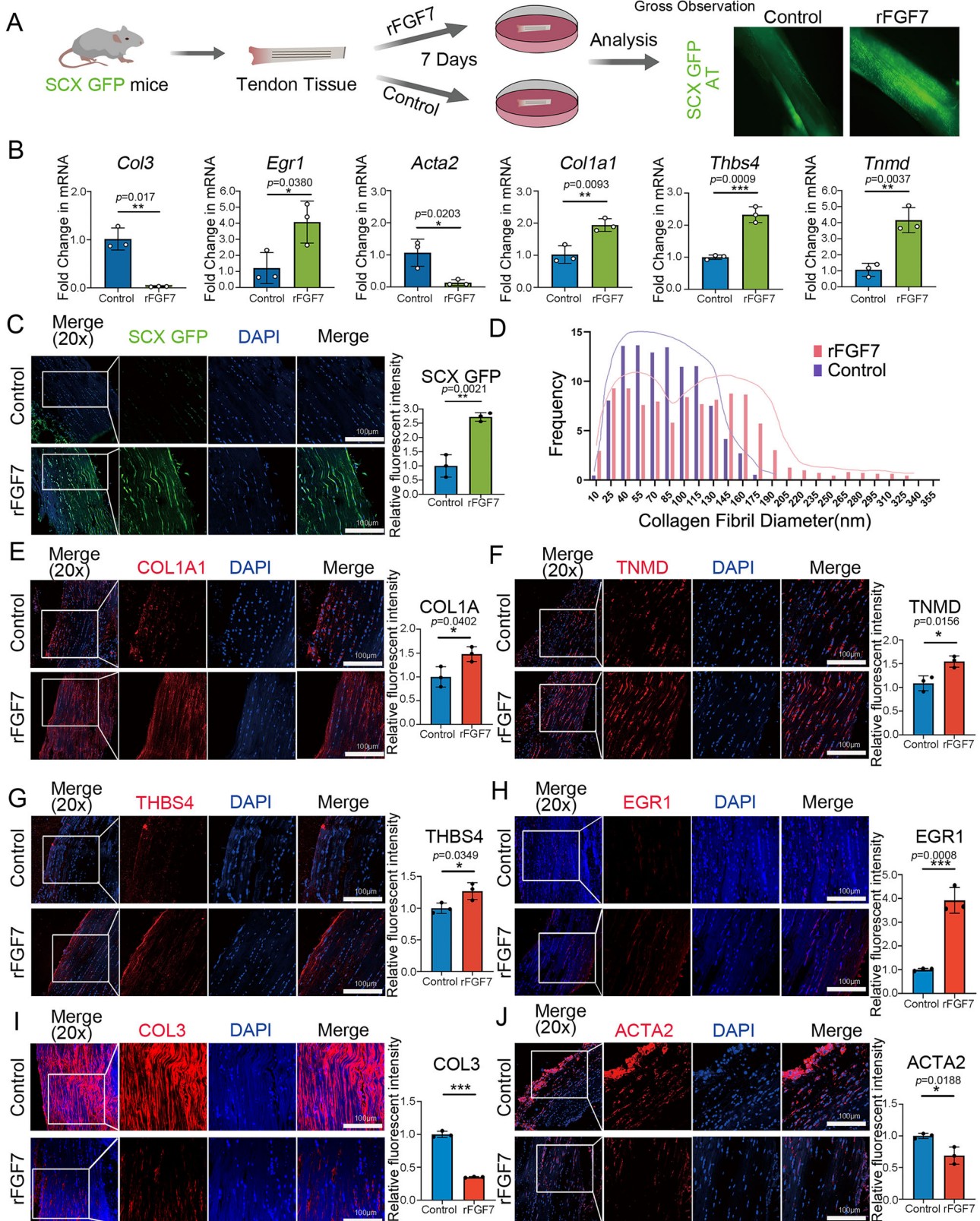

critical areas such as the patellar and Achilles tendons increases, potentially leading to tendon damage due to a loss of tendon homeostasis. However, the distinction between the two mechanisms lies in their respective impacts on tendon integrity: SPARC protein deficiency leads to spontaneous tendon rupture due to reduced type I collagen secretion and a subsequent loss of tendon homeostasis, whereas *Fgf7* deficiency impairs the ability of tendon stem/progenitor cells to

differentiate into fully mature tendons. Instead, these cells aberrantly differentiate towards a pathological degeneration state, resulting in matrix assembly disorder. This was validated by multiple experiments in this research, which included cultivating mouse and human tendon stem/progenitor cells with FGF7 in vitro and using ProTracer lineage tracing mouse and single-cell RNA-sequencing analysis of *Fgf7* knock-out mouse tendons in vivo.

**Fig. 8 | Application of rFGF7 inhibits fibrotic changes in mouse tendon explants. A** Schematic representation of the explant experiment. Achilles tendons from SCXGFP mice were extracted and cultured with rFGF7 or PBS. **B** qPCR analysis of tendon related marker genes (*Egr1*, *Col1a1*, *Thbs4*, *Tnmd*) and tendon fibrosis genes (*Col3*, *Acta2*) in cultured tendon explants from both groups (*n* = 3 independent experiments, bars represent mean ± SD; **p* < 0.05, ***p* < 0.01, ****p* < 0.001, two-tailed Student's *t* test). **C** Confocal immunofluorescence imaging showing GFP protein expression in tendon explants from *SCXGFP* mice (*n* = 3, independent

experiments, bars represent mean ± SD; ***p* < 0.01, two-tailed Student's *t* test). Scale bars, 100 μm. **D** Transmission electron microscopy (TEM) analysis of collagen fiber diameter distribution in transverse sections of tendon explants cultured with rFGF7 or PBS (*n* = 3 per group). **E–J** Immunofluorescence staining and semi-quantitative analysis of tendon related proteins (COL1A1, TNMD, THBS4, EGR1) and fibrosis proteins (COL3, ACTA2) (*n* = 3, independent experiments, bars represent mean ± SD; **p* < 0.05, ****p* < 0.001, two-tailed Student's *t* test). Scale bars, 100 μm. Source data are provided as a Source Data file.

Specifically, we identified a fibrotic cell subpopulation expressing *Uts2r* and *Lox*[25,26], in *Fgf7*-deficient tendon tissue. This enriched *Lox⁺Uts2r⁺* subpopulation is driven by the activation of the transcription factor Creb3l1 via the TGF-β pathway. By utilizing the ProTracer cell proliferation tracking platform[20–22], we could continuously labeled proliferating tendon cells and traced their lineage trajectories. Consequently, we confirmed that the absence of *Fgf7* promotes the differentiation of newly proliferating tendon cells into the *Lox⁺Uts2r⁺* subpopulation. Additionally, through pathological analysis of clinical samples, we found that this subpopulation is also validated in human tendinopathy samples. Fibrosis results from the interplay of various factors during tendon injury repair, including cytokine networks, inflammatory responses, mechanical stress, and extracellular matrix remodeling imbalances. Our study suggests that FGF7 deficiency, particularly in load-bearing tendon tissues, is a significant promoting factor. This indicates an interactive relationship between mechanical stress and the protective role of FGF7. Additionally, chronic inflammatory responses are crucial in driving fibrosis, and FGF7 may indirectly influence this process by modulating inflammation, an area worth exploring in future research.

Inhibiting fibrosis and promoting regeneration remains a significant challenge in the repair of collagen-rich tissues. FGF7, by directing stem cells towards tendon differentiation while preventing fibrotic differentiation within tendons, which represent a predominantly collagenous organ, may offer a novel approach to addressing this challenge. Based on these findings, we developed a sustained-release formulation using GelMA to avoid the need for repetitive injections, thereby mitigating pain and infection, and enhancing the longevity of FGF7's bioactivity. Tendon tissues treated with GelMA-*c*-rFGF7 displayed histological and microstructural characteristics akin to native tendons rather than fibrotic repair. Histologically, the tissue presented a wavy pattern, while microstructurally, the collagen fiber diameter displayed a bimodal distribution. Although motor function was inferior compared to the normal group, the mechanical properties of regenerated tendons showed significant recovery. Overall, in a full-thickness tendon defect model, without cell transplantation, the endogenous recruitment of tendon stem/progenitor cells (TSPCs) under the effect of rFGF7 achieved superior structural and functional tendon regeneration. These suggest that FGF7 has potential as a pharmaceutical candidate for the treatment of tendon disorders, hinting at its translational application prospects.

A limitation of this research is the lack of validation in higher-level animal models, such as larger animals or non-human primates. Due to the difficulty in obtaining normal rotator cuff tissue samples from human subjects, we utilized hamstring tendon tissue as a control. Variability between tendon tissues from different anatomical sites is, of course, unavoidable. Additionally, assessing the translational potential and safety of GelMA hydrogels will require longer follow-up periods in future research. Finally, the safety and efficacy of GelMA-*c*-rFGF7 need to be evaluated in future studies at the large animal level.

In conclusion, our study identifies FGF7 as an important regulator in repressed fibrosis and promoting connective tissue morphogenesis. This represents a straightforward and potentially clinically viable therapeutic approach for preventing fibrotic diseases.

## Methods
The research adheres to all relevant ethical regulations. All procedures were approved by the Institutional Animal Use and Care Committee, Zhejiang University and the Ethics Committee of the Second Affiliated Hospital, School of Medicine, Zhejiang University.

### Ethics statement
This study was approved by the Ethics Committee of the Second Affiliated Hospital, School of Medicine, Zhejiang University, following the Declaration of Helsinki. Animal experiments were approved by the Animal Use and Care Committee of Zhejiang University (ZJU20240291). All samples were collected from the patients who had provided informed consent at Second Affiliated Hospital, Zhejiang University. Ethical approval for this study was obtained from the Ethics Committee of the Second Affiliated Hospital, School of Medicine, Zhejiang University (codes: 2019-168, 2020-080). Human tendon stem/progenitor cells (hTSPCs) were collected from hamstring tendon tissues used as autologous grafts during ACL reconstruction surgeries. Diseased human tendon samples were collected as surgical waste during arthroscopic shoulder surgeries (*n* = 64). Additionally, Human healthy tendons were collected from hamstring tendon tissues used as autologous grafts during ACL reconstruction surgeries (*n* = 11). Healthy tendon samples were obtained from hamstring tendon tissues used as autologous grafts during anterior cruciate ligament (ACL) reconstruction surgeries, while the pathological tendon samples were derived from pathological rotator cuff tendon tissue. Because obtaining truly healthy rotator cuff tendon samples from human donors poses significant practical and ethical challenges. Healthy rotator cuff tendons are rarely available from surgical procedures. The relevant slices have been saved. If necessary, please contact the corresponding author for sharing and use. *Fgf7⁻/⁻* mice and C57BL/6J wildtype mice were sourced from GemPharmatech (China). ProTracer mice were generously donated by Professor Bin Zhou (University of Chinese Academy of Sciences). The 8 weeks female Sprague Dawley rats were sourced from Hangzhou Normal University (China). Animals were kept under a 12-h light/dark cycle with food and water access. Animals showing premature death or tumors were excluded and humanely euthanized at the study's end. The mice were housed under controlled environmental conditions, with the temperature maintained at 22 °C and the relative humidity at approximately 50%. Due to the occurrence of tendon fibrosis across different physiological sexes, our study did not include a comparative analysis based on physiological sex.

### Isolation and culture of tendon stem/progenitor cells (TSPCs)
Human tendon stem/progenitor cells (hTSPCs) were isolated using previously established methods[41]. The isolated hTSPCs were cultured in three distinct media: one containing 10 ng/mL FGF7, another containing 1 μM FGFR inhibitor AZD4547, and a standard medium. The composition of the culture media included Dulbecco's Modified Eagle Medium (L-DMEM), 1% penicillin-streptomycin solution (PS), and 10% fetal bovine serum (FBS). Upon reaching 90% confluence, the cells were transitioned to H-DMEM induction medium containing 50 μg/mL vitamin C and 10% FBS, which supports tendon stem cell stability and maintains the tenogenic phenotype. After 14 days of continued culture, cells were

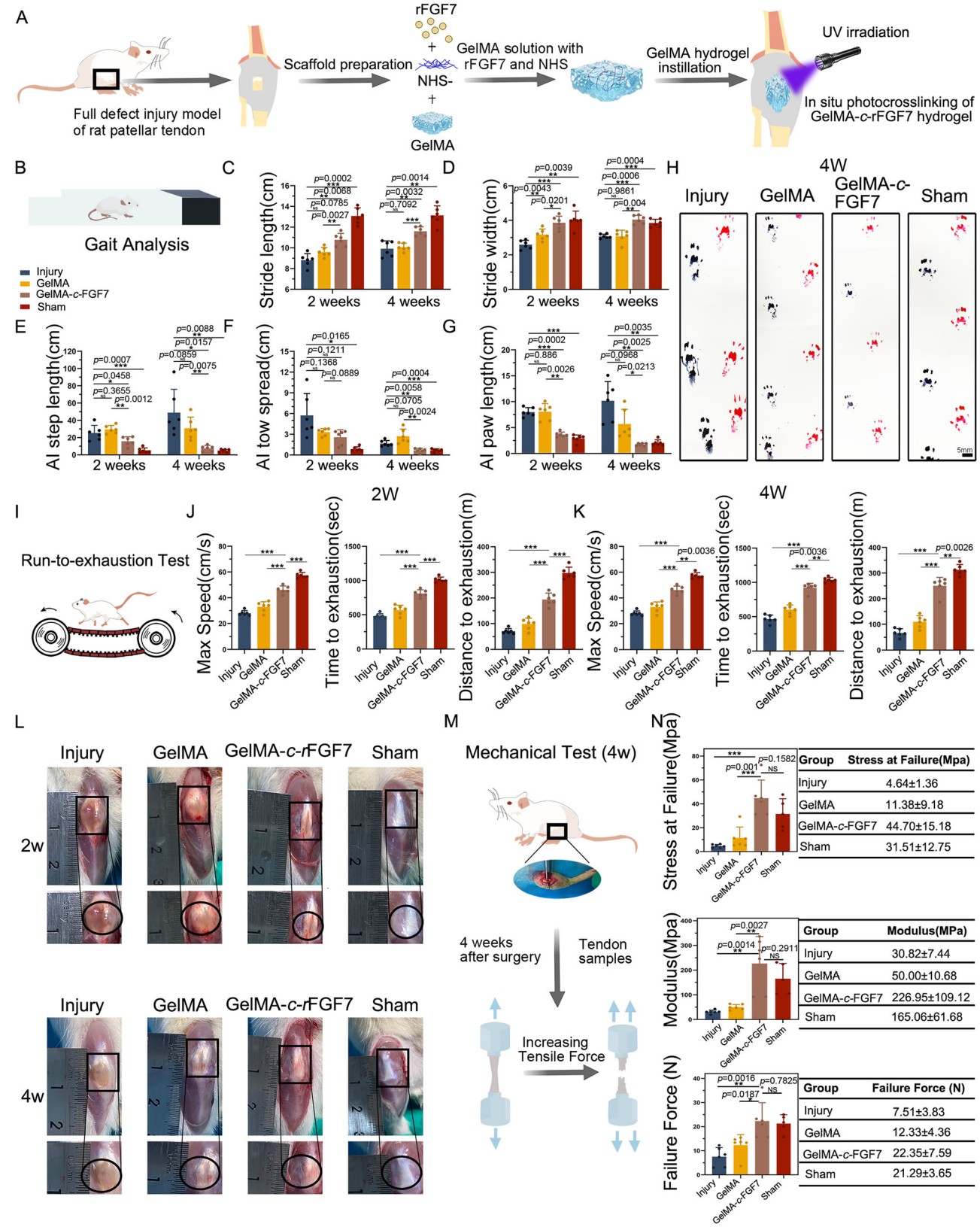

collected, fixed, and subjected to Masson's trichrome and Sirius Red staining. Mouse tendon stem/progenitor cells (mTSPCs) were obtained from Achilles tendon tissues of wild-type C57BL/6J mice and *Fgf7*[−/−] C57BL/6J mice. The tissues were minced and digested with 0.2% type I collagenase in low-glucose L-DMEM for 2 h. Following digestion, the cells were centrifuged at 230 × *g* for 5 min, resuspended in low-glucose L-DMEM containing 10% FBS and 1% PS, and cultured for approximately 7 days. Upon reaching 80–90% confluence, the cells were passaged, and second-passage mTSPCs were utilized for this study. The same protocol was applied to Sprague-Dawley rats to obtain rat tendon stem/progenitor cells (Rat TSPCs), with second-passage Rat TSPCs employed for experimental procedures.

**Fig. 9 | Application of GelMA-*c*-rFGF7 promotes repair of injured patellar tendon and restores motor function in SD rats. A** Schematic illustration of repairing injured patellar tendon tissue using GelMA-*c*-rFGF7. **B** Schematic of the footprint analysis experiment. Gait spatial parameter statistics at different time points (2w, 4w) for different experimental groups (Injury, GelMA, GelMA-*c*-rFGF7, Sham): stride length (**C**), stride width (**D**), step length (**E**), paw spatial parameters: toe spread (**F**), paw length (**G**) (*n* = 6 per group, bars represent mean ± SD; *$p < 0.05$, **$p < 0.01$, ***$p < 0.001$, two-tailed Student's *t* test). **H** Representative paw prints of each group at 4 weeks post-surgery. Scale bars, 5 mm. **I** Schematic of the rat exhaustion experiment. **J** Statistical graphs of maximum exhaustion speed, time, and distance (left to right) at 2 weeks post-surgery for different experimental groups (Injury, GelMA, GelMA-*c*-rFGF7, Sham) (*n* = 6 per group, bars represent mean ± SD; **$p < 0.01$, ***$p < 0.001$, two-tailed Student's *t* test). **K** Statistical graphs of maximum exhaustion speed, time and distance (left to right) at 4 weeks post-surgery for different experimental groups (Injury, GelMA, GelMA-*c*-rFGF7, Sham) (*n* = 6 per group, bars represent mean ± SD; **$p < 0.01$, ***$p < 0.001$, two-tailed Student's *t* test). **L** Representative gross observations of repaired patellar tendon tissue at post-surgery (2w and 4w) for different experimental groups (Injury, GelMA, GelMA-*c*-rFGF7, Sham). **M** Schematic of mechanical testing for different experimental groups (Injury, GelMA, GelMA-*c*-rFGF7, Sham) at 4 weeks post-surgery. **N** Statistical graphs of mechanical testing results at 4 weeks post-surgery for different experimental groups (Injury, GelMA, GelMA-*c*-rFGF7, Sham), showing Stress at Failure, Failure Force and Modulus (from top to bottom) (*n* = 6 per group, wildtype group *n* = 5, bars represent mean ± SD; NS: not significa*n*t, *$p < 0.05$, **$p < 0.01$, ***$p < 0.001$, two-tailed Student's *t* test). Source data are provided as a Source Data file.

## Motor ability test

Wire Hang Test: after 3 days of adaptive feeding, mice underwent 3 days of wire hang adaptation, consisting of 3 sessions per day with 30-min intervals between sessions. On day 4, mice were placed on a wire apparatus positioned 40 cm above the bench. The time was recorded from the moment the mice grasped the horizontal bar until they fell, with a maximum observation period of 300 s. The bar was sanitized with 75% ethanol between trials. Treadmill test: following 3 days of adaptive feeding and 1 day of treadmill training (3 sessions per day, 5–10 min apart), the formal test commenced. The treadmill started at an initial speed of 0 m/min and accelerated at 2 m/min². The treadmill was kept level, and animals were considered exhausted if their hind limbs contacted the grid for more than 10 s.

## TEM analysis

Tendon tissues were processed for TEM using standard procedures[42]. Samples were pre-fixed in 2% glutaraldehyde at 4 °C for 2 h, then fixed in 1% osmium tetroxide at 4 °C for 2 h. Samples were dehydrated in graded ethanol, acetone, embedded, sectioned ultrathin, and stained. Ultrastructural observations were conducted, and collagen fibril diameters were measured using IPP 6.0 software. At least three independent samples per group and approximately 1200 collagen fibrils were analyzed.

## Mechanical testing

Mechanical testing was conducted using an Instron tension/compression system (Model 5543, Instron, Canton, MA) to assess the mechanical properties of repaired tendon tissues, following a previous study[42]. Rat hind limbs were prepared by excising excess soft tissue while preserving the patellar tendon. The femur-patellar tendon-tibia complex (FPTC) was secured in a custom-designed clamp. Mechanical properties, including modulus (MPa), failure energy (J), failure strength (MPa), failure stress (N), and stiffness (N/m), were evaluated through tension tests performed at a constant extension rate of 5 mm/min until failure.

## Gene set enrichment analysis (GSEA)

RNA-Seq data from Achilles tendon tissues of 10-week-old wildtype and *Fgf7*$^{-/-}$ mice were preprocessed and normalized. Gene set enrichment analysis (GSEA) was performed using the Broad Institute software, with gene sets defined from the Molecular Signatures Database (MSigDB). Enrichment scores and significance were calculated, with significant gene sets determined by *p* values and false discovery rate (FDR) control. Enriched gene sets were visualized using enrichment plots. Single-cell RNA-Seq data from wildtype and *Fgf7*$^{-/-}$ mice were similarly analyzed using GSEA.

## RNA-seq experiments and single-cell RNA-sequencing experiments

RNA-seq: The RNA-seq protocol was adapted from a previously established method[23]. All mice used for RNA-seq analysis were strictly matched at 8 weeks of age, emphasizing our control of age consistency during the experimental design phase. Total RNA was extracted from 8 weeks C57BL/6J wildtype and *Fgf7*$^{-/-}$ mice tendon tissue samples using Trizol reagent (Takara Bio Inc.) and reverse transcribed into cDNA using SuperScript II (Invitrogen). All mice used for RNA-seq analysis were strictly matched at 8 weeks of age. Double-stranded cDNA was synthesized with the NEBNext mRNA Second Strand Synthesis Kit (NEB) and purified using AMPure XP beads (Beckman Coulter). Sequencing libraries were constructed using the Nextera XT kit (Illumina) and sequenced on the Illumina X-Ten platform. Differential gene expression was analyzed with DESeq2. Genes with counts ≥1 were considered expressed, excluding genes with counts of 0. DEGs were defined as fold change ≥2 and *q* value ≤ 0.05. Functional enrichment of DEGs was performed using Metascape for GO and KEGG pathway analysis. Single-cell RNA sequencing: Achilles tendon tissues from postnatal day 21 and week 8 C57BL/6J wildtype and *Fgf7*$^{-/-}$ mice were collected, pooled from 6 mice per group. Fresh tissues were stored in sCelLive™ Tissue Preservation Solution (Singleron) on ice, washed with HBSS, minced, and digested in low-glucose DMEM (Gibco) with 0.2% collagenase I (Gibco) at 37 °C for 2 h. The suspension was filtered, centrifuged, and resuspended in PBS. Single-cell suspensions were loaded onto Singleron Matrix® microfluidic chips. Barcoded beads were collected for mRNA reverse transcription and cDNA generation, followed by PCR amplification, fragmentation, and adapter ligation. scRNA-seq libraries were constructed with the GEXSCOPE® Single-Cell RNA Library Kit (Singleron) and sequenced on the Illumina NovaSeq 6000 platform. Digital gene expression matrices were analyzed using Seurat for dimensionality reduction, clustering, and differential expression. Quality control excluded cells with <1200 or >7000 genes, and total counts <25,000 or >1,000,000. Ligand-receptor interactions were analyzed using CellChat. Stemness scoring and ranking were performed with StemID.

## Rat footprint analysis

We designed a walking track system using acrylic material to create a restrictive pathway covered with white paper. Black and red inks were applied to the left and right hind paws of the rats, respectively, with one side being the sham-operated group and the other the experimental group. Rats were placed at the start of the pathway and allowed to walk freely until they reached the end, leaving footprints on the white paper. Footprint-covered paper was collected and photographed using an iPhone 15 to ensure clear images. Six independent experimental samples were used for each group. Footprint parameters, including stride length, stride width, and step length, as well as paw parameters such as paw length and toe spread, were measured using Photoshop software. To better quantify and compare the asymmetry between the sham-operated and experimental groups, the Asymmetry Index (AI) was calculated using the formula: AI (%) = (NS − ES) × (NS + ES) / 0.5 × 100, where NS represents the measurement on the normal side, and ES represents the measurement on the experimental side.

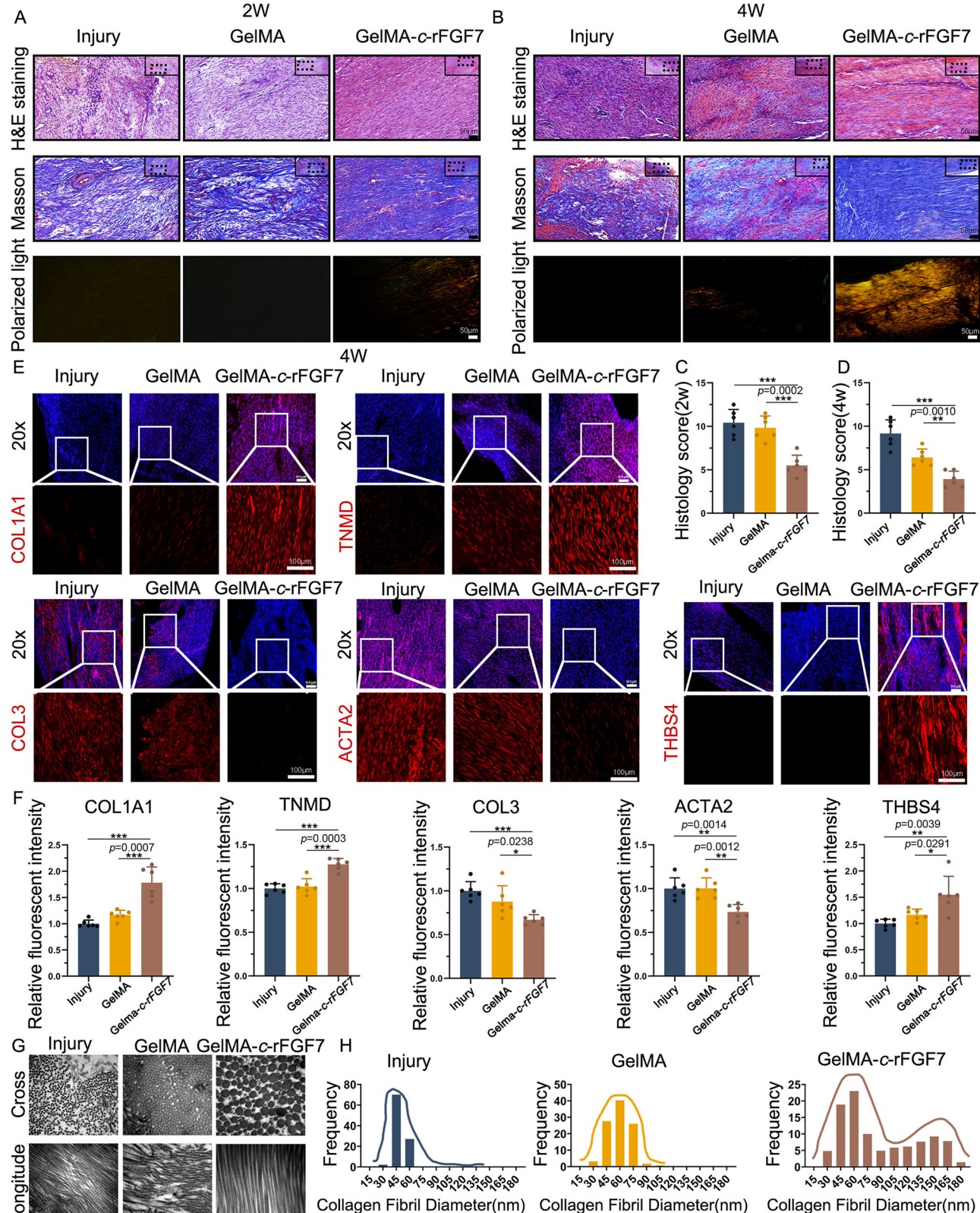

## Animal experiment

The mouse strains used in this study were as follows: all mice were maintained on a C57BL/6 background. ProTracer mice were constructed as previously described[22]. *Fgf7* knockout mice were purchased from GemPharmatech (China). *Fgf7*[+/-] heterozygous mice were obtained from GEM. Littermate *Fgf7*[-/-] homozygous mice and wild-type controls were generated from crosses of these heterozygotes.

*Fgf7* knockout ProTracer mice were generated by breeding ProTracer mice with *Fgf7* knockout mice to obtain *Fgf7* knockout mice on a ProTracer background. SCX-GFP mice were constructed as previously described[30]. *Fgf7* knockout SCX-GFP mice were generated by breeding SCX-GFP mice with *Fgf7* knockout mice to obtain *Fgf7* knockout mice on an SCX-GFP background. Wildtype mice were purchased from GemPharmatech (China). *Fgf7* knockout mice and wildtype mice were

**Fig. 10 | GelMA-*c*-rFGF7 regenerates and repairs microscopic and nanoscopic tendon structures while inhibiting fibrotic protein production. A** Hematoxylin and eosin (HE) staining, Masson staining and polarized light microscopy of regenerated tendon tissue at the repair site at 2 weeks post-surgery for each group. Scale bars, 50 μm. **B** HE staining, Masson staining and polarized light microscopy of regenerated tendon tissue at the repair site at 4 weeks post-surgery for each group. Scale bars, 50 μm. **C** Microscopic-level histological scoring of tendon tissue at the repair site at 2 weeks post-surgery ($n = 6$ per group, bars represent mean ± SD; $**p < 0.01$, $***p < 0.001$, two-tailed Student's $t$ test). **D** Microscopic-level histological scoring of tendon tissue at the repair site at 4 weeks post-surgery ($n = 6$ per group, bars represent mean ± SD; $**p < 0.01$, $***p < 0.001$, two-tailed Student's $t$ test). **E** Immunofluorescence staining for *t*endon-related proteins (COL1A1, TNMD, THBS4) and fibrosis-related proteins (COL3, ACTA2) at the repair site at 4 weeks post-surgery for each group (Injury, GelMA, GelMA-*c*-rFGF7). Scale bars, 100 μm. **F** Semi-quantitative analysis of immunofluorescence staining for tendon-related proteins (COL1A1, TNMD, THBS4) and fibrosis-related proteins (COL3, ACTA2) at the repair site at 4 weeks post-surgery for each group ($n = 6$ per group, bars represent mean ± SD; $*p < 0.05$, $**p < 0.01$, $***p < 0.001$, two-tailed Student's $t$ test). **G** Representative transmission electron microscopy images of the transverse sections of tendon collagen at the repair site at 4 weeks post-surgery for each group (Injury, GelMA, GelMA-*c*-rFGF7). Scale bars, 0.2 μm. **H** Statistical distribution of collagen fiber diameters observed at the repair site at 4 weeks post-surgery using transmission electron microscopy for each group ($n = 6$ per group). Source data are provided as a Source Data file.

sacrificed at embryonic day 18, postnatal day 14, 21, and 8–10 weeks to collect patellar tendon, Achilles tendon, flexor digitorum tendon, and tail tendon tissues for further evaluation and cell acquisition. *Fgf7* knockout ProTracer mice and ProTracer mice received a single intraperitoneal injection of tamoxifen (TMX) (0.1–0.2 mg/kg) at 21 days and were sacrificed at 8 weeks to collect Achilles and patellar tendon tissues for further evaluation. *Fgf7* knockout SCX-GFP mice and SCX-GFP mice were sacrificed at 8 weeks to collect Achilles and patellar tendon tissues for further evaluation.

For surgical modeling, female New Zealand rats (8–10 weeks old, purchased from the Animal Center of Hangzhou Normal University) were used after hair removal and site disinfection with iodine. A lateral incision was made at the knee joint to expose the patellar tendon, and a full-thickness window defect was created in the patellar tendon. GelMA-based materials were injected into the defect site and rapidly crosslinked using 30 s of UV irradiation. The wound was then sutured, and postoperative care was provided. Rats were sacrificed at 2 or 4 weeks post-surgery to collect patellar tendon tissues for further evaluation.

### Preparation of GelMA and GelMA-*c*-rFGF7

Initially, a LAP (lithium phenyl-2,4,6-trimethylbenzoylphosphinate) initiator solution was prepared by dissolving 0.05 g of LAP powder in 20 mL of PBS buffer, followed by heating at 40–50 °C for 15 min to achieve a 0.25% (w/v) LAP solution. Next, 10 μg of lyophilized FGF7 powder (PeproTech) was dissolved in 1 mL of the 0.25% (w/v) LAP solution to prepare a 10 μg/mL FGF7 solution. This FGF7 solution was incubated with AC-PEG-NHS at 4 °C in the dark for 24 h. Meanwhile, 1 g of GelMA powder was dissolved in 10 mL of the 0.25% (w/v) LAP standard solution, heated in the dark for 20 min, and filtered to obtain the GelMA hydrogel precursor solution. Subsequently, 1 mL of the GelMA hydrogel precursor solution was mixed thoroughly with 1 mL of the PEGylated FGF7 solution and centrifuged to obtain the GelMA-*c*-rFGF7 hydrogel mixture. The preparation of GelMA involved dissolving 1 g of GelMA powder in 10 mL of the 0.25% (w/v) LAP solution, heating in the dark for 20 min, and then filtering. The GelMA precursor solution was combined with PEGylated rFGF7 to form a fluidic mixture, which, upon UV irradiation for 30 s, formed a stable GelMA-*c*-rFGF7 hydrogel (Fig. S6B). All reagents, including Phenyl-2,4,6-trimethylbenzoylphosphine oxide (LAP) photoinitiator and Acrylate-PEG-NHS (AC-PEG-NHS), were purchased from Engineering for Life Co., Ltd (Suzhou, China).

### Histological evaluation

Tissue samples were collected as required for the experiment and underwent fixation, dehydration, clearing, and embedding. Hematoxylin and Eosin (HE) staining involved deparaffinization, hydration, hematoxylin staining, differentiation, bluing, eosin staining, dehydration, clearing, and mounting. For Masson trichrome staining, the steps included deparaffinization, hydration, hematoxylin staining, acid fuchsin, phosphomolybdic acid, aniline blue, dehydration, and mounting. Each sample was evaluated, and scores recorded, with lower scores indicating general repair outcomes. The method is as follows: the region of interest was preferentially selected in the reparative tissue adjacent to the implanted scaffold or material, ensuring that the analytical fields were immediately adjoining but did not include the scaffold itself. For samples without a scaffold, areas were randomly chosen within the target regeneration zone. All tissue sections were evaluated under a light microscope at magnifications of ×200 or ×400. For each sample, two to three non-overlapping fields were randomly selected, mandatorily including one field from the interface zone between the reparative and native tissue (or material edge) and one field from the central region of the reparative tissue. An initial gross assessment was performed based on the overall morphological appearance, categorized into a four-tier scale: Normal (score 0), Mildly Abnormal (score 1), Moderately Abnormal (score 2), and Severely Abnormal (score 3). This was followed by a detailed semi-quantitative analysis based on six key histological parameters. A final comprehensive score, utilizing half-point increments (e.g., 1.5, 2.5), was derived by integrating the scores from all parameters. The specific criteria are as follows: a score of 0 represents the ideal condition, characterized by the presence of continuous, dense, and bundled collagen fibers forming prominent undulating waves, parallel fiber alignment with inter-bundle angles of less than 20°, nuclei that are almost exclusively elongated and spindle-shaped (>95%), an absence of inflammatory cell infiltration, no venules within the field, and sparse cellularity. A score of 1 indicates mild abnormality, defined by discernible and bundled fiber structure with reduced waviness, increased inter-bundle angles up to 45°, a majority of spindle-shaped nuclei (60–80%), inflammatory cells infiltrating less than 10% of the tissue area, the presence of no more than two venules, and moderate cellularity. A score of 2 signifies moderate abnormality, featuring disorganized fiber structure without clear bundling and scattered arrangement (inter-bundle angles between 45° and 90°), a mixed nuclear morphology with approximately half becoming round, inflammatory cells occupying 10–30% of the area, up to four venules, and numerous cells. A score of 3 denotes severe architectural disruption, marked by a lack of organized fiber structure with bundles oriented vertically or randomly, nuclei rarely spindle-shaped (<5%), severe inflammatory cell infiltration affecting over 30% of the area, the presence of five or more venules, and abundant, densely packed cells. Furthermore, the presence of empty areas resulting from tissue detachment or residual material was considered during the assessment. The proportional area of this was integrated into the final evaluation, primarily affecting the scoring for collagen fiber architecture and cellular density.

### Immunofluorescence assays

Tissue samples were first fixed in 4% paraformaldehyde and sectioned into 10 μm slices using a cryostat (Cryostat Microtome NX50). The sections underwent sequential permeabilization, blocking, primary and secondary antibody incubation, nuclear staining, and were examined under a confocal fluorescence microscope. For paraffin-embedded samples, deparaffinization, hydration, and antigen retrieval were performed before proceeding with the subsequent steps. For cell

samples, hTSPC or mTSPC were fixed, permeabilized, blocked, incubated with primary and secondary antibodies, stained for nuclei, and analyzed using a confocal fluorescence microscope. A list of antibodies is available in Supplementary Table S2.

## Real-time quantitative PCR (RT-qPCR)

RNA was extracted from Achilles tendon tissues of *Fgf7*$^{-/-}$ and wildtype mice using Trizol reagent (Thermo Fisher Scientific), and mRNA was reverse transcribed into cDNA using reverse transcriptase (ReverTra Ace® qPCR RT Master Mix). PCR was performed using gene-specific primers and SYBR Premix Ex Taq™ (TaKaRa) on the StepOnePlus Real-Time PCR System (Applied Biosystems, USA). For ex vivo explant samples, wild-type Achilles tendon tissues were cultured with rFGF7 (10 ng/mL) or standard medium for 7 days, followed by RNA extraction and qPCR. For cell samples, hTSPCs were cultured with rFGF7 (10 ng/mL) or standard medium for 2 days, followed by RNA extraction and qPCR. The primers used in qPCR are provided in the Supplementary Information file (Supplementary Table S1).

## Statistics and reproducibility

All data are presented as mean ± SD unless specifically mentioned, representing at least three independent experiments. Statistical analysis was performed using GraphPad Prism 8.0 software (GraphPad, La Jolla, USA). The statistical methods, significance levels, and *n* values are described in the figure legends. Student's *t* test was used for comparisons between two groups, while one-way ANOVA followed by Tukey's post hoc test was used for multiple group comparisons. Differences were considered statistically significant at $p < 0.05$. Significance levels are indicated as follows: *$p < 0.05$, **$p < 0.01$, ***$p < 0.001$.

## Reporting summary

Further information on research design is available in the Nature Portfolio Reporting Summary linked to this article.

## Data availability

The raw sequence data of Single-cell RNA sequencing and RNA-Seq reported in this paper have been deposited in the Genome Sequence Archive[43] in National Genomics Data Center[44], China National Center for Bioinformation/Beijing Institute of Genomics, Chinese Academy of Sciences with accessions CRA019874 and CRA019760. Source data are provided with this paper.

## Code availability

Data analyses were performed using R system software (version 4.0.5; https://www.r-project.org/), along with packages from the Bioconductor project, as well as original R code. This paper does not utilize any custom-written code.

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

## Acknowledgements

This work was supported by the National Key Research and Development Program of China (2022YFA1106800), NSFC grants (T2121004, 82222044, 32471211, 82302726), and Technology Project of Zhejiang Province (2024SSYS0026). We extend our gratitude to Professor Bin Zhou from the University of Chinese Academy of Sciences for providing the ProTracer mice. We also acknowledge Shuangshuang Liu and Wei Yi from the Core Facilities, Zhejiang University School of Medicine, for their technical support. Our thanks extend to the Analysis Center of Agrobiology and Environment Sciences of Zhejiang University for their technical assistance. We also thank the Center of Cryo-Electron Microscopy (CCEM), Zhejiang University for technical assistance with transmission electron microscopy.

## Author contributions

The study was conceived by R.L., H.Z., J.L., H.O., X.C., and Z.Y. The experiments were designed by R.L., Y.H., and T.H. Figures were prepared by J.L., T.H., R.L., and Z.W. Manuscript writing was carried out by R.L. and Z.Y. Animal experiments were assisted by Z.W., R.L., F.Y., S.L., and T.F. Clinical samples were collected by F.Y., J.L., W.S., and C.T. Bioinformatics analyses were performed by J.L., R.L., R.Y., and C.F. Molecular experiments were conducted by R.L. and J.L. Figure optimization was done by Y.Z. and X.C.

## Competing interests

The authors declare no competing interests.
