## [Transparent Peer Review file · Nature Communications]

FGF7 promotes load-bearing tendon regeneration and suppresses fibrosis

Corresponding Author: Professor Zi Yin

Version 0:

Reviewer comments:

Reviewer #1

(Remarks to the Author)

Thank you for the opportunity to review the manuscript entitled "Harnessing FGF7 for regeneration: from fibrosis suppression to functional healing." This manuscript explores the role of FGF7 in tissue fibrosis and functional regeneration, particularly in load-bearing connective tissues such as the Achilles and patellar tendons. The authors support their claims through single-cell RNA-sequencing, ProTracer lineage tracing, and the development of a GelMA hydrogel loaded with recombinant FGF7, suggesting that FGF7 plays a crucial role in preventing tendon fibrosis and promoting regeneration. The publication provides many novel and impactful findings; however, additional work should be completed prior to publication in Nature Communications. Please find the following major and minor comments below:

Major Comments:

1. Novelty of the research: The study offers a fresh perspective on the role of FGF7 in tendon regeneration and fibrosis suppression, marking a significant contribution to the field of tendon repair. However, the roles of other FGF family proteins in tendon regeneration have been reported (PMID: 15738660, 39731092). A more in-depth comparison with prior studies is necessary to better highlight the novelty of these findings.
2. Motor Function Assessment: The authors could clarify how the observed decrease in motor ability in *Fgf7*^{-/-} mice can be specifically attributed to tendon fibrosis rather than other potential factors such as muscle atrophy or neurological issues. Additional assessments, such as muscle force measurement or histological analysis of muscle tissue, could help rule out these alternative explanations.
3. Rationalization of the selection of fibrosis markers: The authors could address the potential overlap in extracellular matrix (ECM) components secreted by tenocytes and fibrotic cells, such as type I collagen. When selecting markers to reflect fibrosis and tendon differentiation, consideration should be given to markers that can more clearly distinguish between these two processes.
4. Specificity of AZD4547: The authors should address how the use of AZD4547, a broad-spectrum FGFR inhibitor, might affect other FGF family members. Additional experiments or discussions on the specificity of AZD4547 for FGFRs in the context of tendon fibrosis would strengthen the study. To better understand the specificity of the FGFR inhibitor AZD4547, the authors could include groups treated with AZD4547+rFGF7 and AZD4547+other FGF family proteins in Fig4. This would help determine if the effects of AZD4547 can be rescued by exogenous FGF7 rather than other FGF family proteins.
5. Rescue Experiment: The authors could include more rescue experiments. For example, in FigS3, FGF7 could be reintroduced into FGF7-deficient mice. This would help confirm that the observed effects are specifically due to the absence of FGF7 and can be reversed by its reintroduction.
6. Selection of Lineage Tracing Marker: The authors may justify their choice of Ki67-CreER for lineage tracing of tendon progenitor cells. Alternative markers such as Tppp3-CreER or Prg4-CreER, which are more specific to tendon progenitor cells, could provide more accurate tracing of their differentiation into fibrotic cells.
7. Mechanistic Insights: While the study identifies the *Lox+Uts2r+* fibrotic subpopulation and the role of the TGF- β pathway, the authors could conduct additional experiments to further explore the specific molecular mechanisms by which FGF7 regulates tendon stem cell differentiation and inhibits fibrosis.
8. Suggested supplementary experiments: 1) While the study identifies fibrotic phenotypes in FGF7-deficient tendons, the authors could strengthen their findings by including immunohistochemistry (IHC) staining for key fibrosis markers (e.g., α -SMA, collagen type III) in Fig2; 2) To further confirm the fibrotic phenotype in FGF7-deficient tendons, the authors could consider using ACTA2-GFP reporter mice in Fig3. This would provide a more direct visualization of ACTA2 expression, a key marker of fibrotic cells;
9. Therapeutic Potential Assessment: The authors should discuss the potential side effects or risks associated with the use

of recombinant FGF7 and the GelMA hydrogel system. This would provide a more comprehensive assessment of its therapeutic potential and address potential concerns for clinical translation.

10. Discussion Rigor: 1) The authors generally provide reasonable interpretations of their results. However, the claim that FGF7 deficiency leads to fibrosis should be tempered with a discussion of the potential contributions of other factors in the complex process of tendon healing. Other variables like mechanical stress and inflammatory responses should be considered more thoroughly; 2) The discussion of the study's limitations is somewhat brief. A more comprehensive analysis of limitations, such as the generalizability of findings from animal models to humans and potential long-term effects of the GelMA hydrogel system, should be included to enhance the rigor of the discussion; 3) The authors should more thoroughly compare their results with those of other studies. For example, discussing how their findings align with or differ from previous research on FGF family members in tendon biology could provide a more complete picture of the field.

Minor Comments:

1. Sirius Red Staining in Fig4E and FigS3E: The authors mention that Sirius Red staining can detect type III collagen, which is associated with fibrosis. However, the absence of green fluorescence in the relevant figures (e.g., Fig4E and FigS3E) needs to be addressed. The authors should explain why this expected staining pattern is not observed or provide additional staining protocols that can better visualize type III collagen.

2. Editorial Revisions: The manuscript should be carefully edited for spelling and grammatical errors to enhance readability and clarity.

3. Nomenclature Consistency: The authors could consider using consistent terminology throughout the manuscript, such as distinguishing between "tendon stem cells" and "tendon progenitor cells," to avoid confusion.

4. Image Guidance: In FigS3 B-C, the authors could provide lower-magnification images to include more cells in the field of view. This would help readers better assess the overall distribution and number of cells in the analyzed sections.

Reviewer #2

(Remarks to the Author)

This manuscript from Lin et al., is focused on the role of FGF7 in soft tissue fibrosis and regeneration using the tendon as a model to understand fundamental mechanisms of these processes. The authors are commended on a substantial body of work and the data on the effects of FGF7 on the tendon healing process are very compelling. However, there are substantial concerns about the rigor of several aspects of the experimental design, conclusions that are not well-supported by the data, and potential alternative interpretations of the data that are not discussed, substantially reducing the impact of these studies. Specific comments are below.

Major comments

1. Rigor of the experimental design. There are several major concerns here: The isolation of 'tendon stem cells' from ACL is not appropriate- tendons and ligaments are compositionally different, also these were isolated from ACL reconstructions but text says patients were healthy- any cells taken from the remnant are by definition not healthy. In addition, the interchangeable use of different anatomic tendon sites is not appropriate. Different tendons are functionally and compositionally distinct. As such, use of healthy hamstring tendon cells cannot be used to demonstrate that shoulder tendon pathology is enriched for the fibrotic population as this may be due to differences in composition between these tendons rather than pathology. In addition, it is not clear how the phenotype of Fgf^{-/-} mice, which is developmentally induced (even though the phenotype does not present until ~P21) is the same as mature pathology from shoulder arthroscopy with poorly defined pathology.

2. Related to the above, it does not appear that proper littermate controls as are used for Fgf7^{-/-} as the methods note that wildtype mice were purchased from GEM. This substantially detracts from the rigor of the sequencing data as slight differences in genetic background can impact expression profiles.

3. It is not clear that mouse and rat TSPCS were actually isolated. The isolation protocol would isolate ~all cells in the tendon and all tendon fibroblast populations are adherent- there is no selection for any cells with 'stem' or 'progenitor' properties.

4. The clinical relevance of the biopsy punch model in the patellar tendon is unclear

5. TEM cannot be used to assess mechanics

6. The data do not support a fibrotic phenotype in Fgf7^{-/-}. As the authors outline, fibrosis is characterized by excess and disorganized collagen formation. Fgf7^{-/-} mice have a substantial decline in collagen production/deposition (shown particularly strikingly on picro/PLM). Moreover, there seems to be a decline in tenogenic capacity, although whether this is more of a response to improper collagen patterning is unknown. The overall cell density between WT and Fgf7^{-/-} does not appear to be drastically different. There is a clear increase in Col 3, but the increase in aSMA does not seem to be related to myofibroblasts (e.g. there are not obvious stress fibers and this seems to be matrix staining). The authors do somewhat address this but another explanation is that Fgf7^{-/-} deficiency impairs Col1 production and or maturation and the increase Lox⁺ cells is an attempt to address this deficit by restoring the needed cross-linking.

7. 'Fgf7 primarily expressed in tendon fascia tissue (indicated by OCN protein)'- it is not clear what the authors mean by this. They suggest it is a niche for stem cells but there is broad expression of OCN in the tendon body that is not restricted to the paratenon/epitenon, as such there is not strong evidence for regulation of stemness via this mechanism.

8. In vitro data- the authors conclude that Fgf7 promotes tenogenic fate, however does Fgf7 actually just maintain the baseline tendon cell phenotype better than is typically observed in vitro? This is still an important finding the basis for this interpretation is not clear.

9. The rationale for the inclusion of enthesis cells and MTJ cells in the subsequent tenocyte sub-clustering is unclear. The unique presence of Utsr2/Lox population is clear but it would be helpful to know where these genes are expressed prior to subclustering the tendon cells. The human data does not provide strong support for these findings (e.g., the lack of this

population in the healthy tissue or enrichment in the diseased tissue as neither is an appropriate match (tendon or pathology) for the mouse data. In addition, the bulkseq data is much more consistent within age groups than genotypes but this is not discussed.

10. The conclusion that the Lox/Ustr subpopulation is 'a critical driver of the fibrotic phenotype' is not supported by the data. The population is certainly enriched but this is not causal evidence for the phenotype of Fgf7^{-/-} mice.

11. "Given the continuous cellular turnover in tendons" this is not supported by the current literature- Galloway has demonstrated limited turnover (~5% of cells) in tendons at P21.

Minor comments

1. Much of the introduction on other FGFs is not relevant to the central premise of the study-is more known about Fgf7 and collagen?
2. It is not clear why some data are presented +/- SEM and others have +/- SD.
3. Figure 8- there is no apparent EGR1 staining in the images provided despite the quantification suggesting a difference between groups.
4. The presentation of much of Figure 9 is confusing- the y-axis of several figures suggests that a decrease in stride length and width are 'better' which is not correct.

Reviewer #3

(Remarks to the Author)

Tendon injuries are common and can severely impair patient mobility and productivity, resulting in a significant socioeconomic burden and reduced quality of life. The tendon healing process results in the formation of a fibrotic scar, and as a consequence, injured tendons never regain the mechanical strength of their uninjured counterparts, leading to frequent re-injury. In this study, the authors demonstrated that fibroblast growth factor 7 (FGF7) promotes the proliferation and tenogenic differentiation of tendon stem cells while inhibiting fibrotic progression. Furthermore, a GelMA hydrogel loaded with recombinant FGF7 (GelMA-c-rFGF7) significantly enhanced functional tendon regeneration in vivo. These findings highlight the pivotal role of FGF7 in preventing tendon fibrosis and promoting regeneration, offering promising therapeutic strategies for tendon diseases. The following issues need to be addressed in the revised manuscript:

Major points :

1. The study utilized FGF7 global knockout (Fgf7^{-/-}) mice, which exhibited significant abnormalities in the development of load-bearing tendons. These findings highlight the essential role of FGF7 in regulating tenogenic differentiation. However, the cellular and molecular mechanisms by which FGF7 prevents tendon fibrosis and promotes regeneration remain unclear. The global knockout of FGF7 lacks spatial and temporal specificity, and most experimental data were obtained under physical conditions that do not accurately reflect tendon repair and regeneration, particularly in adult mice. Therefore, it would be beneficial to provide in vivo evidence that clarifies the fate-determinant effects of FGF7 on tendon stem/progenitor cells (TSPCs) and investigates the underlying molecular mechanisms during tendon repair or regeneration, especially under conditions where FGF7 signaling is specifically modified in tendons or TSPCs in future studies.
2. The newly identified Lox+Uts2r+ fibrotic subcluster appears to play a key role in fibrosis during tendon healing, and this should be confirmed in wild-type (WT) mice during tendon repair and regeneration. Furthermore, it is essential to investigate how the cellular composition, particularly the Lox+Uts2r+ fibrotic subcluster, changes during fibrotic tendon repair and FGF7-induced tendon regeneration using single-cell RNA sequencing. Additionally, the cellular origin of the Lox+Uts2r+ fibrotic subcluster warrants further exploration.
3. This study aimed to clarify the effects and mechanisms of FGF7 in preventing tendon fibrosis and promoting regeneration, with a particular focus on the fibroblastic/proliferative phase of tendon healing. However, its anti-fibrotic effects may not extend to fibrosis in other organs or tissues. For instance, intra-articular injection of recombinant FGF7 (rFGF7) has been shown to promote joint inflammation and articular cartilage degradation. Instead of referencing the general process of fibrosis, it is recommended to concentrate on the biological processes of tendon healing and regeneration in the introduction. Additionally, the title should be modified accordingly.

Minor points :

1. The HE staining of paired or adjacent sections of the fluorescence images in Figure 6G should be provided.
2. In Figures 9 and 10, why was the full-thickness window defect model used in the study? Additionally, a 4-week period post-surgery is insufficient for the injured tendon to fully repair and regenerate. As shown in Figures 10A and B, the tendons were far from normal histologically. In this context, why were the time points of 2 and 4 weeks chosen for evaluation?
3. In Figure 10, the cellular composition, particularly the Lox+Uts2r+ fibrotic subcluster, during FGF7-induced tendon regeneration should be provided.

Reviewer #4

(Remarks to the Author)

In the manuscript titled Harnessing FGF7 for regeneration: from fibrosis suppression to functional healing, the authors revealed the role of FGF7 in promoting regeneration and inhibiting fibrosis in mechanically loaded tendons, which was supported by the motor function, structural analysis and experiments in vitro. Single-cell RNA-sequencing was used to investigate the mechanism. A GelMA hydrogel loaded with rFGF7 was manufactured to enhanced functional tendon regeneration in vivo. It is a well-organized study with instructive meaning. There are some issues that the authors need to pay attention to:

1. Unlike fibrosis in solid organs such as the liver or lung, where excessive ECM deposition replaces functional parenchymal tissue, tendon fibrosis represents a maladaptive remodeling process within an already ECM-rich connective tissue. Tendon fibrosis is characterized by a disruption in the composition, spatial organization, and functional integrity of the collagenous matrix. Therefore, it is essential to provide a clear definition of tendon fibrosis in the Introduction, taking into account the unique structural and functional characteristics of tendon tissue. Distinguishing tendon fibrosis from fibrosis in other organs helps to clarify its specific pathological mechanisms and underscores the importance of context-specific therapeutic strategies.
2. ACTA2 was used as a marker of fibrosis in this study, but it is also a marker of myofibroblasts or vascular smooth muscle cells, and even as a TSPC marker in some tendon-related studies. So please provide a reasonable basis for ACTA2 as a marker of tendon fibrosis.
3. The trends of Figure 2C and 2E are contrary to the description in Result, which needs to be checked.
4. Line 138th: Additionally, we found that the deficiency and fibrosis phenotype at post-P21 and 10 weeks was not presented in the positional tail tendons.
Line 142th: At P21, wildtype mice showed increased mature collagen fibers, whereas FGF7^{-/-} mice displayed fewer and discontinuous collagen fibers same as fibrosis process.
What fibrotic phenotype or fibrosis process are you referring to and what is the supporting evidence for this statement?
5. To support the in vitro findings, a concentration gradient experiment was supposed to perform to identify the most effective dose of recombinant FGF7 protein.
6. Line 243th: In summary, we confirmed that FGF7 inhibits fibrosis by promoting the proliferation and tenogenic differentiation of both mouse and human tendon stem/progenitor cells.
The conclusion is supposed to modify exactly; because the results in Figure 4 do not reveal the relation between the promoting role of proliferation and tenogenic differentiation and the inhibition role of fibrosis.
7. Additional evidence is needed to prove the conclusion that the persistence and fibrosis-promoting activity of the Lox⁺Uts2r⁺ fibrotic subcluster in FGF7KO tendons are maintained through the regulation of the Creb3l1 transcription factor by TGF- β signaling.
8. Literature support or data is needed to demonstrate that isolated tendon tissue from SCX-GFP mice, when removed from its in vivo environment, develops fibrosis.
9. More material characterization experiments of GelMA-c-rFGF7 hydrogel, including SEM, need to be supplemented.
10. In the Materials and Methods section, please check whether CellChat or CellpjdDB is used to reveal the cell-cell interaction in the single-cell RNA-sequencing analysis.
11. Is it possible to construct tendon or Achilles tendon specific FGF7KO mice?
12. Some English grammar mistakes need to be corrected.

Version 1:

Reviewer comments:

Reviewer #1

(Remarks to the Author)

The authors have made appreciable efforts in addressing previous comments, and the manuscript has been improved accordingly. However, a number of essential points still require further attention to solidify the central claims of this study. The following suggestions are offered to help enhance the clarity and robustness of the work:

1. In Lines 303–313, the authors describe single-cell RNA sequencing performed on tendon tissue from FGF7 KO mice. It would be helpful to provide references or rationale supporting the criteria used for cell clustering to enhance the methodological transparency.
2. A putative fibrotic LOX⁺UTS2R⁺ subpopulation was identified in FGF7 KO mice. However, as indicated in Fig. 5E, both LOX and UTS2R are also highly expressed in certain fibroblast subpopulations. Therefore, relying solely on these two markers may not be sufficient to distinguish a unique fibrosis-driving cluster. It would be important to clarify whether this subpopulation also expresses other markers or fibrosis-related genes at elevated levels.
3. Throughout the manuscript, the interpretation of key data—particularly regarding tendon tissue morphology and marker localization—would be strengthened by including wider-field views for both H&E staining and fluorescence images (e.g., Fig. 5F, 6G, 10E). Overview images would help readers better assess the anatomical context and cellular distribution of the signals presented.
4. In Lines 360–363, the observation of GFP⁺LOX⁺UTS2R⁺ cells at a single time point is interesting but may not fully support the proposed differentiation trajectory. Multi-time-point observations would help more definitively establish whether GFP⁺LOX⁻UTS2R⁻ cells can differentiate into GFP⁺LOX⁺UTS2R⁺ cells over time.
5. In Figures R4A and R5A, the images shown for the control and AZD4547-treated groups appear very similar. Moreover, they resemble images previously presented in Figure 4G. It would be important to confirm that these experimental and control conditions were conducted under comparable and simultaneous conditions. In addition, inclusion of statistical analysis for these figures would strengthen the interpretation of the results.

6. Regarding Figure R6, the in vitro data are valuable but may not be sufficient to demonstrate that FGF7 readministration attenuates fibrosis or promotes tendon regeneration in FGF7 KO mice. If possible, in vivo rescue experiments in FGF7 KO mice would provide more direct evidence. If such experiments are not feasible, a brief justification would be helpful.
7. The pseudotime analysis suggesting that CD44⁺Tppp3⁺ TSPCs may give rise to LOX⁺UTS2R⁺ cells is intriguing. However, given that proliferating cells often differentiate into fibroblasts, additional in vitro or in vivo validation would strengthen the conclusion that CD44⁺Tppp3⁺ TSPCs can indeed differentiate into LOX⁺UTS2R⁺ cells.

Reviewer #2

(Remarks to the Author)

The revised manuscript has been partially responsive to the prior critique. However, there are two main issues that have not been resolved. The primary issue relates to the authors conclusion that the developmental phenotype in Fgf7^{-/-} tendons is fibrotic. While the response to the prior comment does provide a more nuanced explanation for the rationale for calling this a fibrotic phenotype the data do not support this. For example, the new data demonstrating decreased diameters across Fgf7^{-/-} tendon suggest this not a true fibrotic state (e.g., excess and disorganized matrix). Similar comment for the picrosirius/ PLM data, decreased Col1 expression, and the lack of aSMA⁺ stress fiber + cells. There is clearly a collagen assembly, cross-linking etc phenotype but not fibrosis.

Second, the authors use the conclusion of fibrosis to justify the identification of the Uts2r⁺ Lox⁺ cells as a driver of fibrosis. However, there is no direct evidence to demonstrate a causal role for these cells in fibrosis. The evidence of greater Tgfb signaling within/to/from this population is not very compelling, given the strong Tgfb signaling signature in other populations (e.g., the Prg4⁺ population). Moreover, changes in the presence of this population were not assessed in the GelMA experiments. It would be helpful to use publicly available mouse scSeq data to demonstrate evidence of this population in either fibrotic tendon healing or tendon degeneration.

There seem to be far more Utsr2⁺ Lox⁺ cells in 5B, vs D, presumably because some are in the 21 day time point? It is important to clarify whether the lack of this population is maturation state dependent.

Several pieces of data including the in response to reviewers should be included in the manuscript or supplementary data:

1. Which populations from 5A do the Uts2r⁺ Lox⁺ cells map to? Figure R11B
2. Fig R12
3. Fig R9A. Related, in Figure 6 F -H the authors should be very clear about the tendons that these samples were isolated from.

The nomenclature in Figure 7 is very confusing. Are the WT mice sham controls with no injury? The difference between this group and the 'control' is not immediately clear from the text.

Minor:

1. Intro- the sentence on lines 80-81 have no relevance, and lack clarity, thus should be removed.
2. The introduction is excessively long and unfocused

Reviewer #3

(Remarks to the Author)

The authors have made comprehensive revisions in accordance with our comments, and we have no further concerns.

Reviewer #4

(Remarks to the Author)

I believe the authors have addressed the issues. The revised manuscript could be published in this journal now.

Version 2:

Reviewer comments:

Reviewer #1

(Remarks to the Author)

The authors have addressed the comments appropriately.

Reviewer #2

(Remarks to the Author)

The authors have sufficiently addressed the prior comments.

**REVIEWER COMMENTS**

Reviewer #1 (Remarks to the Author):

Thank you for the opportunity to review the manuscript entitled “Harnessing FGF7 for
regeneration: from fibrosis suppression to functional healing.” This manuscript
explores the role of FGF7 in tissue fibrosis and functional regeneration, particularly in
load-bearing connective tissues such as the Achilles and patellar tendons. The authors
support their claims through single-cell RNA-sequencing, ProTracer lineage tracing,
and the development of a GelMA hydrogel loaded with recombinant FGF7, suggesting
that FGF7 plays a crucial role in preventing tendon fibrosis and promoting regeneration.
The publication provides many novel and impactful findings; however, additional work
should be completed prior to publication in Nature Communications. Please find the
following major and minor comments below:

We sincerely appreciate the reviewer for the encouraging feedback and recognition
of the novelty and impactful value of our study. Below, we have provided a point-by-
point response for your review. Your comments are presented in black text, while our
responses are in blue text, with any revisions or additions to the manuscript highlighted
in bold blue. In the revised manuscript, all modifications and additions are displayed in
blue font. Thank you once again for your valuable feedback.

Major Comments:

1. Novelty of the research: The study offers a fresh perspective on the role of FGF7 in
tendon regeneration and fibrosis suppression, marking a significant contribution to the
field of tendon repair. However, the roles of other FGF family proteins in tendon
regeneration have been reported (PMID: 15738660, 39731092). A more in-depth
comparison with prior studies is necessary to better highlight the novelty of these
findings.

**Response:**

Thank you very much for your insightful suggestions. We have compared our
findings with studies on other FGF family proteins (PMID: 15738660, 39731092) and
have reiterated the innovative aspects of our research.

The study PMID: 15738660[1] showed that low doses of FGF-2 promote BMSC
differentiation towards a fibroblast-like phenotype, increasing collagen type I and
fibronectin mRNA expression. However, it only assessed extracellular matrix (ECM)
components and did not assess the tenogenic effect, which prevents a distinction
between tendonogenesis and fibrosis.

The study PMID: 39731092[2] found that aligned and random PCL fiber scaffolds
with bFGF nanoparticles enhance tendon-associated gene and protein expression.
However, this study primarily focused on the scaffold design and did not elucidate the
impact of FGF2 on tendon tissue.

We have conducted a more in-depth comparison with previous studies about other
FGF family proteins to further emphasize the novelty of our research findings. we have
made the following modifications to the discussion section, highlighted in blue in the
revised manuscript (Page 23, Line 496-523):

**“The FGF family represents a crucial pathway in tendon development [3, 4].**
**However, it remains unclear which specific FGF subtype plays a significant role**
**in tenogenesis. Some studies have reported that FGF4 effectively regulates the**
**expression of tendon-related proteins, such as TNMD and THBS2, during chick**
**limb development [5, 6]. However, its role in mouse limb tendon development**
**has been shown to be ineffective and suppresses the expression of SCX in**
**mouse tendon stem cells [7]. FGF8 has been demonstrated to play an important**
**role in tendon development in chickens [8], yet its expression has not been**
**detected during mouse tendon development [5]. Additionally, studies have**
**indicated that activated FGF8 inhibits the expression of Scx, Tnmd, and TnC in**
**the tendon tissue of the mandible [9]. FGF2 has been shown to enhance the**

production of extracellular matrix in injured tendon tissues [10]. However, the
tenogenic effect of FGF2 on stem cells remain controversial[11-13]. The specific
role of different FGF subtypes in tendon development is still not fully understood.
In this study, we have identified that FGF7 plays an important role in promoting
normal tendon development by utilizing FGF7 knockout (FGF7KO) mice.

In vivo experiments, our findings highlight the diminished motor function and
identified a novel *Lox⁺Uts2r⁺* fibrotic subpopulation in *Fgf7* knockout mice,
underscoring the necessity of FGF7 for maintaining tendon integrity and
inhibiting fibrosis. In vitro experiments demonstrated that FGF7 can promote the
tenogenic differentiation of tendon stem cells while inhibiting their fibrotic
trajectory. Additionally, we also conducted tendon explant cultures to confirm
the phenotype maintenance and anti-fibrotic effects of FGF7. Finally, we
employed the GelMA-c-rFGF7 system to deliver FGF7 to damaged tendon sites,
resulting in the functional regeneration of injured tendons. Overall, our findings
provide valuable insights into FGF specific subtype FGF7 promotes tendon
commitment of TSPCs and regulate tendon differentiation during development
and pathology. This presents a novel potential strategy for the treatment of
tendon injuries in the future.”

**References:**

- [1] Hankemeier S, Keus M, Zeichen J, et al. Modulation of proliferation and differentiation of human
bone marrow stromal cells by fibroblast growth factor 2: potential implications for tissue engineering of
tendons and ligaments. *Tissue Eng.* 2005. 11(1-2): 41-9.
- [2] Li Y, Ge Z, Liu Z, et al. Integrating electrospun aligned fiber scaffolds with bovine serum albumin-
basic fibroblast growth factor nanoparticles to promote tendon regeneration. *J Nanobiotechnology.* 2024.
22(1): 799.
- [3] Schweitzer R, Zelzer E, Volk T. Connecting muscles to tendons: tendons and musculoskeletal
development in flies and vertebrates. *Development.* 2010. 137(17): 2807-17.
- [4] Brent AE, Schweitzer R, Tabin CJ. A somitic compartment of tendon progenitors. *Cell.* 2003. 113(2):
235-48.
- [5] Edom-Vovard F, Schuler B, Bonnin MA, Teillet MA, Duprez D. *Fgf4* positively regulates scleraxis and
tenascin expression in chick limb tendons. *Dev Biol.* 2002. 247(2): 351-66.
- [6] Havis E, Bonnin MA, Esteves de Lima J, Charvet B, Milet C, Duprez D. TGF β and FGF promote

tendon progenitor fate and act downstream of muscle contraction to regulate tendon differentiation during
chick limb development. *Development*. 2016. 143(20): 3839-3851.

[7] Brown JP, Finley VG, Kuo CK. Embryonic mechanical and soluble cues regulate tendon progenitor
cell gene expression as a function of developmental stage and anatomical origin. *J Biomech*. 2014. 47(1):
214-22.

[8] Eloy-Trinquet S, Wang H, Edom-Vovard F, Duprez D. Fgf signaling components are associated with
muscles and tendons during limb development. *Dev Dyn*. 2009. 238(5): 1195-206.

[9] Liu C, Zhou N, Li N, et al. Disrupted tenogenesis in masseter as a potential cause of micrognathia.
*Int J Oral Sci*. 2022. 14(1): 50.

[10] Tokunaga T, Shukunami C, Okamoto N, et al. FGF-2 Stimulates the Growth of Tenogenic Progenitor
Cells to Facilitate the Generation of Tenomodulin-Positive Tenocytes in a Rat Rotator Cuff Healing Model.
*Am J Sports Med*. 2015. 43(10): 2411-22.

[11] Havis E, Bonnin MA, Olivera-Martinez I, et al. Transcriptomic analysis of mouse limb tendon cells
during development. *Development*. 2014. 141(19): 3683-96.

[12] Li K, Zhang X, Wang D, Tuan RS, Ker D. Synergistic effects of growth factor-based serum-free
medium and tendon-like substrate topography on tenogenesis of mesenchymal stem cells. *Biomater Adv*.
2023. 146: 213316.

[13] Li B, Jha RK, Qi YJ, et al. Early cellular responses of BMSCs genetically modified with bFGF/BMP2
co-cultured with ligament fibroblasts in a three-dimensional model in vitro. *Int J Mol Med*. 2016. 38(5):
1578-1586.

2. Motor Function Assessment: The authors could clarify how the observed decrease
in motor ability in *Fgf7*^{-/-} mice can be specifically attributed to tendon fibrosis rather
than other potential factors such as muscle atrophy or neurological issues. Additional
assessments, such as muscle force measurement or histological analysis of muscle
tissue, could help rule out these alternative explanations.

**Response:**

Thank you very much for your insightful suggestions. Based on your suggestions,
we performed hematoxylin-eosin staining on muscle tissues from both *Fgf7*^{-/-} mice and
wild-type mice. The muscle diameters of *Fgf7*^{-/-} mice did not show significant
differences compared to wild-type muscle tissue (Fig. R1). Furthermore, skeletal
muscle comprises approximately 40% of adult body weight[1]. The lack of substantial
changes in body weight indicates that motor dysfunction is not attributed to significant

alterations in muscle tissues (Fig. 1B). Finally, we have observed alterations in tendon
 malformation in *Fgf7*^{-/-} mice at both micro and nano levels (Fig. 2B, D, H-I, J-K). These
 analyses strengthen our assertion that the decline in motor function is associated with
 changes in tendon fibrosis rather than other potential factors.

**Fig. R1A, B.** H&E staining of TA (tibialis anterior) muscles collected from the wildtype
 and *Fgf7*^{-/-} mice at the age of 10 weeks. Scale bar: 20 μ m. Distribution of fiber size is
 shown. n = 4 mice. Error bars represent SDs; NS: not significant.)

**Fig. 1B.** Statistical analysis of body weight (right) at postnatal days 7, 14, 21, and 10
 133 weeks for *Fgf7*^{-/-} and wildtype mice (*n*= 8 per time point. Error bars represent SDs; NS:
 not significant, two-tailed Student's *t* test).

**Fig. 2B, D, H-I, J-K.** **B.** Hematoxylin and eosin (HE) staining of the patellar tendon at
 postnatal days 14, 21, and 10 weeks in *Fgf7*^{-/-} and wild-type mice. Scale bars, 20 μ m.
 **D.** HE staining of the Achilles tendon at postnatal days 14, 21, and 10 weeks in *Fgf7*^{-/-}
 and wildtype mice. Scale bars, 20 μ m. **H-I.** Transmission electron microscopy (TEM)
 images (H) showing transverse and longitudinal sections of collagen fibers in the
 patellar tendon at 10 weeks in *Fgf7*^{-/-} and wildtype mice (*n*=3), scale bars: 200nm, and
 their respective collagen diameter distribution (I). Scale bars, 0.2 μ m. **J-K.** TEM images
 (J) showing transverse and longitudinal sections of collagen fibers in the Achilles tendon
 at 10 weeks in *Fgf7*^{-/-} and wildtype mice (*n*=3), scale bars: 200nm, and their respective
 collagen diameter distribution (K). Scale bars, 0.2 μ m.

**References:**

[1] Yin L, Li N, Jia W, et al. Skeletal muscle atrophy: From mechanisms to treatments. *Pharmacol Res.* 2021. 172:
 105807.

3. Rationalization of the selection of fibrosis markers: The authors could address the
potential overlap in extracellular matrix (ECM) components secreted by tenocytes
and fibrotic cells, such as type I collagen. When selecting markers to reflect fibrosis
and tendon differentiation, consideration should be given to markers that can more
clearly distinguish between these two processes.

**Response:**

Thank you very much for your valuable suggestions. We have conducted additional
sequencing data from both pathological and normal human tendon tissues to
demonstrate the validity of our selected markers for reflecting fibrosis and tendon
differentiation.

Based on our human transcriptomic data of pathological($n=64$) and normal
tendons($n=11$), we further validated the specificity of the selected markers: α -SMA
(ACTA2) and type III collagen (COL3A1) for fibrosis, TNMD and MKX for tendon
differentiation. (Fig. R2A-D). The markers we employed for tendon differentiation and
maturation, such as TNMD and MKX, have been reported by other research groups in
relevant literature and are widely recognized as tendon differentiation markers[1, 2].
MKX mRNA has been shown to be strongly expressed in differentiating tendon cells
during embryogenesis and in tendon sheath cells in postnatal stages, highlighting Mxk
as a significant regulator of tendon development[3]. Scx directly activates Tnmd,
thereby positively regulating the differentiation and maturation of tendon cells.
Scleraxis has been identified as a transcriptional activator that regulates the
expression of Tenomodulin, a marker of mature tenocytes[4]. There are also reports
suggesting that Tnmd serves as a marker for tendon formation and that Scx positively
regulates Tnmd expression in a tendon cell lineage-dependent manner[5].

In the context of tendon fibrosis, we selected α -SMA (ACTA2) and COL3 as fibrosis
markers. During the tendon injury repair process, ACTA2 is widely expressed in scar
tissue during the early stages of tendon damage[6]. It is also utilized as a marker for

tendon fibrosis [7]. In tendon tissue, a marked increase in COL3 often signifies that the
extracellular matrix (ECM) is undergoing significant alterations during the repair and
fibrotic remodeling phase, underscoring the substantial role of ECM changes
associated with COL3 in the fibrosis process[8]. We did not rely solely on a single
marker to assess tendon fibrosis or tendon differentiation. Instead, we utilized ACTA2
and type III collagen as indicators of fibrosis, while TNMD and THBS4 were employed
to evaluate tendon differentiation (Fig. 10 E).

**Fig. R2A-D.** The gene (*COL3*, *ACTA2*, *MKX*, *TNMD*) transcriptomic data of human
normal tendon samples($n=11$) and tendinopathy samples($n=64$), with corresponding
statistical analysis (each dot represents one human sample; * $P<0.05$, ** $P<0.01$,
*** $P<0.001$, two-tailed Student's t test).

**Fig. 10E.** Immunofluorescence staining for ACTA2, COL3, TNMD, THBS4 at the
 repair site at 4 weeks post-surgery for each group (Control, GelMA, GelMA-c-rFGF7).
 Scale bars, 100 µm.

**References:**

- [1] Ito Y, Toriuchi N, Yoshitaka T, et al. The Mohawk homeobox gene is a critical regulator of tendon
 differentiation. *Proc Natl Acad Sci U S A.* 2010. 107(23): 10538-42.
- [2] Dex S, Alberton P, Willkomm L, et al. Tenomodulin is Required for Tendon Endurance Running and Collagen
 I Fibril Adaptation to Mechanical Load. *EBioMedicine.* 2017. 20: 240-254.
- [3] Liu W, Watson SS, Lan Y, et al. The atypical homeodomain transcription factor Mohawk controls tendon
 morphogenesis. *Mol Cell Biol.* 2010. 30(20): 4797-807.
- [4] Shukunami C, Takimoto A, Nishizaki Y, et al. Scleraxis is a transcriptional activator that regulates the
 expression of Tenomodulin, a marker of mature tenocytes and ligamentocytes. *Sci Rep.* 2018. 8(1): 3155.
- [5] Delgado Caceres M, Angerpointner K, Galler M, et al. Tenomodulin knockout mice exhibit worse late healing
 outcomes with augmented trauma-induced heterotopic ossification of Achilles tendon. *Cell Death Dis.* 2021. 12(11):
 1049.
- [6] Dymnt NA, Hagiwara Y, Matthews BG, Li Y, Kalajzic I, Rowe DW. Lineage tracing of resident tendon
 progenitor cells during growth and natural healing. *PLoS One.* 2014. 9(4): e96113.
- [7] Morita W, Snelling S, Wheway K, et al. Comparison of Cellular Responses to TGF-β1 and BMP-2 Between
 Healthy and Torn Tendons. *Am J Sports Med.* 2021. 49(7): 1892-1903.
- [8] Millar NL, Gilchrist DS, Akbar M, et al. MicroRNA29a regulates IL-33-mediated tissue remodelling in tendon
 disease. *Nat Commun.* 2015. 6: 6774.

4. Specificity of AZD4547: The authors should address how the use of AZD4547, a
broad-spectrum FGFR inhibitor, might affect other FGF family members. Additional
experiments or discussions on the specificity of AZD4547 for FGFRs in the context of
tendon fibrosis would strengthen the study. To better understand the specificity of the
FGFR inhibitor AZD4547, the authors could include groups treated with
AZD4547+rFGF7 and AZD4547+other FGF family proteins in Fig4. This would help
determine if the effects of AZD4547 can be rescued by exogenous FGF7 rather other
FGF family proteins.

**Response:**

Thank you very much for your insightful suggestions. In response to your
recommendations, we have conducted additional experiments that include the
AZD4547 treatment group supplemented with FGF7, as well as groups treated with
AZD4547 supplemented with rFGF2, rFGF2 alone, AZD4547 supplemented with
rFGF4, and rFGF4 alone (Fig. R3; Fig. R4; Fig. R5). The AZD4547 treatment group
supplemented with FGF7 did not effectively restore the expression levels of COL1A1,
MKX, and EGR1, nor did it reduce the expression of ACTA2 and COL3. This indicates
that FGF7 could not rescue the effects of AZD4547 on tendon stem cells,
demonstrating that FGF7 plays a relevant role in tendon stem cells through receptors
inhibited by AZD4547.

**Fig. R3A.** Immunofluorescence staining of tendon-related proteins (COL1A1, MKX,
EGR1) and fibrotic proteins (ACTA2, COL3) in hTSPCs treated with rFGF7, PBS,
AZD4547, AZD4547 supplemented with rFGF7, $n=3$ independent experiments, Scale
bars, 100 μm .

**Fig. R4A.** Immunofluorescence staining of tendon-related proteins (COL1A1, MKX,
EGR1) and fibrotic proteins (ACTA2, COL3) in hTSPCs treated with rFGF4, PBS,
AZD4547, AZD4547 supplemented with rFGF4, $n=3$ independent experiments, Scale
bars, 100 μm .

**Fig. R5A.** Immunofluorescence staining of tendon-related proteins (COL1A1, MKX,
EGR1) and fibrotic proteins (ACTA2, COL3) in hTSPCs treated with rFGF2, PBS,
AZD4547, AZD4547 supplemented with rFGF2, $n=3$ independent experiments, Scale

bars, 100 μ m.

5. Rescue Experiment: The authors could include more rescue experiments. For
example, in FigS3, FGF7 could be reintroduced into FGF7-deficient mice. This would
help confirm that the observed effects are specifically due to the absence of FGF7
and can be reversed by its reintroduction.

**Response:**

Thank you very much for your professional suggestions. Based on your
recommendations, we have conducted rescue experiments. We introduced rFGF7 into
TSPCs from *Fgf7^{-/-}* mice for the rescue experiments, and found that the expression of
tendon-related proteins MKX, EGR1, and TNMD significantly increased compared to
TSPCs from *Fgf7^{-/-}* mice. In contrast, the expression of the fibrotic proteins ACTA2 and
COL3 decreased relative to the TSPCs from *Fgf7^{-/-}* mice (Fig. R6). These findings
confirm that the observed effects are specifically due to the absence of FGF7 and can
be reversed by its reintroduction.

**Fig. R6A-F.** Immunofluorescence staining of tendon-related proteins (COL1A1, MKX,
 EGR1, TNMD) and fibrotic proteins (ACTA2, COL3) in mTSPCs treated with rFGF7,
 PBS, mTSPCs from *Fgf7*^{-/-} mice and mTSPCs from *Fgf7*^{-/-} mice supplemented with r
 FGF7, *n*=5 independent experiments, Scale bars, 100 μm.

6. Selection of Lineage Tracing Marker: The authors may justify their choice of Ki67-
 CreER for lineage tracing of tendon progenitor cells. Alternative markers such as
 Tpp3-CreER or Prg4-CreER, which are more specific to tendon progenitor cells,
 could provide more accurate tracing of their differentiation into fibrotic cells.

**Response:**

Thank you very much for your helpful suggestion. We utilized the *Ki67-CreER* tracing
 system (ProTracer system) primarily to label newly generated cells within tendon

tissues. This approach allows us to gain a more comprehensive understanding of the
differentiation pathways of the continuously renewing cells in tendon tissue.
Consequently, we observed that after FGF7 knockout, a portion of these cells
differentiated into the fibrotic *Lox⁺Uts2r⁺* cell subpopulation.

The *ProTracer* system, developed by Professor Zhou Bin's group, enhances
traditional Cre systems through a *Cre-Dre* dual recombinase mechanism involving
three mouse lines: *DreER*, *Ki67-CreER*, and a *R26-GFP* reporter. This system allows
for continuous genetic recording of cell proliferation, distinguishing it from conventional
transient labeling methods. Tamoxifen-induced recombination activates the Ki67
promoter, leading to persistent tracking of Ki67+ cells throughout the animal's lifespan
and confirming their transition into the *Lox⁺Uts2r⁺* fibrotic subpopulation after *Fgf7*
loss[1].

We fully agree that *Tppp3* and *Prg4* are more suitable TSPC markers. However, after
reviewing the available resources, we found that neither the GemPharmatech Co., Ltd
nor the Shanghai Model Organisms Center, Inc currently offers ready-made *Tppp3-*
*DreER* or *Prg4-DreER*, as well as *Tppp3-CreER* or *Prg4-CreER* mouse systems.
Therefore, we can only explore these options in the future.

**References:**

[1] He L, Pu W, Liu X, et al. Proliferation tracing reveals regional hepatocyte generation in liver homeostasis
and repair. *Science*. 2021. 371(6532). PMID: 33632818

7. Mechanistic Insights: While the study identifies the *Lox⁺Uts2r⁺* fibrotic
subpopulation and the role of the TGF- β pathway, the authors could conduct additional
experiments to further explore the specific molecular mechanisms by which FGF7
regulates tendon stem cell differentiation and inhibits fibrosis.

**Response:**

Thank you very much for your in-depth consideration of our study. We revealed that
the specific molecular mechanism behind the presence of *Lox⁺Uts2r⁺* subpopulation

cells following FGF7 knockout is the activation of the *Creb3l1* transcription factor. This
 conclusion is supported by our single-cell sequencing data (Fig. 7E, F, G) and
 experiments involving the inhibition of *Creb3l1*.

Additionally, we implemented inhibition experiments targeting *Creb3l1*. By
 introducing *Creb3l1* siRNA into tendon stem cells derived from FGF7 knockout mice,
 we observed a reduction in *Lox⁺Uts2r⁺* double-positive cells. This was accompanied
 by an increase in the expression of the tendon-specific protein TNMD and a decrease
 in the fibrotic markers ACTA2 and COL3 (Fig. R7).

**Fig. 7E.** SCENIC analysis of single-cell data reveals results for the nine tendon
 subclusters, highlighting the involvement of *Creb3l1* genes in TGF- β signaling. **F.** A
 dot plot illustrates the expression levels of the *Creb3l1* transcription factor across the
 nine tendon subclusters. The red box indicates the highest expression levels found in
 the *Lox⁺Uts2r⁺* fibrotic subcluster. **G.** UMAP plots of the nine tendon subclusters
 demonstrate the concentrated and highest expression of the *Creb3l1* transcription factor
 specifically in the *Lox⁺Uts2r⁺* fibrotic subcluster.

**Fig. R7A.** Immunofluorescence staining of LOX and UTS2R in mTSPCs from *Fgf7*^{-/-}
 mice treated with Si-*Creb311* or Si-NC, Scale bars, 100 μm. **B.** Immunofluorescence
 staining of tendon-related proteins (TNMD) and fibrotic proteins (ACTA2, COL3) in
 mTSPCs from *Fgf7*^{-/-} mice treated with Si-*Creb311* or Si-NC, 100 μm.

8. Suggested supplementary experiments: While the study identifies fibrotic
 phenotypes in FGF7-deficient tendons, the authors could strengthen their findings by
 including immunohistochemistry (IHC) staining for key fibrosis markers (e.g., α-SMA,
 collagen type III) in Fig2. To further confirm the fibrotic phenotype in FGF7-deficient
 tendons, the authors could consider using ACTA2-GFP reporter mice in Fig3. This
 would provide a more direct visualization of ACTA2 expression, a key marker of fibrotic
 cells.

**Response:**

Thanks for your constructive suggestions. We validated the expression of relevant
 fibrotic markers through immunofluorescence experiments, as shown in Fig. 3E. The
 COL3 protein is diffusely distributed within the tendon substance of *Fgf7*^{-/-} mice, and
 its expression is significantly enhanced compared to the control group. Additionally, the

ACTA2 protein also shows a marked increase in expression in the tendon tissues of
*Fgf7^{-/-}* mice when compared to the control group.

The Acta2-GFP gene reporter mice would be extremely beneficial for our research.
We sincerely appreciate your suggestions regarding our future work. However, our
research group, in collaboration with GemPharmatech Co., Ltd and the Shanghai
Model Organisms Center, currently lacks access to Acta2-GFP reporter mice. This
limitation prevents us from directly demonstrating the impact of Fgf7 deficiency on
ACTA2 expression.

We have conducted relevant immunofluorescence staining (Fig. 3E) and western
blotting (Fig. 3C) for ACTA2. In the future, we plan to construct or collaborate on this
mouse model.

**Fig. 3E.** Immunofluorescence staining of COL3 and ACTA2 in wildtype and *Fgf7^{-/-}*
mice tendon tissues. Independent experiments $n=4$. Scale bars, 100 μm .

**Fig. 3C.** Western blot showing protein expression level of ACTA2 in tendon tissues of
wildtype and *Fgf7^{-/-}* mice. Independent experiments $n=3$.

9. Therapeutic Potential Assessment: The authors should discuss the potential side
effects or risks associated with the use of recombinant FGF7 and the GelMA hydrogel
system. This would provide a more comprehensive assessment of its therapeutic

potential and address potential concerns for clinical translation.

**Response:**

We sincerely appreciate the reviewer for raising this important and constructive point.
We recognize the inherent risks associated with both components of the system. The
GelMA Hydrogel System faces challenges such as batch consistency and
photopolymerization limitations, necessitating rigorous quality control and alternative
photoinitiators for clinical translation. Recombinant FGF7 (rFGF7), as a therapeutic
agent, is a recombinant protein that may present risks distinct from its natural form,
including immunogenicity and stability issues. Addressing these risks will require
thorough characterization, optimization of dosage and comprehensive preclinical
safety studies to ensure successful clinical application.

10. Discussion Rigor: 1) The authors generally provide reasonable interpretations of
their results. However, the claim that FGF7 deficiency leads to fibrosis should be
tempered with a discussion of the potential contributions of other factors in the complex
process of tendon healing. Other variables like mechanical stress and inflammatory
responses should be considered more thoroughly;

**Response:**

Thank you very much for your thorough review of our manuscript and for your
valuable suggestions. To provide a more comprehensive reflection of the complexities
involved in tendon fibrosis, we have made the following modifications to the discussion
section, highlighted in blue in the revised manuscript (Page 26, Line 568-575):

**“Fibrosis results from the interplay of various factors during tendon injury**
**repair, including cytokine networks, inflammatory responses, mechanical stress,**
**and extracellular matrix remodeling imbalances. Our study suggests that Fgf7**
**deficiency, particularly in load-bearing tendon tissues, is a significant promoting**
**factor. This indicates an interactive relationship between mechanical stress and**

the protective role of FGF7. Additionally, chronic inflammatory responses are
crucial in driving fibrosis, and FGF7 may indirectly influence this process by
modulating inflammation, an area worth exploring in future research.”

2) The discussion of the study's limitations is somewhat brief. A more comprehensive
analysis of limitations, such as the generalizability of findings from animal models to
humans and potential long-term effects of the GelMA hydrogel system, should be
included to enhance the rigor of the discussion;

**Response:**

We greatly value your observation regarding the insufficient discussion of the study's
limitations. we have made the following modifications to the discussion section,
highlighted in blue in the revised manuscript (Page 27, Line 594-601):

“A limitation of this research is the lack of validation in higher-level animal
models, such as larger animals or non-human primates. Due to the difficulty in
obtaining normal rotator cuff tissue samples from human subjects, we utilized
hamstring tendon tissue as a control. Variability between tendon tissues from
different anatomical sites is, of course, unavoidable. Additionally, assessing the
translational potential and safety of GelMA hydrogels will require longer follow-
up periods in future research. Finally, the safety and efficacy of GelMA-c-rFGF7
need to be evaluated in future studies at the large animal level.”

3) The authors should more thoroughly compare their results with those of other
studies. For example, discussing how their findings align with or differ from previous
research on FGF family members in tendon biology could provide a more complete
picture of the field.

**Response:**

We have conducted a more in-depth comparison with previous studies to further

emphasize the novelty of our research findings. we have made the following
modifications to the discussion section, highlighted in blue in the revised manuscript
(Page 27, Line 496-523):

“The FGF family represents a crucial pathway in tendon development [3, 4].
However, it remains unclear which specific FGF subtype plays a significant role
in tenogenesis. Some studies have reported that FGF4 effectively regulates the
expression of tendon-related proteins, such as TNMD and THBS2, during chick
limb development [5, 6]. However, its role in mouse limb tendon development
has been shown to be ineffective and suppresses the expression of SCX in
mouse tendon stem cells [7]. FGF8 has been demonstrated to play an important
role in tendon development in chickens [8], yet its expression has not been
detected during mouse tendon development [5]. Additionally, studies have
indicated that activated FGF8 inhibits the expression of Scx, Tnmd, and TnC in
the tendon tissue of the mandible [9]. FGF2 has been shown to enhance the
production of extracellular matrix in injured tendon tissues [10]. However, the
tenogenic effect of FGF2 on stem cells remain controversial[11-13]. The specific
role of different FGF subtypes in tendon development is still not fully understood.
In this study, we have identified that FGF7 plays an important role in promoting
normal tendon development by utilizing FGF7 knockout (FGF7KO) mice.

In vivo experiments, our findings highlight the diminished motor function and
identified a novel *Lox⁺Uts2⁺* fibrotic subpopulation in *Fgf7* knockout mice,
underscoring the necessity of FGF7 for maintaining tendon integrity and
inhibiting fibrosis. In vitro experiments demonstrated that FGF7 can promote the
tenogenic differentiation of tendon stem cells while inhibiting their fibrotic
trajectory. Additionally, we also conducted tendon explant cultures to confirm
the phenotype maintenance and anti-fibrotic effects of FGF7. Finally, we
employed the GelMA-c-rFGF7 system to deliver FGF7 to damaged tendon sites,
resulting in the functional regeneration of injured tendons. Overall, our findings
provide valuable insights into FGF specific subtype FGF7 promotes tendon

**commitment of TSPCs and regulate tendon differentiation during development**
**and pathology. This presents a novel potential strategy for the treatment of**
**tendon injuries in the future.”**

**References:**

- [1] Hankemeier S, Keus M, Zeichen J, et al. Modulation of proliferation and differentiation of human
bone marrow stromal cells by fibroblast growth factor 2: potential implications for tissue engineering of
tendons and ligaments. *Tissue Eng.* 2005. 11(1-2): 41-9.
- [2] Li Y, Ge Z, Liu Z, et al. Integrating electrospun aligned fiber scaffolds with bovine serum albumin-
basic fibroblast growth factor nanoparticles to promote tendon regeneration. *J Nanobiotechnology.* 2024.
22(1): 799.
- [3] Schweitzer R, Zelzer E, Volk T. Connecting muscles to tendons: tendons and musculoskeletal
development in flies and vertebrates. *Development.* 2010. 137(17): 2807-17.
- [4] Brent AE, Schweitzer R, Tabin CJ. A somitic compartment of tendon progenitors. *Cell.* 2003. 113(2):
235-48.
- [5] Edom-Vovard F, Schuler B, Bonnin MA, Teillet MA, Duprez D. Fgf4 positively regulates scleraxis and
tenascin expression in chick limb tendons. *Dev Biol.* 2002. 247(2): 351-66.
- [6] Havis E, Bonnin MA, Esteves de Lima J, Charvet B, Milet C, Duprez D. TGF β and FGF promote
tendon progenitor fate and act downstream of muscle contraction to regulate tendon differentiation during
chick limb development. *Development.* 2016. 143(20): 3839-3851.
- [7] Brown JP, Finley VG, Kuo CK. Embryonic mechanical and soluble cues regulate tendon progenitor
cell gene expression as a function of developmental stage and anatomical origin. *J Biomech.* 2014. 47(1):
214-22.
- [8] Eloy-Trinquet S, Wang H, Edom-Vovard F, Duprez D. Fgf signaling components are associated with
muscles and tendons during limb development. *Dev Dyn.* 2009. 238(5): 1195-206.
- [9] Liu C, Zhou N, Li N, et al. Disrupted tenogenesis in masseter as a potential cause of micrognathia.
*Int J Oral Sci.* 2022. 14(1): 50.
- [10] Tokunaga T, Shukunami C, Okamoto N, et al. FGF-2 Stimulates the Growth of Tenogenic Progenitor
Cells to Facilitate the Generation of Tenomodulin-Positive Tenocytes in a Rat Rotator Cuff Healing Model.
*Am J Sports Med.* 2015. 43(10): 2411-22.
- [11] Havis E, Bonnin MA, Olivera-Martinez I, et al. Transcriptomic analysis of mouse limb tendon cells
during development. *Development.* 2014. 141(19): 3683-96.
- [12] Li K, Zhang X, Wang D, Tuan RS, Ker D. Synergistic effects of growth factor-based serum-free
medium and tendon-like substrate topography on tenogenesis of mesenchymal stem cells. *Biomater Adv.*
2023. 146: 213316.
- [13] Li B, Jha RK, Qi YJ, et al. Early cellular responses of BMSCs genetically modified with bFGF/BMP2
co-cultured with ligament fibroblasts in a three-dimensional model in vitro. *Int J Mol Med.* 2016. 38(5):

1578-1586.

**Minor Comments:**

1. Sirius Red Staining in Fig4E and FigS3E: The authors mention that Sirius Red
staining can detect type III collagen, which is associated with fibrosis. However, the
absence of green fluorescence in the relevant figures (e.g., Fig4E and FigS3E) needs
to be addressed. The authors should explain why this expected staining pattern is not
observed or provide additional staining protocols that can better visualize type III
collagen

**Response:**

Thank you very much for your valuable constructive feedback. In Fig 4E, the primary
purpose of employing Sirius Red staining was to assess collagen deposition. It is not
possible to distinguish type III collagen at the cellular level using brightfield microscopy.
In Fig S3E, we focused on detecting the expression of type I collagen through
immunofluorescence.

We visualized type III collagen using the following method: hTSPCs or mTSPCs
were fixed, permeabilized, and blocked. They were then incubated with a primary
antibody against type III collagen, followed by the corresponding secondary antibodies.
Finally, nuclei were stained, and the samples were analyzed using a confocal
fluorescence microscope. For the assessment of type III collagen, we primarily relied
on the immunofluorescence results presented in Fig. 4G and Fig. S3J.

Based on your inquiry for Fig4E, we have also added additional experiments and
data to evaluate the expression of COL3 in human tendon stem/progenitor cells after
14 days of tenogenic induction, using immunofluorescence experiments. Compared to
the AZD4547 treatment group, the addition of rFGF7 significantly suppressed the
production of COL3 protein in human tendon stem/progenitor cells (Fig. R8).

**Fig. 4E.** Sirius Red staining of hTSPCs treated with rFGF7, PBS or AZD4547 after 14
 501 days of tendon lineage culture. Scale bars, 100 μm. **G.** Immunofluorescence staining of
 502 tendon-related proteins (COL1A1, MKX, EGR1) and fibrotic proteins (ACTA2, COL3)
 in hTSPCs treated with rFGF7, PBS or AZD4547 Scale bars, 100 μm.

**Fig. S3E.** Immunofluorescence staining of COL1A1 in mTSPCs treated with rFGF7,
 PBS and mTSPCs from *Fgf7*^{-/-} mice. Scale bars, 100 μm. **J.** Immunofluorescence
 staining of COL3 in mTSPCs treated with rFGF7, PBS and mTSPCs from *Fgf7*^{-/-} mice.
 Scale bars, 100 μm.

**Fig. R8A.** Sirius Red staining of hTSPCs treated with rFGF7, PBS or AZD4547 after

14 days of tendon lineage culture. Independent experiments $n=4$. Scale bars, 100 μm .

2. Editorial Revisions: The manuscript should be carefully edited for spelling and
grammatical errors to enhance readability and clarity.

**Response:**

Thank you very much for your thorough review of our manuscript and for your
valuable constructive feedback. We have meticulously revised the text of our article for
spelling and grammatical errors to enhance readability and clarity.

3. Nomenclature Consistency: The authors could consider using consistent
terminology throughout the manuscript, such as distinguishing between "tendon stem
cells" and "tendon progenitor cells," to avoid confusion.

**Response:**

Thank you very much for your careful review of our manuscript and for your valuable
constructive feedback. We have standardized the terminology throughout the article
named "**tendon stem/progenitor cells**" [1] and highlighted the changes in blue in the
relevant sections.

**References:**

[1] Bi Y, Ehrchiou D, Kilts TM, et al. Identification of tendon stem/progenitor cells and the role of the
extracellular matrix in their niche. Nat Med. 2007. 13(10): 1219-27.

4. Image Guidance: In FigS3 B-C, the authors could provide lower-magnification
images to include more cells in the field of view. This would help readers better assess
the overall distribution and number of cells in the analyzed sections.

**Response:**

Thank you very much for your valuable constructive feedback. In response to your
suggestion, we have conducted lower-magnification images in Fig S3B-C to capture a
wider field of view, thereby incorporating more cells (Fig. S3B-C). We have added
additional experiments and data to the supplementary materials of the article (Fig. S9)
and are highlighted in blue in the revised manuscript (Page 66, Line 1294-1300). A
detailed description of the results can be found in the revised manuscript (Page 11,
Line 231-236) as follows:

“After three days of culture, we observed that rFGF7-treated TSPCs exhibited
significantly enhanced proliferation compared to the control group, as evidenced
by an increased proportion of Ki67⁺ cells and a higher number of EDU-
incorporating cells. Conversely, the *Fgf7*^{-/-} group showed reduced proliferative
capacity, with fewer Ki67⁺ cells (Fig. S3B) and fewer EDU-incorporating cells (Fig.
S3C). This indicates that FGF7 promotes the proliferation of mouse TSPCs.”

**Fig. S3B-C.** Immunofluorescence staining of KI67 protein in mTSPCs treated with
rFGF7, PBS, and mTSPCs from wildtype mice or *Fgf7*^{-/-} mice(i) and semi-quantitative
fluorescence analysis (ii) (*n*=3 independent experiments, bars represent mean ± SD;
**P*<0.05, ***P*<0.01, two-tailed Student's *t* test). Scale bars, 100 μm. **C.**
Immunofluorescence staining of EDU protein in mTSPCs treated with rFGF7, PBS,
and mTSPCs from *Fgf7*^{-/-} mice (i) and semi-quantitative fluorescence analysis(ii) (*n*=5
independent experiments, bars represent mean ± SD; ****P*<0.001, two-tailed Student's
*t* test). Scale bars, 100 μm.

Reviewer #2 (Remarks to the Author):

This manuscript from Lin et al., is focused on the role of FGF7 in soft tissue fibrosis
and regeneration using the tendon as a model to understand fundamental mechanisms
of these processes. The authors are commended on a substantial body of work and
the data on the effects of FGF7 on the tendon healing process are very compelling.
However, there are substantial concerns about the rigor of several aspects of the
experimental design, conclusions that are not well-supported by the data, and potential
alternative interpretations of the data that are not discussed, substantially reducing the
impact of these studies. Specific comments are below.

We are truly grateful for your positive feedback and for recognizing the originality
and significant impact of our study. Your insights have been invaluable in enhancing
the quality of our work. We appreciate your encouragement and support, which
motivate us to continue exploring this important area of research. Your comments are
presented in black text, while our responses are in blue text, with any revisions or
additions to the manuscript highlighted in bold blue. In the revised manuscript, all
modifications and additions are displayed in blue font. Thank you once again for your
thoughtful review.

Major comments

1. Rigor of the experimental design. There are several major concerns here:

(1) The isolation of 'tendon stem cells' from ACL is not appropriate- tendons and
ligaments are compositionally different, also these were isolated from ACL
reconstructions but text says patients were healthy- any cells taken the from the
remnant are by definition not healthy.

**Response:**

Thank you very much for highlighting these important issues. Our previous wording
regarding the source of the cells may have led to some ambiguity. To clarify, the tendon
stem/progenitor cells (TSPCs) used in our study were not directly derived from the
anterior cruciate ligament (ACL) itself, but rather from the hamstring tendon tissues
obtained during ACL reconstruction surgery, where they were utilized as autologous
grafts.

We have revised the methodology section accordingly, highlighted in blue in the
revised manuscript (Page 29, Line 615-616): "**Human tendon stem/progenitor cells
(hTSPCs) were collected from hamstring tendon tissues used as autologous
grafts during ACL reconstruction surgeries.**"

(2) In addition, the interchangeable use of different anatomic tendon sites is not
appropriate. Different tendons are functionally and compositionally distinct. As such,
use of healthy hamstring tendon cells cannot be used to demonstrate that shoulder
tendon pathology is enriched for the fibrotic population as this may be due to
differences in composition between these tendons rather than pathology.

**Response:**

We sincerely appreciate the reviewer's insightful comment. We acknowledge that
the ideal control for rotator cuff pathology would be healthy rotator cuff tendon tissue.
However, we did not use healthy rotator cuff tendon tissue primarily because:

1. Obtaining truly healthy rotator cuff tendon samples from human donors presents
considerable practical and ethical challenges. Healthy rotator cuff tendons are rarely
available from surgical procedures. Due to these substantial difficulties in sourcing
normal rotator cuff tendon tissue, we utilized healthy hamstring tendon samples as a
readily available surrogate control.

2. To ensure the integrity of our manuscript, we have made the following limitation
modifications to the discussion section, highlighted in blue in the revised manuscript
(Page 27, Line 595-598): **"Due to the difficulty in obtaining normal rotator cuff
tissue samples from human subjects, we utilized hamstring tendon tissue as a
control. Variability between tendon tissues from different anatomical sites is, of
course, unavoidable."**

3. While we recognize the theoretical possibility of inter-tendon heterogeneity, we
have made every effort to address your concerns by comparing our single-cell
sequencing data from healthy hamstring tendon tissue with single-cell data from
normal rotator cuff tendon tissue found in the literature (PMID: 37280595) [1]. We
extracted three normal rotator cuff single-cell datasets from that study and combined
them with our own seven normal hamstring tendon single-cell datasets for clustering
and cell proportion analysis (Fig. R9). The proportions of tendon cell subpopulations in
both datasets were found to be largely similar. Therefore, we contend that the observed
enrichment of the fibrotic subpopulation (*Lox⁺Uts2r⁺*) in pathological rotator cuff
tendons, as compared to healthy hamstring tendons, is likely attributable to disease
pathology rather than inherent baseline differences between the two tendon sites.

**Fig. R9A.** UMAP plots of single-cell sequencing data from normal rotator cuff and
 normal hamstring tendon tissues. **B-C.** Statistical data on the proportions of each cell
 subpopulation, along with specific numerical values for each proportion.

**References:**

[1] Fu W, Yang R, Li J. Single-cell and spatial transcriptomics reveal changes in cell heterogeneity during
 progression of human tendinopathy. BMC Biol. 2023. 21(1): 132.

(3) In addition, it is not clear how the phenotype of *Fgf^{-/-}* mice, which is developmentally
 induced (even though the phenotype does not present until ~P21) is the same as
 mature pathology from shoulder arthroscopy with poorly defined pathology.

**Response:**

We acknowledge the differences in etiology between the *Fgf7^{-/-}* mouse model and
 clinical mature pathologies. Thank you very much for your careful review of our
 manuscript and for your valuable constructive feedback. Choosing the *Fgf7^{-/-}* mouse
 model provides insights into how stem/progenitor cells are misdirected towards a
 fibrotic fate when critical FGF7 signaling is disrupted. This approach aligns with a core
 principle of regenerative medicine: effective tissue regeneration often requires
 recapitulating key developmental processes. We hypothesize that pathological fibrosis
 following tendon injury arises from the failure to activate critical developmental
 programs or from the erroneous activation of fibrotic pathways. By utilizing this

developmental model, we have clarified that the absence of FGF7 signaling misdirects
 stem cells towards fibrosis and identified the key *Lox⁺Uts2r⁺* pathological cell
 subpopulation associated with tendon fibrosis. Notably, this pathological subpopulation
 is also more prevalent in human pathological tendon samples compared to normal
 human tendon samples.

While the developmental defects observed in mice differ from human rotator cuff
 tendon disorders, they also offer valuable insights into the regulatory mechanisms
 governing the choice between fibrosis and tendon differentiation.

2. Related to the above, it does not appear that proper littermate controls as are used
 for *Fgf7*^{-/-} as the methods note that wildtype mice were purchased from GEM. This
 substantially detracts from the rigor of the sequencing data as slight differences in
 genetic background can impact expression profiles.

**Response:**

We fully agree that the use of stringent littermate controls is essential for
 minimizing genetic background differences and ensuring the reliability of the gene
 expression results.

In our experiments, all *Fgf7*^{-/-} mice and their matched wild-type (WT) controls were
 derived from the same litter. To accurately reflect our experimental design and
 eliminate any ambiguity, we have made the following modifications to the methods
 section, highlighted in blue in the revised manuscript (Page 34, Line 739-741): "***Fgf7*^{+/-}
 heterozygous mice were obtained from GEM. Littermate *Fgf7*^{-/-} homozygous
 mice and wild-type controls were generated from crosses of these
 heterozygotes**"

To clearly indicate that the wild-type control group consists of littermate wild-type
 mice. We will also include a screenshot of the relevant information from GEM regarding
 the purchased mice.

A	B	C	D	E	F	G	H	I	J
ID No.	Sex	Birth Date	Age (weeks)	Gene Fgf7	Generation	Breeding Cage	Coat C ₀	Tail Clipping Time	Mouse Procedure
♂	2024/5/22	10w	WT/WT	F1P2F5	26	Black	2024/5/30	Y
♂	2024/5/22	10w	-2340bp/-2340bp	F1P2F5	26	Black	2024/5/30	Y
♂	2024/5/22	10w	-2340bp/wt	F1P2F5	26	Black	2024/5/30	Y
♂	2024/5/22	10w	-2340bp/-2340bp	F1P2F5	26	Black	2024/5/30	Y
♂	2024/5/22	10w	WT/WT	F1P2F5	26	Black	2024/5/30	Y

3. It is not clear that mouse and rat TSPCS were actually isolated. The isolation
protocol would isolate ~all cells in the tendon and all tendon fibroblast populations are
adherent- there is no selection for any cells with 'stem' or 'progenitor' properties.

**Response:**

Thank you for your valuable feedback and for your detailed and professional review.
We utilized tendon stem/progenitor cells (TSPCs) in our study. We have followed the
methods outlined in Nature Medicine for this process[1].

Mouse and rat tendons were dissected as follows: we stripped off the tendon sheath
and the surrounding paratenon, cut tendon tissues into small pieces and digested with
collagenase type I and dispase in PBS for 1 h at 37 °C. Single-cell suspensions were
cultured in a-MEM (Gibco), supplemented with 20% lot-selected FBS (Equitech-Bio)
and 100 mM 2-mercaptoethanol (Gibco) for 8-10 d at 5% CO₂, 37°C. We utilized rat
TSPCs and mouse TSPCs that exhibited colony-forming ability following clonal
selection for subsequent experiments.

In our previous research, we performed flow cytometric analysis of the
corresponding stem cells markers[2]. In the current study, we conducted colony-
forming and tri-lineage differentiation assays for rat tendon stem/progenitor cells
(TSPCs) (Fig. R10) as well as for mouse TSPCs (Fig. R13). The results are as follows:

**Fig. R10A.** Multipotential differentiation capacity for Osteogenesis, Adipogenesis and

Chondrogenesis of mTSPCs (Scale bar: 100 μm). **B.** CFU assay for assessing the self-
renewal of mTSPCs. **C.** Multipotential differentiation capacity for Osteogenesis,
Adipogenesis and Chondrogenesis of Rat TSPCs (Scale bar: 100 μm). **D.** CFU assay
for assessing the self-renewal of Rat TSPCs.

**References:**

- [1] Bi Y, Ehrichtou D, Kilts TM, et al. Identification of tendon stem/progenitor cells and the role of the
extracellular matrix in their niche. *Nat Med.* 2007. 13(10): 1219-27.
[2] Yin Z, Hu JJ, Yang L, et al. Single-cell analysis reveals a nestin(+) tendon stem/progenitor cell population
with strong tenogenic potentiality. *Sci Adv.* 2016. 2(11): e1600874.

4. The clinical relevance of the biopsy punch model in the patellar tendon is unclear

**Response:**

We sincerely appreciate the reviewer for raising this important point. The primary
purpose of designing and utilizing a full-thickness defect model in the tendon for our
research is to simulate clinically challenging tendon tear that are difficult to heal on
their own. This high degree of stability and reproducibility in modeling is crucial for
conducting rigorous and comparable preclinical studies. The model we utilized, a full-
thickness tendon defect, has also been employed in previous tendon research[1-4].
This model allows for the precise and consistent creation of full-thickness tissue
defects of predetermined size and depth within the tendon.

**References:**

- [1] Wang Y, Jin S, Luo D, et al. Functional regeneration and repair of tendons using biomimetic scaffolds loaded
with recombinant periostin. *Nat Commun.* 2021. 12(1): 1293.
[2] Wang Z, Lee WJ, Koh B, et al. Functional regeneration of tendons using scaffolds with physical anisotropy
engineered via microarchitectural manipulation. *Sci Adv.* 2018. 4(10): eaat4537.
[3] Yuan X, Zhu Z, Xia P, et al. Tough Gelatin Hydrogel for Tissue Engineering. *Adv Sci (Weinh).* 2023. 10(24):
e2301665.
[4] Zhang H, Chen Y, Fan C, et al. Cell-subpopulation alteration and FGF7 activation regulate the function of
tendon stem/progenitor cells in 3D microenvironment revealed by single-cell analysis. *Biomaterials.* 2022. 280:
121238.

5. TEM cannot be used to assess mechanics

**Response:**

We sincerely appreciate your valuable feedback. We fully agree with you. As you
pointed out, transmission electron microscopy (TEM) is primarily used for observing
ultrastructural details and is not suitable for directly evaluating the macroscopic
mechanical properties of materials. In this study, we strictly adhered to the standards
for biomechanical testing and specifically employed the widely recognized method of
using an Instron tensile/compression system (Model 5543, Instron, Canton, MA) to
accurately measure the mechanical properties of the repaired tendon tissue. The
relevant experimental data and schematic representations are provided below (Fig. 9
M-N):

**Fig. 9M-N.** Schematic of mechanical testing for different experimental groups (Control,
GelMA, GelMA-c-rFGF7, WT: wildtype) at 4 weeks post-surgery; Statistical graphs of
mechanical testing results at 4 weeks post-surgery for different experimental groups
(Control, GelMA, GelMA-c-rFGF7, WT: wildtype), showing Stress at Failure,
Modulus and Failure Force (from top to bottom) ($n=6$ per group, wildtype group $n=5$,

bars represent mean \pm SD; NS: not significant, * P <0.05, ** P <0.01, *** P <0.001, two-
tailed Student's t test).

6. The data do not support a fibrotic phenotype in *Fgf7*^{-/-}. As the authors outline,
fibrosis is characterized by excess and disorganized collagen formation. *Fgf7*^{-/-} mice
have a substantial decline in collagen production/deposition (shown particularly
strikingly on picro/PLM). Moreover, there seems to be a decline in tenogenic capacity,
although whether this is more of a response to improper collagen patterning is
unknown. The overall cell density between WT and *Fgf7*^{-/-} does not appear to be
drastically different. There is a clear increase in Col 3, but the increase in α SMA does
not seem to be related to myofibroblasts (e.g. there are not obvious stress fibers and
this seems to be matrix staining). The authors do somewhat address this but another
explanation is that *Fgf7*^{-/-} deficiency impairs Col1 production and or maturation and
the increase *Lox*⁺ cells is an attempt to address this deficit by restoring the needed
cross-linking.

**Response:**

We sincerely appreciate your in-depth and insightful reviews and comments. We
completely agree with your observations regarding the *Fgf7*^{-/-} deficiency impeding the
production and maturation of collagen type I, as well as the increase in *Lox*⁺ cell
numbers, which appears to compensate for this deficiency by restoring the necessary
cross-linking activities. We have made the following modifications to the discussion
section, highlighted in blue in the revised manuscript (Page 24, Line 524-533):

**“In the context of load-bearing tendon regeneration, the characterization of**
**fibrosis transcends mere collagen overproduction, focusing instead on**
**pathological matrix remodeling. Tissues attempt to aberrantly activate**
**processes, such as the increase of α -SMA, and exhibit abnormal cross-linking,**
**as indicated by the rise in *Lox*⁺ cells and the increased presence of more easily**
**reshaped yet mechanically weaker collagen type III, in an effort to compensate**
**for mechanical deficiencies. This compensation fails to yield functional tissue,**

ultimately leading to structural stiffness, increased brittleness, and a decline in
functional capacity for tendon formation. The culmination of these pathological
processes results in tissues that cannot withstand physiological loads,
rendering them prone to rupture and healing failure.”

7. 'Fgf7 primarily expressed in tendon fascia tissue (indicated by OCN protein)'- it is
not clear what the authors mean by this. They suggest it is a niche for stem cells but
there is broad expression of OCN in the tendon body that is not restricted to the
paratenon/epitenon, as such there is not strong evidence for regulation of stemness
via this mechanism.

**Response:**

We sincerely appreciate your meticulous review and valuable feedback. While
previous studies have reported OCN as a marker for tendon paratenon/epitenon tissue
[1], to maintain rigor in our study, we have opted to remove the co-staining data for
OCN and FGF7, along with the associated textual descriptions, to eliminate any
potential ambiguity.

Our results confirm that FGF7 is primarily localized within the tendon
paratenon/epitenon, which is consistent with the distribution patterns reported by other
research groups[3]. We have also used the reported marker Sca-1 to identify tendon
paratenon/epitenon tissue[2]. We believe that the co-staining results with Sca-1 and
the findings from other studies regarding FGF7 expression sufficiently support the
notion that FGF7 is primarily expressed in tendon paratenon/epitenon tissue (Fig. S2
F).

Literature indicates that tendon paratenon/epitenon tissue is an important source of
tendon stem/progenitor cells[2, 4]. Based on the spatial enrichment evidence of FGF7
in tendon tissue and our functional data, we speculate that FGF7 expressed within the
paratenon/epitenon microenvironment could be a crucial local factor influencing the
behavior of stem/progenitor cells in that region. Of course, this does not exclude the

possibility that other mechanisms or factors regulating stem cell characteristics may
exist within the tendon. The role of FGF7 may also be synergistic with other secreted
factors from the fascia. This is an area that we aim to explore further in our future
research.

**References:**

[1] Wang Y, Zhang X, Huang H, et al. Osteocalcin expressing cells from tendon sheaths in mice contribute to
tendon repair by activating Hedgehog signaling. *Elife*. 2017. 6: e30474 [pii]. PMID: 29244023

[2] Harvey T, Flamenco S, Fan CM. A Tpp3(+) Pdgfra(+) tendon stem cell population contributes to
regeneration and reveals a shared role for PDGF signalling in regeneration and fibrosis. *Nat Cell Biol*. 2019.
21(12): 1490-1503. PMID: 31768046

[3] Ren Y, Wan R, Zhao G, et al. Gene expression of Postn and FGF7 in canine chordae tendineae and their
effects on flexor tenocyte biology. *J Orthop Res*. 2024. 42(5): 961-972. PMID: 37990927

[4] Huang Z, Yin Z, Xu J, et al. Tendon Stem/Progenitor Cell Subpopulations and Their Implications in Tendon
Biology. *Front Cell Dev Biol*. 2021. 9: 631272. PMID: 33681210

8. In vitro data- the authors conclude that Fgf7 promotes tenogenic fate, however
does Fgf7 actually just maintain the baseline tendon cell phenotype better than is
typically observed in vitro? This is still an important finding the basis for this
interpretation is not clear.

**Response:**

We sincerely appreciate your insightful comments and valuable feedback. Fgf7 both
maintains the baseline tendon cell phenotype and actively drives these cells toward a
more defined and mature tendon cell phenotype.

Referring to our data, under standard control culture conditions (without FGF7),
tendon stem/progenitor cells did not express high levels of tendon-specific markers.
Upon the addition of FGF7, we observed a significant and sustained upregulation of
tendon marker protein expression. In the absence of FGF7, the cell phenotype undergo
regression (Fig S3E, F, G, H, K). Our data demonstrate that the effects of FGF7

treatment represent a positive tenogenic fate promotion. For instance, *Tnmd*, *Egr1* and
 *Mkx* (the markers for tendon cell phenotype) are significantly upregulated, supporting
 the concept of promoting differentiation/maturation (Fig S3 F, G, H, K).

In our in vitro tendon explant culture data, we found that the inclusion of FGF7 better
 maintains the baseline tendon cell phenotype, as evidenced by sustained the
 expression of tendon cell phenotype proteins, such as the high level of *Scx* (Fig 8A,
 C).

**Fig. S3E-H.** Immunofluorescence staining of COL1A1, MKX, EGR1, TNMD protein
 in mTSPCs treated with rFGF7, PBS and mTSPCs from *Fgf7*^{-/-} mice. Independent
 experiments $n=5$. Scale bars, 100 μm. **K.** The corresponding semi-quantitative
 fluorescence analysis of tendon-related proteins (COL1A1, MKX, EGR1, TNMD) in
 mTSPCs treated with rFGF7, PBS and mTSPCs from *Fgf7*^{-/-} mice ($n=5$ independent
 experiments, bars represent mean \pm SD; ** $P<0.01$, *** $P<0.001$, two-tailed Student's t
 test).

**Fig. 8. A.** Schematic representation of the explant experiment. Achilles tendons from
 SCXGFP mice were extracted and cultured with rFGF7 or PBS. **C.** Confocal
 immunofluorescence imaging showing GFP protein expression in tendon explants from
 SCXGFP mice ($n=3$, independent experiments, bars represent mean \pm SD; ** $P<0.01$,
 two-tailed Student's t test). Scale bars, 100 μm .

9. (1) The rationale for the inclusion of enthesis cells and MTJ cells in the
 subsequent tenocyte sub-clustering is unclear.

**Response:**

We sincerely appreciate your valuable feedback and thorough review. Our
 preliminary clustering analysis demonstrates that many cells from these regions exhibit
 overall transcriptomic characteristics typical of tendon cells, rather than being distinctly
 categorized as chondrocytes or myocytes. In the tendon insertion sites, such as the
 tendon-bone interface, we identified a subset of cells (subgroup 3) that expresses both
 chondrogenic and tendinous characteristics (Fig S4C). Previous studies have
 indicated that the fibrocartilage cells from the enthesis and specific cell types from the
 MTJ share many key molecular characteristics and signaling pathways with the core
 tenocytes, and these cells are routinely included in tendon tissue clustering analyses[1,
 2]. Excluding these cells might overlook important cellular states that could play a role
 in the overall function or pathological processes of the tendon.

**Fig. S4A.** Uniform Manifold Approximation and Projection (UMAP) plot displaying
 31,756 cells divided into 17 distinct subpopulations. The black circle subpopulation
 includes subgroups 0, 1, 2, 3, and 4, which exhibit tendon characteristics. The red circle
 includes subgroup 3 (Enthesis cells). **C.** UMAP plots indicating the expression of
 specific marker genes for various cell subpopulations: tenocytes (*Tnmd*, *Scx*, *Thbs4*,
 *Tppp3*) and enthesis cells (*Col2a1*). Notably, the red circle highlights subgroup 3
 (enthesis cells), which includes the expression of tendon-specific genes (*Tnmd*, *Scx*,
 *Thbs4*).

**References:**

- [1] Fang F, Xiao Y, Zelzer E, Leong KW, Thomopoulos S. A mineralizing pool of Gli1-expressing progenitors
 builds the tendon enthesis and demonstrates therapeutic potential. *Cell Stem Cell*. 2022. 29(12): 1669-1684.e6.
 [2] Yan R, Zhang H, Ma Y, et al. Discovery of Muscle-Tendon Progenitor Subpopulation in Human Myotendinous
 Junction at Single-Cell Resolution. *Research (Wash D C)*. 2022. 2022: 9760390.

(2) The unique presence of *Utsr2/Lox* population is clear but it would be helpful to
 know where these genes are expressed prior to subclustering the tendon cells.

**Response:**

We sincerely appreciate your valuable feedback and thorough review. We have
 added additional data analysis to the UMAP plots of the overall cells in tendon tissue
 prior to subclustering. we have observed that the expression of *Uts2r* and *Lox* is highly

specific and co-localized in tendon cell population (Fig. R11). The results are illustrated
in the figure below:

**Fig. R11A.** Uniform Manifold Approximation and Projection (UMAP) plot displaying
31,756 cells divided into 17 distinct subpopulations. The circle subpopulation includes
subgroups 0, 1, 2, 3, and 4, which exhibit tendon characteristics. **B.** UMAP plots
indicating the expression of Lox^+Uts2r^+ fibrotic subcluster is primarily concentrated
within the tendon characteristics subpopulation.

(3) The human data does not provide strong support for these findings (e.g., the
lack of this population in the healthy tissue or enrichment in the diseased tissue as
neither is an appropriate match (tendon or pathology) for the mouse data.

**Response:**

We sincerely appreciate your valuable suggestions. We fully agree that the
mismatch limits the use of human data as direct evidence for mouse-specific cell
populations enriched in Lox^+Uts2r^+ . Nevertheless, our immunofluorescence results
reveal significant results that extend beyond mere validation. In clinical samples of
diseased tendon tissue, we have observed significant expression of Lox^+Uts2r^+ in the
extracellular matrix of diseased tendon cells, with almost no co-expression in healthy
tendon tissue (Fig 6G). The identification of this phenotype in human samples is more
translationally relevant than perfect matches with mouse validation results. The core
value of utilizing human samples lies in suggesting that this phenotype is associated
with clinical tendon pathology and laying the groundwork for therapies targeting
abnormal matrix remodeling (for example, the potential application of *Lox* inhibitors in

tendon disorders). Our findings based on tendon disease patients—who experience a
high prevalence of this condition—may contribute to the identification of future
biomarkers for diagnostic and therapeutic purposes. In summary, the human data from
our study do not merely replicate the conclusions drawn from mouse models. They
reveal a conserved and actionable pathological mechanism in tendons through
clinically relevant samples.

**Fig. 6G.** Fluorescence staining showing co-expression of UTS2R and LOX in human
normal tendon samples(n=11) and tendinopathy samples(n=64). Scale bars, 100 µm.

(4) In addition, the bulkseq data is much more consistent within age groups than
genotypes but this is not discussed.

**Response:**

We sincerely appreciate your detailed review of our manuscript and your valuable
feedback. We fully agree that rigorously controlling for age variables is essential for
accurately interpreting genotype differences. We apologize for any ambiguity caused
by our previous descriptions, as we utilized 10-week-old mice of the same age. We
have made the following modifications to the method section, highlighted in blue in the
revised manuscript (Page 32, Line 693-694):

**“All mice used for RNA-seq analysis were strictly matched at 10 weeks of age,
emphasizing our control of age consistency during the experimental design**

phase.”

10. The conclusion that the Lox/Ustr subpopulation is ‘a critical driver of the fibrotic
phenotype’ is not supported by the data. The population is certainly enriched but this
is not causal evidence for the phenotype of Fgf7^{-/-} mice.

**Response:**

We sincerely appreciate your profound insights regarding the key conclusions of our
manuscript. We have made the following modifications and highlighted in blue in the
revised manuscript (Page 15, Line 318-320):

**“We identified the Lox⁺Ust2^{r+} subpopulation (subcluster 0) is a key**
**contributing factor to the fibrotic phenotype observed in 8-week Fgf7^{-/-} tendon**
**tissues, exhibiting a fibrotic state and propagating fibrosis-related signaling to**
**other subpopulations.”**

11. “Given the continuous cellular turnover in tendons” this is not supported by the
current literature- Galloway has demonstrated limited turnover (~5% of cells) in
tendons at P21.

**Response:**

We sincerely appreciate your detailed and professional review. Thank you for your
valuable feedback and for highlighting the important research by Galloway et al^[35]. We
fully acknowledge your point, especially as Galloway et al. clearly demonstrate that the
turnover rate of tendon cells at P21 is very limited (approximately 5%). We will revise
the term "continuous" to avoid misrepresentation. We have made the following
modifications and highlighted in blue in the revised manuscript (Page 15, Line 328-
331):

**“Given the evidence of cellular turnover within tendon tissue (Galloway et al.**
**reported approximately 5% turnover in tendon cells at P21) [1] , and the near**

**absence of the *Lox⁺Uts2r⁺* fibrotic subcluster in normal mature tendons, we noted**
**its substantial presence in the fibrotic tendon tissues of *Fgf7^{-/-}* mice.”**

**References:**

[1] Grinstein M, Dingwall HL, O'Connor LD, Zou K, Capellini TD, Galloway JL. A distinct transition from cell
growth to physiological homeostasis in the tendon. *Elife*. 2019. 8: e48689.

**Minor comments**

1. Much of the introduction on other FGFs is not relevant to the central premise of the
study-is more known about *Fgf7* and collagen?

**Response:**

Thank you for your valuable suggestions regarding the introduction section. We have
enhanced the introduction by incorporating additional information related to FGF7 and
collagen, highlighted in blue in the introduction part of revised manuscript (Page 5, Line
100-107) as follows:

**“FGF7 promotes collagen formation, positively influencing wound healing**
**during the skin wound healing process[1]. Existing studies indicate that the III**
**domain of proteoglycans, which interact with collagen in the matrix, can bind to**
**FGF7. This binding not only serves as a reservoir for FGF7 but also facilitates**
**its release when needed, thereby promoting cell proliferation and**
**differentiation[2, 3]. In a CCL4-induced acute liver injury model, FGF7 has been**
**shown to assist hepatocyte survival, consequently mitigating the extent of**
**fibrosis[4].”**

**References:**

[1] Kim YS, Lee JS, Jeong MY, Jang JW, Kim MS. Recombinant human fibroblast growth factor 7 obtained
from stable Chinese hamster ovary cells enhances wound healing. *Biotechnol J*. 2024. 19(5): e2300596.

[2] Mongiat M, Taylor K, Otto J, et al. The protein core of the proteoglycan perlecan binds specifically to
fibroblast growth factor-7. *J Biol Chem*. 2000. 275(10): 7095-100.

[3] Hayes AJ, Farrugia BL, Biose IJ, Bix GJ, Melrose J. Perlecan, A Multi-Functional, Cell-Instructive, Matrix-
Stabilizing Proteoglycan With Roles in Tissue Development Has Relevance to Connective Tissue Repair and

Regeneration. Front Cell Dev Biol. 2022. 10: 856261.

[4] Geervliet E, Terstappen L, Bansal R. Hepatocyte survival and proliferation by fibroblast growth factor 7
attenuates liver inflammation, and fibrogenesis during acute liver injury via paracrine mechanisms. Biomed
Pharmacother. 2023. 167: 115612.

2. It is not clear why some data are presented +/- SEM and others have +/- SD.

**Response:**

We sincerely appreciate your valuable feedback and your thorough review of our
manuscript. We have carefully revised the methods section to clearly indicate that data
variability is represented using standard deviation (SD) rather than standard error of
the mean (SEM). The modifications made to the methods section are highlighted in
blue in the revised manuscript (Page 37, Line 815-816).

**“All data are presented as mean \pm SD, representing at least three independent
experiments.”**

3. Figure 8- there is no apparent EGR1 staining in the images provided despite the
quantification suggesting a difference between groups.

**Response:**

Thank you very much for your valuable feedback. We have provided higher-
resolution images for better readability and have replaced them with more
representative images from the same group. The updated high-resolution images are
as follows in the revised manuscript (Page 57, Line 1159-1162):

**Fig. 8H.** Immunofluorescence staining of tendon related proteins (EGR1). Scale bars,
100 μm .

4. The presentation of much of Figure 9 is confusing- the y-axis of several figures
suggests that a decrease in stride length and width are 'better' which is not correct.

**Response:**

Thank you very much for your valuable feedback. We have corrected the relevant
images by removing the terms "better" and the arrows from the y-axis. The revised
results are as follows in the revised manuscript (Page 59, Line 1167-1172):

**Fig. 9. (B)** Schematic of the footprint analysis experiment. **(C-G)** Gait spatial parameter
statistics at different time points (2w, 4w) for different experimental groups (Control,
GelMA, GelMA-c-rFGF7, wildtype): stride length (C), stride width (D), step length (E),
paw spatial parameters: toe spread (F), paw length (G) ($n=6$ per group, bars represent
mean \pm SD; * $P<0.05$, ** $P<0.01$, *** $P<0.001$, two-tailed Student's t test).

Reviewer #3 (Remarks to the Author):

Tendon injuries are common and can severely impair patient mobility and
productivity, resulting in a significant socioeconomic burden and reduced quality of life.

The tendon healing process results in the formation of a fibrotic scar, and as a

consequence, injured tendons never regain the mechanical strength of their uninjured
counterparts, leading to frequent re-injury. In this study, the authors demonstrated that
fibroblast growth factor 7 (FGF7) promotes the proliferation and tenogenic
differentiation of tendon stem cells while inhibiting fibrotic progression. Furthermore, a
GelMA hydrogel loaded with recombinant FGF7 (GelMA-c-rFGF7) significantly
enhanced functional tendon regeneration in vivo. These findings highlight the pivotal
role of FGF7 in preventing tendon fibrosis and promoting regeneration, offering
promising therapeutic strategies for tendon diseases. The following issues need to be
addressed in the revised manuscript:

Thank you for your affirmation of our innovative contributions. It truly motivates us to
continue our research efforts. Your comments are presented in black text, while our
responses are in blue text, with any revisions or additions to the manuscript highlighted
in bold blue. In the revised manuscript, all modifications and additions are displayed in
blue font. Thank you once again for your thoughtful review.

Major points:

1. The study utilized FGF7 global knockout (Fgf7^{-/-}) mice, which exhibited significant
abnormalities in the development of load-bearing tendons. These findings highlight the
essential role of FGF7 in regulating tenogenic differentiation. However, the cellular and
molecular mechanisms by which FGF7 prevents tendon fibrosis and promotes
regeneration remain unclear. The global knockout of FGF7 lacks spatial and temporal
specificity, and most experimental data were obtained under physical conditions that
do not accurately reflect tendon repair and regeneration, particularly in adult mice.
Therefore, it would be beneficial to provide in vivo evidence that clarifies the fate-
determinant effects of FGF7 on tendon stem/progenitor cells (TSPCs) and investigates
the underlying molecular mechanisms during tendon repair or regeneration, especially
under conditions where FGF7 signaling is specifically modified in tendons or TSPCs in
future studies.

**Response:**

We sincerely appreciate your thorough review of our manuscript and the valuable
insights you have provided. We will thoroughly investigate the limitation you mentioned
regarding the lack of spatial and temporal specificity in our future research. Currently,
our use of *Fgf7*^{-/-} mice primarily highlights the protective role of FGF7 in load-bearing
tendon tissues and its antifibrotic effects during tendon injury healing. In our future
research, we will employ spatiotemporally specific genetic manipulation models in
conjunction with tendon injury models to elucidate the precise cellular and molecular
mechanisms by which FGF7 signaling regulates the fate of tendon stem/progenitor
cells, inhibits fibrosis, and promotes functional tendon regeneration within repair
environment. We believe that these subsequent studies will greatly enhance our
understanding of the mechanisms underlying tendon repair and regeneration.

2. The newly identified *Lox*⁺*Uts2r*⁺ fibrotic subcluster appears to play a key role in
fibrosis during tendon healing, and this should be confirmed in wild-type (WT) mice
during tendon repair and regeneration. Furthermore, it is essential to investigate how
the cellular composition, particularly the *Lox*⁺*Uts2r*⁺ fibrotic subcluster, changes during
fibrotic tendon repair and FGF7-induced tendon regeneration using single-cell RNA
sequencing. Additionally, the cellular origin of the *Lox*⁺*Uts2r*⁺ fibrotic subcluster
warrants further exploration.

**Response:**

We sincerely appreciate your insightful suggestions. We have supplemented our
study with observations of the dynamic changes of the *Lox*⁺*Uts2r*⁺ fibrotic subcluster
during tendon fibrosis repair in wildtype (WT) mice. We analyzed the single-cell RNA
sequencing results from fibrotic tendon repair and healthy mouse tendon samples. The
circled regions in the figure (Fig. R12) indicated cell populations that increased
following injury. Our findings reveal a significant increase in the *Lox*⁺*Uts2r*⁺ fibrotic
subcluster (as indicated in Cluster 0) during the tendon injury process compared to
normal tendons (Fig. R12).

Following your recommendation, we conducted pseudotime analysis based on the
scRNA-seq data to elucidate the differentiation trajectory of this subpopulation in our
*Fgf7*^{-/-} and wildtype mice tendon tissue. We found that these cells are likely derived
from the *Cd44*⁺*Tppp3*⁺ cell subpopulation, which will be a focus of our future
investigations (Fig. R13).

**Fig. R12A.** A Uniform Manifold Approximation and Projection (UMAP) plot is

presented divided into 9 distinct subpopulations. **B.** UMAP plots illustrating the
expression of fibrotic marker genes *Lox*, *Uts2r*, *Acta2*. The subpopulation within the
black circle represents the increased number of cells following injury, where the
Lox^+Uts2r^+ cell subpopulation shows elevated expression.

**Fig. R13A.** Pseudotime trajectory of the 9 tendon characteristics subclusters depicting
the differentiation of subpopulations.

3. This study aimed to clarify the effects and mechanisms of FGF7 in preventing tendon
fibrosis and promoting regeneration, with a particular focus on the
fibroblastic/proliferative phase of tendon healing. However, its anti-fibrotic effects may
not extend to fibrosis in other organs or tissues. For instance, intra-articular injection
of recombinant FGF7 (rFGF7) has been shown to promote joint inflammation and
articular cartilage degradation. Instead of referencing the general process of fibrosis,
it is recommended to concentrate on the biological processes of tendon healing and
regeneration in the introduction. Additionally, the title should be modified accordingly.

**Response:**

We sincerely appreciate your valuable constructive feedback. We completely agree
your suggestion to focus on tendon-specific biological processes while it also providing
insights relevant to the study of fibrosis repair in collagen-rich organs. We have made
the following modifications to the introduction section, highlighted in blue in the revised
manuscript (Page 4, Line 62-79):

**“Tendon injuries and degenerative diseases, such as tendinitis and**
**tendinopathy, present significant clinical challenges, often associated with poor**
**repair and collagen remodeling. Therefore, understanding tendon fibrosis, a key**
**pathological outcome following tendon injury, is essential. Tendon fibrosis**
**refers to a pathological remodeling process that occurs in tendon tissue after**
**injury or chronic lesions. Its core characteristic involves destructive changes in**
**the intrinsic structure and function of the tendon’s extracellular matrix (ECM).**
**As a dense and highly organized connective tissue primarily composed of type**
**I collagen, tendon fibrosis is not characterized by the replacement of functional**
**tissue with excessive ECM, as seen in liver or lung fibrosis. Instead, it manifests**
**as a dysregulation of the inherent ECM composition, including an abnormal**
**increase in type III collagen, and disorganization of the spatial structure, which**
**includes the loss of the alternating parallel arrangement of collagen fibers, a**
**reduction in fibril diameter leading to a small and uniform appearance, abnormal**
**proteoglycan deposition, and neovascularization. Under transmission electron**
**microscopy, the ultrastructure of the pathological extracellular matrix (ECM)**
**reveals a fine and uniform distribution of collagen diameters. These alterations**
**result in a significant decline in the biomechanical properties of the tendon, such**
**as elasticity and tensile strength, severely impacting its load transmission**
**function and ultimately leading to tendon re-rupture and poor functional**
**outcomes for patients.”**

In light of your comments, we have modified the title to (Page 1, Line 1-2; Page 62,
Line 1224-1225):

**"Harnessing FGF7 for load-bearing tendon regeneration: from fibrosis**
**suppression to functional healing."**

Minor points:

1.The HE staining of paired or adjacent sections of the fluorescence images in Figure

6G should be provided.

**Response:**

Thank you very much for your careful review of our manuscript and your valuable
constructive feedback. We have added additional experiments that provided the HE
staining results for the paired or adjacent sections of the fluorescence images in Figure
6G (Fig. R14).

**Fig. R14A.** Fluorescence staining showing co-expression of UTS2R and LOX in
human normal tendon sample and tendinopathy sample. Hematoxylin and eosin (HE)
showing histological morphology of human normal tendon sample and tendinopathy
sample. Scale bars, 100 μm.

2.(1) In Figures 9 and 10, why was the full-thickness window defect model used in
the study?

**Response:**

Thank you very much for your careful review of our manuscript and your valuable
constructive feedback. The full-thickness window defect model was utilized in our study
to replicate severe tendon injury that are difficult to heal on their own. This model
provides a standardized, reproducible, and clinically relevant platform for rigorous

testing. It facilitates histological staining observations and biomechanical testing,
allowing for easier visualization, measurement, and quantification of injury zone
formation, structure, composition, and integration with the uninjured tissue.
Consequently, it offers a clear target area for assessing regenerative outcomes. The
model we utilized, a full-thickness tendon defect, has also been employed in tendon
injury studies relevant to tendon research[1-4].

**References:**

- [1] Wang Y, Jin S, Luo D, et al. Functional regeneration and repair of tendons using biomimetic scaffolds loaded
with recombinant periostin. *Nat Commun.* 2021. 12(1): 1293.
- [2] Wang Z, Lee WJ, Koh B, et al. Functional regeneration of tendons using scaffolds with physical anisotropy
engineered via microarchitectural manipulation. *Sci Adv.* 2018. 4(10): eaat4537.
- [3] Yuan X, Zhu Z, Xia P, et al. Tough Gelatin Hydrogel for Tissue Engineering. *Adv Sci (Weinh).* 2023. 10(24):
e2301665.
- [4] Zhang H, Chen Y, Fan C, et al. Cell-subpopulation alteration and FGF7 activation regulate the function of
tendon stem/progenitor cells in 3D microenvironment revealed by single-cell analysis. *Biomaterials.* 2022. 280:
121238.

(2) Additionally, a 4-week period post-surgery is insufficient for the injured tendon to
fully repair and regenerate. As shown in Figures 10A and B, the tendons were far from
normal histologically. In this context, why were the time points of 2 and 4 weeks chosen
for evaluation?

**Response:**

Thank you very much for carefully reviewing our manuscript and for taking the time
to provide valuable constructive feedback on our research. We have conducted
additional experiments to assess the repair potential of Glema-c-rFGF7 in the
damaged tendons of 8-week-old rabbits, and our results indicate that Glema-c-rFGF7
effectively facilitates tendon tissue repair (Fig. R15).

Tendon healing is a lengthy process that often takes several months to fully mature.

Assessing the 2-week and 4-week marks represents critical early stages of healing. At
2 weeks, we observe the onset of extracellular matrix (ECM) development,
characterized by pathological type III collagen and the initiation of angiogenesis.
Evaluating this time point reveals whether treatment effectively modulates
inflammation, accelerates cell recruitment, and promotes matrix synthesis. By 4 weeks,
cellular activity peaks, and preliminary tissue structures begin to form, allowing for
clearer differentiation among treatment groups regarding cell density, early tissue
architecture, and collagen fiber diameter. The core objective is to determine whether
experimental treatments significantly accelerate healing or enhance quality compared
to the control group.

**Fig. R15. A.** Hematoxylin and eosin (HE) staining of regenerated tendon tissue
at the repair site at 8 weeks post-surgery for each group. Scale bars, 50 μ m.

3. In Figure 10, the cellular composition, particularly the Lox+Uts2r+ fibrotic subcluster,
during FGF7-induced tendon regeneration should be provided.

**Response:**

We sincerely appreciate your valuable constructive feedback. In the FGF7-induced
tendon regeneration group, the proportion of the *Uts2r⁺Lox⁺* fibrotic subcluster is
reduced compared to the control group. We have added immunofluorescence staining
to illustrate the *Uts2r⁺Lox⁺* fibrotic subcluster during FGF7-induced tendon
regeneration, as shown in Fig. R16. The results indicate that, following tendon injury
at 2 weeks and 4 weeks, the *Uts2r⁺Lox⁺* fibrotic subcluster is nearly absent in the
FGF7-induced tendon regeneration group. In contrast, this fibrotic subcluster remains
present in both the injury group and the group treated solely with GelMA.

**Fig. R16. A.** Immunofluorescence staining for UTS2R and LOX at the repair site at 2
1238 weeks post-surgery for each group (Control, GelMA, GelMA-c-rFGF7). Scale bars, 100
1239 μm. **B.** Immunofluorescence staining for UTS2R and LOX at the repair site at 4 weeks
post-surgery for each group (Control, GelMA, GelMA-c-rFGF7). Scale bars, 100 μm.

Reviewer #4 (Remarks to the Author):

In the manuscript titled Harnessing FGF7 for regeneration: from fibrosis suppression
to functional healing, the authors revealed the role of FGF7 in promoting regeneration
and inhibiting fibrosis in mechanically loaded tendons, which was supported by the
motor function, structural analysis and experiments in vitro. Single-cell RNA-
sequencing was used to investigate the mechanism. A GelMA hydrogel loaded with
rFGF7 was manufactured to enhanced functional tendon regeneration in vivo. It is a
well-organized study with instructive meaning. There are some issues that the authors
need to pay attention to:

Thank you for recognizing our innovative contributions. Your affirmation genuinely
inspires us to further our research endeavors. Your comments are presented in black
text, while our responses are in blue text, with any revisions or additions to the
manuscript highlighted in bold blue. In the revised manuscript, all modifications and
additions are displayed in blue font. Thank you once again for your thoughtful review.

1. Unlike fibrosis in solid organs such as the liver or lung, where excessive ECM
deposition replaces functional parenchymal tissue, tendon fibrosis represents a
maladaptive remodeling process within an already ECM-rich connective tissue. Tendon
fibrosis is characterized by a disruption in the composition, spatial organization, and
functional integrity of the collagenous matrix. Therefore, it is essential to provide a clear
definition of tendon fibrosis in the Introduction, taking into account the unique structural
and functional characteristics of tendon tissue. Distinguishing tendon fibrosis from
fibrosis in other organs helps to clarify its specific pathological mechanisms and
underscores the importance of context-specific therapeutic strategies.

**Response:**

Thank you very much for your valuable feedback. We fully agree with your
perspective and we have made the following modifications to the introduction section,

highlighted in blue in the revised manuscript (Page 3, Line 62-79):

**“Tendon injuries and degenerative diseases, such as tendinitis and**
**tendinopathy, present significant clinical challenges, often associated with poor**
**repair and collagen remodeling. Therefore, understanding tendon fibrosis, a key**
**pathological outcome following tendon injury, is essential. Tendon fibrosis**
**refers to a pathological remodeling process that occurs in tendon tissue after**
**injury or chronic lesions. Its core characteristic involves destructive changes in**
**the intrinsic structure and function of the tendon’s extracellular matrix (ECM).**
**As a dense and highly organized connective tissue primarily composed of type**
**I collagen, tendon fibrosis is not characterized by the replacement of functional**
**tissue with excessive ECM, as seen in liver or lung fibrosis. Instead, it manifests**
**as a dysregulation of the inherent ECM composition, including an abnormal**
**increase in type III collagen, and disorganization of the spatial structure, which**
**includes the loss of the alternating parallel arrangement of collagen fibers, a**
**reduction in fibril diameter leading to a small and uniform appearance, abnormal**
**proteoglycan deposition, and neovascularization. Under transmission electron**
**microscopy, the ultrastructure of the pathological extracellular matrix (ECM)**
**reveals a fine and uniform distribution of collagen diameters. These alterations**
**result in a significant decline in the biomechanical properties of the tendon, such**
**as elasticity and tensile strength, severely impacting its load transmission**
**function and ultimately leading to tendon re-rupture and poor functional**
**outcomes for patients.”**

2. ACTA2 was used as a marker of fibrosis in this study, but it is also a marker of
myofibroblasts or vascular smooth muscle cells, and even as a TSPC marker in some
tendon-related studies. So please provide a reasonable basis for ACTA2 as a marker
of tendon fibrosis.

**Response:**

Thank you very much for your valuable feedback. We agree that ACTA2 is also a
marker of myofibroblasts and vascular smooth muscle cells. In response to your
question regarding the use of ACTA2 as a marker for assessing tendon fibrosis, we
have conducted additional experiments. We supplemented our analysis with
transcriptomic data for ACTA2 and type III collagen in both human pathological tendon
samples and normal tendon tissues. Notably, the expression of Acta2 and type III
collagen was significantly elevated in human pathological tendon samples exhibiting
fibrotic characteristics (Fig. R2).

Additionally, we conducted further data analysis. We included single-cell data from
both injury and normal mouse tendons. Based on our single-cell data, we observed a
significant increase in *Acta2* expression in the single-cell sequencing samples from the
injured fibrosis tendons compared to the normal group (Fig. R17).

Although ACTA2 does mark VSMC, the morphological and distribution patterns of
these cells are primarily found in the vascular wall, characterized by a specific layered
structure. This is distinct from the scattered distribution of spindle-shaped cells
expressing ACTA2, which primarily reside within the collagen bundles or scar core.
Moreover, reports regarding expressing ACTA2 in normal tendon tissues primarily
focus on pericyte cells[1] and have also been noted in epitenon/paratenon cells[2, 3].
These cells are consistently located in the anatomical regions of the tendon sheath. In
contrast, in the pathological context of tendon fibrosis, characterized by strong
differentiation, the *Acta2*-positive cells are diffusely distributed within the core of the
fibrotic lesions and are accompanied by substantial collagen deposition. This
distribution aligns more closely with terminally differentiated, highly active
myofibroblasts rather than undifferentiated or proliferating progenitor cells.

Furthermore, ACTA2 is widely accepted and utilized as a marker for myofibroblasts
and the degree of fibrosis in the broader field of fibrosis studies[4-6]. The changes in
its expression levels often directly reflect the severity of the fibrotic pathology and the

**Fig. R2. A-D.** The gene (*COL3*, *ACTA2*, *MKX*, *TNMD*) transcriptomic data of human
normal tendon samples($n=11$) and tendinopathy samples($n=64$), with corresponding
statistical analysis (each dot represents one human sample; * $P<0.05$, ** $P<0.01$,
*** $P<0.001$, two-tailed Student's t test).

**Fig. R17A.** A Uniform Manifold Approximation and Projection (UMAP) plot is
 presented divided into 9 distinct subpopulations. **B.** The UMAP plots illustrate the
 increased expression of the fibrotic marker gene *Acta2* and the black circle indicates
 the increased number of cells following injury.

**References:**

- [1] Yea JH, Gomez-Salazar M, Onggo S, et al. Tpp3(+) synovial/tendon sheath progenitor cells contribute to
 heterotopic bone after trauma. *Bone Res.* 2023. 11(1): 39.
- [2] Dymnt NA, Liu CF, Kazemi N, et al. The paratenon contributes to scleraxis-expressing cells during patellar
 tendon healing. *PLoS One.* 2013. 8(3): e59944.
- [3] Delgado Caceres M, Angerpointner K, Galler M, et al. Tenomodulin knockout mice exhibit worse late healing
 outcomes with augmented trauma-induced heterotopic ossification of Achilles tendon. *Cell Death Dis.* 2021. 12(11):
 1049.
- [4] Yang J, Zhuang H, Li J, et al. The secreted micropeptide C4orf48 enhances renal fibrosis via an RNA-binding
 mechanism. *J Clin Invest.* 2024. 134(10): e178392.
- [5] Reichenbach V, Fernández-Varo G, Casals G, et al. Adenoviral dominant-negative soluble PDGFR β improves
 hepatic collagen, systemic hemodynamics, and portal pressure in fibrotic rats. *J Hepatol.* 2012. 57(5): 967-73.
- [6] Liu Q, Han M, Wu Z, et al. DDX5 inhibits hyaline cartilage fibrosis and degradation in osteoarthritis via
 alternative splicing and G-quadruplex unwinding. *Nat Aging.* 2024. 4(5): 664-680.

3. The trends of Figure 2C and 2E are contrary to the description in Result, which
needs to be checked.

**Response:**

Thank you very much for pointing out this issue. We have re-examined our figures
and the corresponding results description, and we have corrected the presentation of
the results in Figures 2C and 2E highlighted in blue in the revised manuscript (Page 3,
Line 1022-1030):

**Fig. 2C.** Statistical analysis of patellar tendon thickness and patellar tendon paratenon
thickness in HE-stained sections at postnatal days 14, 21, and 10 weeks in *Fgf7^{-/-}* and
wildtype mice ($n=6$ per time point, Error bars represent SDs; WT: wildtype. NS: not
significant, ** $P < 0.01$, *** $P < 0.001$, two-tailed Student's t test).

**Fig. 2E.** Statistical analysis of Achilles tendon thickness and Achilles tendon paratenon
thickness in HE-stained sections at postnatal days 14, 21, and 10 weeks in *Fgf7^{-/-}* and
wildtype mice ($n=6$ per time point, Error bars represent SDs; WT: wildtype. NS: not
significant, *** $P < 0.001$, two-tailed Student's t test).

4. Line 138th: Additionally, we found that the deficiency and fibrosis phenotype at post-
P21 and 10 weeks was not presented in the positional tail tendons.

Line 142th: At P21, wildtype mice showed increased mature collagen fibers, whereas
FGF7^{-/-} mice displayed fewer and discontinuous collagen fibers same as fibrosis
process.

What fibrotic phenotype or fibrosis process are you referring to and what is the
supporting evidence for this statement?

**Response:**

We sincerely appreciate your valuable feedback. We agree with your point that our
previous explanation lacked clarity. When we refer to the fibrotic phenotype, we mean
that at 10 weeks, we observed a distinct tendon fibrotic phenotype in load-bearing
tendons. Evidence supporting this phenotype encompasses the active expression of
fibrosis marker proteins such as ACTA2 and COL3 (Fig. 3E), as well as collagen
morphology characterized by a significant presence of numerous small, immature
collagen fibers, exhibiting a unimodal distribution rather than the typical bimodal
distribution pattern of normal collagen, with diameters significantly smaller than normal
(Fig. 2I, K). Additionally, functional impairment is evident, as indicated by a decline in
relevant animal mobility assessment metrics (Fig. 1C-H).

At P21, we observed hypoplastic phenotype in load-bearing tendons such as the
patellar and Achilles tendons, which were not present in non-load-bearing tendons like
the tail tendon. Therefore, our statement in the original manuscript describing the
expression at P21 as "the same as the fibrosis process" is inaccurate. We have made
the following modifications to the results section, highlighted in blue in the revised
manuscript (Page 7, Line 156-159; Page 8, Line 161-165; Page 8, Line 168-169):

**"To determine whether the hypoplastic phenotype in high mechanical load**
**tendons arises, we performed histological examinations of load-bearing**

tendons such as the patellar and Achilles tendons at P14, P21, and 10 weeks."

"However, by P21, the diameters of both the patellar and Achilles tendons were
significantly reduced in *Fgf7*^{-/-} mice compared to wildtype mice (Fig. 2C, E; Fig.
S1A-D). Additionally, we found that the deficiency phenotype at post-P21 was
not presented in the positional tail tendons."

"At P21, wildtype mice showed increased mature collagen fibers, whereas
*Fgf7*^{-/-} mice displayed fewer and discontinuous collagen fibers."

**Fig. 3E.** Immunofluorescence staining of COL3 and ACTA2 in wildtype and *Fgf7*^{-/-}
mice tendon tissues. Scale bars, 100 μm.

**Fig. 2I, K.** Transmission electron microscopy (TEM) statistical analysis of their
respective collagen diameter distribution in the patellar tendon (PT) (left) and Achilles
tendon (AT) (right) at 10 weeks in *Fgf7*^{-/-} and wildtype mice(n=3).

**Fig. 1C-D.** Schematic representation of the wire hang test protocol (C) and statistical
 analysis of hanging time in the wire hang test at postnatal days 10 weeks for *Fgf7^{-/-}* and
 wildtype mice (D) ($n= 8$ per time point. Error bars represent SDs; WT: wildtype. NS:
 not significant, $*P < 0.05$, $**P < 0.01$, two-tailed Student's t test). **E-H.** Schematic
 representation of the treadmill test (E) and Statistical analysis of maximum running
 speed (F), maximum running distance (G) and maximum running time (H) in the
 treadmill exhaustion test at postnatal 10 weeks for *Fgf7^{-/-}* and wildtype mice ($n= 11$ per
 time point, Error bars represent SDs; WT: wildtype. $*P < 0.05$, $***P < 0.001$, two-tailed
 Student's t test).

5. To support the in vitro findings, a concentration gradient experiment was supposed
 to perform to identify the most effective dose of recombinant FGF7 protein.

**Response:**

According to your valuable suggestions. We added the protein phenotype
 assessments using various concentrations of recombinant FGF7 protein (0, 10, 20,
 and 40 ng/ml) in human tendon stem/progenitor cells. We found that 10 ng/ml of
 recombinant FGF7 protein is the most effective dose for promoting tenogenic
 differentiation and maintaining the tenogenic phenotype (COL1A1, MKX, EGR1) while
 inhibiting the fibrotic phenotype (ACTA2, COL3) in vitro as follows (Fig. R18):

**Fig. R18 A.** Immunofluorescence staining of tendon-related proteins (COL1A1, MKX,
 EGR1) and fibrotic proteins (ACTA2, COL3) in hTSPCs treated with varying
 concentrations of recombinant FGF7: 0 ng/ml (Control), 10 ng/ml, 20 ng/ml, and 40
 1440 ng/ml.

6. Line 243th: In summary, we confirmed that FGF7 inhibits fibrosis by promoting the
 proliferation and tenogenic differentiation of both mouse and human tendon
 stem/progenitor cells.

The conclusion is supposed to modify exactly; because the results in Figure 4 do not
 reveal the relation between the promoting role of proliferation and tenogenic
 differentiation and the inhibition role of fibrosis.

**Response:**

Thank you very much for your detailed review of our manuscript and for your
 valuable feedback. We completely agree with your observations that the results in
 Figure 4 do not reveal the relation between the promoting role of proliferation and

tenogenic differentiation and the inhibition role of fibrosis.

We have corrected the conclusion highlighted in blue in the revised manuscript (Page
13, Line 269-271) as follows:

**"In summary, we confirmed that FGF7 inhibits fibrosis and concomitantly**
**promotes the proliferation and tenogenic differentiation of both mouse and**
**human tendon stem/progenitor cells."**

7. Additional evidence is needed to prove the conclusion that the persistence and
fibrosis-promoting activity of the *Lox⁺Uts2r⁺* fibrotic subcluster in FGF7KO tendons
are maintained through the regulation of the *Creb3l1* transcription factor by TGF- β
signaling.

**Response:**

Thank you very much for your detailed review of our manuscript and for your
valuable feedback. We have added the following experimental results to the
supplementary materials (Fig. S9), highlighted in blue in the revised manuscript (Page
76, Line 1445-1449). A detailed description of the results can be found in the revised
manuscript as follows (Page 18, Line 386-390):

**"Specifically, we implemented inhibition experiments targeting *Creb3l1*. By**
**introducing *Creb3l1* siRNA into tendon stem cells derived from FGF7 knockout**
**mice, we observed a reduction in *Lox⁺Uts2r⁺* double-positive cells. This was**
**accompanied by an increase in the expression of the tendon-specific protein**
**TNMD and a decrease in the fibrotic markers ACTA2 and COL3 (Fig. S9)."**

**Fig. S9.A.** Immunofluorescence staining of LOX and UTS2R in mTSPCs from *Fgf7*^{-/-}
 mice treated with Si-*Creb311* or Si-NC, Scale bars, 100 μm. **B.** Immunofluorescence
 staining of tendon-related proteins (TNMD) and fibrotic proteins (ACTA2, COL3) in
 mTSPCs from *Fgf7*^{-/-} mice treated with Si-*Creb311* or Si-NC, 100 μm.

8. Literature support or data is needed to demonstrate that isolated tendon tissue
 from SCX-GFP mice, when removed from its in vivo environment, develops fibrosis.

**Response:**

Thank you very much for your detailed review of our manuscript and for your
 valuable feedback. Isolated tendon tissue from SCX-GFP mice does exhibit fibrotic
 changes, and we did not provide a detailed explanation in the manuscript, which may
 have led to some misunderstanding. We have made the following modifications to the
 results section, highlighted in blue in the revised manuscript (Page 19, Line 399-402):

**“Previous studies have reported that tendon tissue from SCX-GFP mice loses**
 **its SCX phenotype immediately after explant culture[1]. The loss of SCX**
 **expression in tendon-related genes is also a characteristic of tendon fibrosis.”**

Additionally, we conducted experiments to verify the occurrence of fibrosis in the
 explants after 7 days and observed a significant upregulation of fibrosis markers. For
 instance, through immunofluorescence experiments, we found that the markers ACTA2
 and COL3 were markedly elevated. Additionally, DAPI staining revealed a noticeable
 disarray in cell arrangement (Fig. 8I, J). Using qPCR, we detected increased RNA
 expression of fibrosis-related genes Acta2 and Col3 (Fig. 8B). Furthermore,
 ultrastructural examination of the explanted tendons via transmission electron
 microscopy showed that the collagen diameter became significantly smaller and more
 uniform, resembling the fibrotic phenotype observed in *in vivo* tendon tissue (Fig. 8D;
 Fig. R19).

 **Fig. 8B.** qPCR analysis of tendon fibrosis genes (*Col3*, *Acta2*) in cultured tendon
 explants from both groups ($n=3$ independent experiments, bars represent mean \pm SD;
 $*P<0.05$, $**P<0.01$, two-tailed Student's *t* test). **D.** Transmission electron microscopy
 (TEM) statistical analysis of their respective collagen diameter distribution in
 transverse sections of tendon explants cultured with rFGF7 or PBS ($n=3$). **I.**
 Immunofluorescence staining of COL3 proteins in mouse tendon explants. Scale bars,
 100 μm. **J.** Immunofluorescence staining of ACTA2 proteins in mouse tendon explants.
 Scale bars, 100 μm.

**Fig. R19A.** Transmission electron microscopy (TEM) images showing transverse
sections of collagen fibers of tendon explants cultured with rFGF7 or PBS ($n=3$), scale
bars: 200nm.

**References:**

[1] Zhang C, Zhang E, Yang L, et al. Histone deacetylase inhibitor treated cell sheet from mouse tendon
stem/progenitor cells promotes tendon repair. *Biomaterials*. 2018. 172: 66-82.

9. More material characterization experiments of GelMA-c-rFGF7 hydrogel, including
SEM, need to be supplemented.

**Response:**

Thank you very much for your detailed review of our manuscript and for your
valuable suggestions. We have conducted scanning electron microscopy (SEM) on the
GelMA-c-rFGF7 hydrogel. The results of this experiment have been added to the
supplementary materials (Fig. S10) and are highlighted in blue in the revised
manuscript (Page 77, Line 1470-1472). A detailed description of the results can be
found in the revised manuscript as follows (Page 20, Line 435-438):

**“We have conducted scanning electron microscopy (SEM) on the GelMA-c-**
**rFGF7 hydrogel to elucidate its internal microarchitecture. Representative SEM**
**images (Fig. S10A) revealed that the hydrogel exhibited a highly porous and**
**interconnected three-dimensional network.”**

We also employed infrared spectroscopy to the characterize the chemical structure
of GelMA hydrogels (Fig. R20). Compared to native gelatin, the amide I and amide II
bands in the GelMA spectrum shifted to higher wavenumbers (from 1631 cm^{-1} to 1635

1534 cm^{-1} for amide I, and from 1535 cm^{-1} to 1538 cm^{-1} for amide II). This shift confirms the
1535 successful modification of the gelatin molecules, as the introduction of methacrylamide
groups alters the hydrogen bonding environment, resulting in the observed changes in
the infrared absorption bands.

**Fig. S10A.** Representative scanning electron microscope (SEM) images, scale bars:
$100\mu\text{m}$ (right).

**Fig. R20.A.** Fourier-transform infrared (FTIR) spectra of gelatin and GelMA. The
shift in the absorption bands of amide I (from 1631 cm^{-1} to 1635 cm^{-1}) and amide II
(from 1535 cm^{-1} to 1528 cm^{-1}) indicates the successful conjugation of methacrylamide
groups onto the gelatin backbone.

10. In the Materials and Methods section, please check whether CellChat or

CellphoneDB is used to reveal the cell-cell interaction in the single-cell RNA-
sequencing analysis.

**Response:**

Thank you very much for your detailed review of our manuscript and for your
valuable suggestions. We did not use CellphoneDB to demonstrate the cell-cell
interactions in the single-cell RNA-sequencing analysis. Instead, in Fig. 7C and Fig.
7D, we utilized CellChat for this analysis. We have revised the relevant expression in
the Methods section of the manuscript and highlighted it in blue in the revised
manuscript (Page 33, Line 716-718) as follows:

"Quality control excluded cells with <1,200 or >7,000 genes, and total counts
<25,000 or >1,000,000. Ligand-receptor interactions were analyzed using
CellChat."

11. Is it possible to construct tendon or Achilles tendon specific FGF7KO mice?

**Response:**

We sincerely appreciate your thorough review of this manuscript and your valuable
suggestions. Tendon-specific FGF7 knockout (FGF7KO) mice would be extremely
beneficial for our research. Unfortunately, our research group have not currently ready-
made mice system with loxP sites flanking the critical exons of the Fgf7 gene. In future
studies, we plan to develop this mouse model to support our ongoing research.

12. Some English grammar mistakes need to be corrected.

**Response:**

Thank you for kindly reminding. We have revised the text of our article to enhance
its readability and ensure professional English standards.

**REVIEWER COMMENTS**

**Reviewer #1 (Remarks to the Author):**

The authors have made appreciable efforts in addressing previous comments, and
the manuscript has been improved accordingly. However, a number of essential points
still require further attention to solidify the central claims of this study. The following
suggestions are offered to help enhance the clarity and robustness of the work:

We thank you for your insightful comments, which have been instrumental in
strengthening our manuscript. We have addressed the feedback, and our point-by-
point responses are provided below. For easy reference, reviewer comments are in
black, our responses are in blue, and changes in the manuscript are in blue. We
appreciate the reviewers' time and valuable contributions to our work.

1. In Lines 303–313, the authors describe single-cell RNA sequencing performed on
tendon tissue from FGF7 KO mice. It would be helpful to provide references or rationale
supporting the criteria used for cell clustering to enhance the methodological
transparency.

**Response:**

Thank you for your insightful suggestions. Regarding *Lox⁺Uts2r⁺* fibrotic cells
(Cluster 0), *Lox* is not only a marker gene associated with liver fibrosis but also a
significant marker for the initial collagen cross-linking in tendon tissue[1, 2]. The
presence of initial collagen cross-linking, which fails to develop into more mature cross-
linking, results in the formation of thinner collagen fibers—one of the characteristic
features of tendon fibrosis. *Uts2r* is implicated in matrix cross-linking, recognized as a
marker gene for fibrotic diseases, and exerts direct pro-fibrotic effects[3]. For the
tenocytes subcluster (Cluster 1), characterized by *Postn* and *Apod*, *Postn* is
upregulated in tendon-specific cell populations[4] and is regarded as a beneficial
molecule that facilitates functional tendon regeneration[5]. Additionally, it has been
reported that *Apod* is present within tenocyte subpopulations[6]. The cells in this

subcluster primarily contribute to the maintenance and functional roles of tendon tissue.
In the case of the tenoblasts subcluster (Cluster 2), characterized by *Thbs4* and *Comp*,
*THBS4* is considered a tendon-specific gene expressed in tendon fibroblasts[7]. *Comp*
has been identified as a key protein secreted by the normal tendon extracellular matrix
(ECM)[8], The tenoblasts subgroup exhibit enhanced ECM secretion characteristics.
For *Cd44⁺Tppp3⁺* tendon progenitors (Cluster 3), *Cd44* is widely recognized as a
marker gene for tendon stem/progenitor cells (TSPCs), playing a vital role in
maintaining the characteristics of tendon stem cells[9, 10]. *Tppp3*-positive cells are
involved in the repair of tendon tissue damage and are classified as a type of TSPC[11].
Regarding Tenogenic cells (Cluster 4), characterized by *Eln* and *Cygb*, literature
suggests that *Eln* enzymatically promotes the degradation of tendon tissue
structure[12] and is considered a relevant protein in tendon formation or differentiation
processes[13]. *Cygb* is a marker protein whose expression increases in tendon lineage
cells and participates in tendon differentiation[14]. This subcluster is primarily
characterized by providing conditions or a microenvironment conducive to tendon
tissue generation. For enthesoblasts (Cluster 5), characterized by *Col2a1* and *Cnmd*,
*Col2a1* is recognized as a key marker protein at the tendon-bone interface[15].
Furthermore, it has been reported that *Cnmd* is present in the transitional zone
between tendon and cartilage[16]. Regarding *Prg4⁺* tendon progenitors (Cluster 6),
characterized by *Prg4* and *Tspan15*, both *Prg4* and *Tspan15* constitute a
subpopulation within tendon stem cells that are associated with the differentiation of
progenitor cells found in connective tissues such as tendons and ligaments[17]. For
Muscle-tendon junction cells (Cluster 7), characterized by *Col22a1* and *Frem2*,
*Col22a1* has been reported to predominantly express at the muscle-tendon
junction[18], while *Frem2* serves as a marker at the tendon-muscle junction[19]. Finally,
for Mineralizing chondrocytes (Cluster 8), characterized by *Runx2* and *Bglap*, *Runx2*
is a protein associated with hypertrophic chondrocyte differentiation[20], indicating that
the cells in this subcluster are transitioning towards a mineralizing chondrocyte state
at the tendon-bone interface. *Bglap* is also considered a gene related to tissue
mineralization[21].

**References:**

- [1] Chen W, Yang A, Jia J, Popov YV, Schuppan D, You H. Lysyl Oxidase (LOX) Family Members: Rationale
and Their Potential as Therapeutic Targets for Liver Fibrosis. *Hepatology*. 2020. 72(2): 729-741.
- [2] Marturano JE, Xylas JF, Sridharan GV, Georgakoudi I, Kuo CK. Lysyl oxidase-mediated collagen
crosslinks may be assessed as markers of functional properties of tendon tissue formation. *Acta Biomater*.
2014. 10(3): 1370-9.
- [3] Tzanidis A, Hannan RD, Thomas WG, et al. Direct actions of urotensin II on the heart: implications for
cardiac fibrosis and hypertrophy. *Circ Res*. 2003. 93(3): 246-53.
- [4] Gomez-Collignon A, Brown R, Carr A, et al. Single cell multi-omics characterise discrete human tendon
cells populations that persist in vitro and on fibrous scaffolds. *Eur Cell Mater*. 2022. 44: 1-20.
- [5] Wang Y, Jin S, Luo D, et al. Functional regeneration and repair of tendons using biomimetic scaffolds
loaded with recombinant periostin. *Nat Commun*. 2021. 12(1): 1293.
- [6] Kendal AR, Layton T, Al-Mossawi H, et al. Multi-omic single cell analysis resolves novel stromal cell
populations in healthy and diseased human tendon. *Sci Rep*. 2020. 10(1): 13939.
- [7] Jelinsky SA, Archambault J, Li L, Seeherman H. Tendon-selective genes identified from rat and human
musculoskeletal tissues. *J Orthop Res*. 2010. 28(3): 289-97.
- [8] Li C, Wang N, Schäffer AA, et al. Mutations in COMP cause familial carpal tunnel syndrome. *Nat*
*Commun*. 2020. 11(1): 3642.
- [9] Bi Y, Ehrichiou D, Kilts TM, et al. Identification of tendon stem/progenitor cells and the role of the
extracellular matrix in their niche. *Nat Med*. 2007. 13(10): 1219-27.
- [10] Chen S, Lin Y, Yang H, et al. A CD26(+) tendon stem progenitor cell population contributes to tendon
repair and heterotopic ossification. *Nat Commun*. 2025. 16(1): 749.
- [11] Harvey T, Flamenco S, Fan CM. A Tppp3(+)/Pdgfra(+) tendon stem cell population contributes to
regeneration and reveals a shared role for PDGF signalling in regeneration and fibrosis. *Nat Cell Biol*.
2019. 21(12): 1490-1503.
- [12] Wu YT, Wu YT, Huang TC, Su FC, Jou IM, Wu CC. Sequential inflammation model for Achilles
tendinopathy by elastin degradation with treadmill exercise. *J Orthop Translat*. 2020. 23: 113-121.
- [13] Kraus A, Sattler D, Wehland M, Luetzenberg R, Abuagela N, Infanger M. Vascular Endothelial Growth
Factor Enhances Proliferation of Human Tenocytes and Promotes Tenogenic Gene Expression. *Plast*
*Reconstr Surg*. 2018. 142(5): 1240-1247.
- [14] Zhang H, Chen Y, Fan C, et al. Cell-subpopulation alteration and FGF7 activation regulate the function of

- tendon stem/progenitor cells in 3D microenvironment revealed by single-cell analysis. *Biomaterials*. 2022.
280: 121238.
- [15] Luzzi AJ, Ferrer X, Fang F, et al. Hedgehog Activation for Enhanced Rotator Cuff Tendon-to-Bone
Healing. *Am J Sports Med*. 2023. 51(14): 3825-3834.
- [16] Yukata K, Matsui Y, Shukunami C, et al. Differential expression of Tenomodulin and Chondromodulin-1 at
the insertion site of the tendon reflects a phenotypic transition of the resident cells. *Tissue Cell*. 2010.
42(2): 116-20.
- [17] Tachibana N, Chijimatsu R, Okada H, et al. RSPO2 defines a distinct undifferentiated progenitor in the
tendon/ligament and suppresses ectopic ossification. *Sci Adv*. 2022. 8(33): eabn2138.
- [18] Jacobson KR, Lipp S, Acuna A, Leng Y, Bu Y, Calve S. Comparative Analysis of the Extracellular Matrix
Proteome across the Myotendinous Junction. *J Proteome Res*. 2020. 19(10): 3955-3967.
- [19] Karlsen A, Yeung CC, Schjerling P, et al. Distinct myofibre domains of the human myotendinous junction
revealed by single-nucleus RNA sequencing. *J Cell Sci*. 2023. 136(8): jcs260913 [pii].
- [20] Chan B, Glogauer M, Wang Y, et al. Adseverin, an actin-binding protein, modulates hypertrophic
chondrocyte differentiation and osteoarthritis progression. *Sci Adv*. 2023. 9(31): eadf1130.
- [21] Yan R, Ding J, Yang Q, et al. Lead acetate induces cartilage defects and bone loss in zebrafish embryos by
disrupting the GH/IGF-1 axis. *Ecotoxicol Environ Saf*. 2023. 253: 114666.

2. A putative fibrotic LOX⁺UTS2R⁺ subpopulation was identified in FGF7 KO mice.
However, as indicated in Fig. 5E, both LOX and UTS2R are also highly expressed in
certain tenoblast subpopulations. Therefore, relying solely on these two markers may
not be sufficient to distinguish a unique fibrosis-driving cluster. It would be important to
clarify whether this subpopulation also expresses other markers or fibrosis-related
genes at elevated levels.

**Response:**

Thank you very much for your insightful suggestions. We conducted an analysis of
the expression of additional important fibrotic markers, especially those promoting
fibrotic collagen fiber formation and cross-linking, in the *Lox⁺Uts2r⁺* subpopulation
including *Mfap4* (a marker for hepatic and organ fibrosis[1, 2]), *Loxl2* (associated with
hepatic and cardiac fibrosis[3, 4]), *P4ha1* (organ fibrosis, particularly hepatic fibrosis[5,

6]), *Cthrc1* (related to bronchial and cardiac fibrosis[7, 8]), *Ccn4* (linked to hepatic
 fibrosis and fibrotic diseases in humans[9, 10]), and *Fkbp10* (associated with knee and
 pulmonary fibrosis[11, 12]). Our findings indicate that the levels of promoting collagen
 fiber formation and cross-linking genes in this subpopulation are significantly elevated
 compared to those in other subpopulations (Fig. S11). We have added additional
 experiments and data to the supplementary materials of the article (Fig. S11) and are
 highlighted in blue in the revised manuscript (Page 80, Line 1474-1478). A detailed
 description of the results can be found in the revised manuscript (Page 15, Line 317-
 321) as follows:

“Consistent with the observed tendon matrix assembly disorder in the patellar and
 Achilles tendons of *Fgf7^{-/-}* mice, analysis of subcluster 0 revealed a positive correlation
 with multiple markers in promoting collagen fiber formation and cross-linking compared
 to other subpopulations, such as *Lox*(25), *Uts2r*(26), *Mfap4*(27), and *Acta2*(28), among
 others(Fig. 5E; Fig. S11).”

**Fig. S11.** The Dot plot of the fibrosis-related genes in 9 subclusters from 8-week *Fgf7^{-/-}*
 138 ^{-/-} and 8-week wildtype mice Achilles tendon samples. (A) Dot plot showing selected

promoting collagen fiber formation and cross-linking genes (*Mfap4*, *Loxl2*, *P4ha1*,
*Cthrc1*, *Ccn4*, *Fkbp10*) enriched in the *Lox⁺Uts2r⁺* subpopulation.

**References:**

[1] Mölleken C, Ahrens M, Schlosser A, et al. Direct-acting antivirals-based therapy decreases hepatic fibrosis
serum biomarker microfibrillar-associated protein 4 in hepatitis C patients. *Clin Mol Hepatol*. 2019. 25(1):
42-51.

[2] Wozny MR, Nelea V, Siddiqui I, et al. Microfibril-associated glycoprotein 4 forms octamers that mediate
interactions with elastogenic proteins and cells. *Nat Commun*. 2024. 15(1): 4015.

[3] Ikenaga N, Peng ZW, Vaid KA, et al. Selective targeting of lysyl oxidase-like 2 (LOXL2) suppresses
hepatic fibrosis progression and accelerates its reversal. *Gut*. 2017. 66(9): 1697-1708.

[4] Yang J, Savvatis K, Kang JS, et al. Targeting LOXL2 for cardiac interstitial fibrosis and heart failure
treatment. *Nat Commun*. 2016. 7: 13710.

[5] Li J, Ghazwani M, Zhang Y, et al. miR-122 regulates collagen production via targeting hepatic stellate cells
and suppressing P4HA1 expression. *J Hepatol*. 2013. 58(3): 522-8.

[6] Yang X, Zhang D, Li M, Shao Y, Zhang X, Xue Y. P4HA1: an important target for treating fibrosis related
diseases and cancer. *Front Pharmacol*. 2024. 15: 1493420.

[7] Ruiz-Villalba A, Romero JP, Hernández SC, et al. Single-Cell RNA Sequencing Analysis Reveals a Crucial
Role for CTHRC1 (Collagen Triple Helix Repeat Containing 1) Cardiac Fibroblasts After Myocardial
Infarction. *Circulation*. 2020. 142(19): 1831-1847.

[8] Fang Y, Chung S, Xu L, et al. RUNX2 promotes fibrosis via an alveolar-to-pathological fibroblast
transition. *Nature*. 2025. 640(8057): 221-230.

[9] Singh K, Oladipupo SS. An overview of CCN4 (WISP1) role in human diseases. *J Transl Med*. 2024.
22(1): 601.

[10] Xi Y, LaCanna R, Ma HY, et al. A WISP1 antibody inhibits MRTF signaling to prevent the progression of
established liver fibrosis. *Cell Metab*. 2022. 34(9): 1377-1393.e8.

[11] Lim J, Lietman C, Grol MW, et al. Localized chondro-ossification underlies joint dysfunction and motor
deficits in the *Fkbp10* mouse model of osteogenesis imperfecta. *Proc Natl Acad Sci U S A*. 2021. 118(25):
e2100690118.

[12] Staab-Weijnitz CA, Fernandez IE, Knüppel L, et al. FK506-Binding Protein 10, a Potential Novel Drug
Target for Idiopathic Pulmonary Fibrosis. *Am J Respir Crit Care Med*. 2015. 192(4): 455-67.

3. Throughout the manuscript, the interpretation of key data—particularly regarding
tendon tissue morphology and marker localization—would be strengthened by
including wider-field views for both H&E staining and fluorescence images (e.g., Fig.
5F, 6G, 10E). Overview images would help readers better assess the anatomical
context and cellular distribution of the signals presented.

**Response:**

Thank you very much for your insightful suggestions. As suggested, Figures 5F, 6G,
and 10E have been updated with H&E staining from serial sections and fluorescence
images with a wider field of view. We have added the data to the article and highlighted
in blue in the revised manuscript

1. Representative H&E staining and fluorescence images were added to Figure 5F:

**Fig. 5F.** Confocal fluorescence imaging and showing the co-expression of *Lox* and
*Uts2r* in Achilles tendon tissues from 8-week *Fgf7*^{-/-} and wildtype mice, with
representative HE-stained images. Scale bars, 100 μm.

2. Representative H&E staining and fluorescence images were added to Figure 6G:

**Fig. 6G.** Co-expression of *Lox* and *Uts2r* in human normal tendon samples and
 tendinopathy samples. **(G)** Representative H&E staining images and corresponding
 wide-field confocal fluorescence micrographs for the *Lox* and *Uts2r* co-expression in
 human normal tendon samples and tendinopathy samples. Scale bars, 100 μm .

3. Representative H&E staining and fluorescence images from Figure 10E were
 added to S19:

**Fig. S19.** The observation of representative H&E staining and immunofluorescence
 staining at the repair site at 4 weeks post-surgery for each group (Injury, GelMA,
 GelMA-*c*-rFGF7). (A-E) Representative H&E staining and immunofluorescence
 staining for tendon-related proteins (COL1A1, TNMD, THBS4) and fibrosis-related
 proteins (COL3, ACTA2) at the repair site at 4 weeks post-surgery for each group
 (Injury, GelMA, GelMA-*c*-rFGF7). Scale bars, 100 μ m.

4. In Lines 360–363, the observation of GFP⁺LOX⁺UTS2R⁺ cells at a single time point

is interesting but may not fully support the proposed differentiation trajectory. Multi-
time-point observations would help more definitively establish whether
GFP⁺LOX⁻UTS2R⁻ cells can differentiate into GFP⁺LOX⁺UTS2R⁺ cells over time.

**Response:**

Thank you very much for your insightful suggestions. We have conducted additional
multi-time-point observations at 5 weeks and 6 weeks, which indicate that
GFP⁺Lox⁻Uts2r⁻ cells can differentiate into GFP⁺Lox⁺Uts2r⁺ cells over time. This
demonstrates that under FGF7 knockout (KO) conditions, the newly proliferated
tendon cells transition into Lox⁺Uts2r⁺ cells (Fig. S14). We have added additional
experiments and data to the supplementary materials of the article (Fig. S14) and are
highlighted in blue in the revised manuscript (Page 83, Line 1494-1506). A detailed
description of the results can be found in the revised manuscript (Page 17, Line 369-
372) as follows:

“As tendon abnormalities in *Fgf7*^{-/-} mice primarily emerge after postnatal day 21, we
administered tamoxifen to *ProTracer* mice and *ProTracer*-background *Fgf7*^{-/-} mice at
postnatal day 21 and collected tendon tissues at 5, 6 and 8 weeks (Fig. 6C; Fig. S14).”

**Fig. S14.** The observation of $GFP^+Lox^+Uts2r^+$ cells at multi-time-point. **(A)** Strategy
 for the seamless recording of cell proliferation and the schematic of the experimental
 design. **(B-C)** Confocal fluorescence staining showing co-expression of GFP, UTS2R
 and LOX in tendon tissues from ProTracer mice and $Fgf7^{-/-}$ ProTracer mice at 5 weeks,
 with corresponding statistical analysis of the proportion of ProTracer labeled cells
 differentiating into Lox^+Uts2r^+ cells ($n=4$ independent experiments, bars represent
 mean \pm SD; *** $P<0.001$, two-tailed Student's t test). Scale bars, 100 μ m. **(D-E)** Confocal
 fluorescence staining showing co-expression of GFP, UTS2R and LOX in tendon
 tissues from ProTracer mice and $Fgf7^{-/-}$ ProTracer mice at 6 weeks, with corresponding
 statistical analysis of the proportion of ProTracer labeled cells differentiating into
 Lox^+Uts2r^+ cells ($n=4$ independent experiments, bars represent mean \pm SD; *** $P<0.001$,
 two-tailed Student's t test). Scale bars, 100 μ m.

5. In Figures R4A and R5A, the images shown for the control and AZD4547-treated
 groups appear very similar. Moreover, they resemble images previously presented in

Figure 4G. It would be important to confirm that these experimental and control
conditions were conducted under comparable and simultaneous conditions. In addition,
inclusion of statistical analysis for these figures would strengthen the interpretation of
the results.

**Response:**

Thank you very much for your insightful suggestions. In response to your
recommendations, we conducted re-experiments for Figure 4G and Figures R4A and
R5A, which included the AZD4547 treatment group supplemented with FGF7, as well
as groups treated with AZD4547 supplemented with rFGF2, rFGF2 alone, AZD4547
supplemented with rFGF4, and rFGF4 alone (Figure 4G-H; Figure S4; Figure S5). We
ensured that these experimental and control conditions were conducted under
comparable and simultaneous settings. The AZD4547 treatment group supplemented
with FGF7 did not effectively restore the expression levels of COL1A1, MKX, and
EGR1, nor did it reduce the expression of ACTA2 and COL3. These results indicate
that FGF7 failed to rescue the inhibitory effects of AZD4547 on tendon stem cells,
whereas both FGF2 and FGF4 restored the expression levels suppressed by AZD4547.
This demonstrates that FGF7 exerts its specific role in tendon stem cells through
signaling receptors that are targeted and inhibited by AZD4547. The results are as
follows:

**Fig. S4.** Effects of recombinant FGF2 on tendon-related markers in hTSPCs. **(A-B)**
 Immunofluorescence staining of tendon-related proteins (COL1A1, MKX, EGR1) and
 fibrotic proteins (ACTA2, COL3) in hTSPCs treated with rFGF2, PBS, AZD4547, co-
 treated with rFGF2 and AZD4547, corresponding semi-quantitative fluorescence
 analysis, $n=3$ independent experiments, $**P<0.01$, $***P<0.001$, two-tailed Student's t
 test, Scale bars, 100 μm .

**Fig. S5.** Effects of recombinant FGF4 on tendon-related markers in hTSPCs. **(A-B)**
 Immunofluorescence staining of tendon-related proteins (COL1A1, MKX, EGR1) and
 fibrotic proteins (ACTA2, COL3) in hTSPCs treated with rFGF4, PBS, AZD4547, co-
 treated with rFGF4 and AZD4547, corresponding semi-quantitative fluorescence
 analysis, $n=3$ independent experiments, ** $P<0.01$, *** $P<0.001$, two-tailed Student's t
 test, Scale bars, 100 μm .

**Fig. 4. (G-H)** Immunofluorescence staining of tendon-related proteins (COL1A1,
 MKX, EGR1) and fibrotic proteins (ACTA2, COL3) in hTSPCs treated with rFGF7,
 PBS, AZD4547, co-treated with rFGF7 and AZD4547, corresponding semi-quantitative
 fluorescence analysis, $n=3$ independent experiments, NS: not significant, two-tailed
 Student's t test, Scale bars, 100 μ m.

6. Regarding Figure R6, the in vitro data are valuable but may not be sufficient to
 demonstrate that FGF7 readministration attenuates fibrosis or promotes tendon
 regeneration in FGF7 KO mice. If possible, in vivo rescue experiments in FGF7 KO
 mice would provide more direct evidence. If such experiments are not feasible, a brief
 justification would be helpful.

**Response:**

We sincerely thank the reviewer for this insightful suggestion. We fully agree that in
 vivo rescue experiments would provide stronger evidence. In line with this comment,
 we had indeed considered and designed corresponding rescue experiments during the
 early planning phase of our study. However, in the process, we encountered several

fundamental technical challenges that currently render such in vivo rescue studies
unfeasible. Firstly, due to the minute size of tendons in developing neonatal mice, the
implantation of any sustained-release carrier (e.g., hydrogel) would cause substantial
physical disruption to the delicate tendon structure. Secondly, the therapeutic
application of recombinant FGF7 is significantly hampered by its pharmacokinetic
properties. Similar to other FGF family members, FGF7 is known to have a very short
half-life in vivo—on the order of minutes to a few hours—due to rapid proteolytic
degradation and clearance. Maintaining effective local concentrations would therefore
require frequent injections, which itself would cause repeated tissue injury, disrupt the
healing microenvironment, and mask the true phenotypic outcomes. We appreciated
this comment, which highlights a key challenge in the field. Our future work will focus
on developing stable protein delivery strategies to overcome these limitations.

7. The pseudotime analysis suggesting that CD44⁺Tppp3⁺ TSPCs may give rise to
LOX⁺UTS2R⁺ cells is intriguing. However, given that proliferating cells often
differentiate into fibroblasts, additional in vitro or in vivo validation would strengthen the
conclusion that CD44⁺Tppp3⁺ TSPCs can indeed differentiate into LOX⁺UTS2R⁺ cells.

**Response:**

We sincerely appreciate the profound insights and valuable suggestions you
provided. We strongly agree that validating the differentiation potential of *Cd44⁺Tppp3⁺*
tendon stem/progenitor cells (TSPCs) into *Lox⁺Uts2r⁺* cells through in vitro or in vivo
experiments would significantly strengthen our conclusions. In an effort to directly test
this hypothesis, we attempted to isolate the *Cd44⁺Tppp3⁺* TSPC population. However,
the *Tppp3* is an intracellular protein, which prevents its use as a surface marker for
flow cytometry-based sorting. This presents a technical challenge for directly isolating
and tracking the *Cd44⁺Tppp3⁺* population. We considered whether isolating *Cd44*-
positive cells alone could serve as a surrogate for the *Cd44⁺Tppp3⁺* population. This
consideration was based on the fact that CD44 is a surface protein, making it amenable
to fluorescence-activated cell sorting (FACS). However, upon re-examining our

scRNA-seq data, we found that *Cd44*-single-positive cells constituted a higher
proportion of the cells and were more widely distributed than the *Cd44*⁺*Tppp3*⁺ double-
positive population (Fig. R1). Isolating *Cd44* positive cells alone would not yield results
representative of the differentiation trajectory from *Cd44*⁺*Tppp3*⁺ TSPCs to *Lox*⁺*Uts2r*⁺
cells. To address this, we plan to develop a *Cd44*⁺*Tppp3*⁺ gene-modified mouse model
in future studies. This will allow us to employ lineage tracing experiments both in vitro
and in vivo to clarify the differentiation fate of this specific cell population. The potential
differentiation of *Cd44*⁺ *Tppp3*⁺ TSPCs into *Lox*⁺ *Uts2r*⁺ cells require future validation,
but it does not impact our main findings regarding FGF7's role in development and
fibrosis suppression. We appreciate the reviewer's valuable feedback.

**Fig. R1. (A)** UMAP plots illustrating the expression of *Tppp3* and *Cd44* in single-cell
RNA sequencing samples of Achilles tendon tissues from wild-type (WT) and *Fgf7*^{-/-}
mice. **(B)** The proportion of *Cd44*⁺ cells and *Cd44*⁺*Tppp3*⁺ cells.

Reviewer #2 (Remarks to the Author):

The revised manuscript has been partially responsive to the prior critique. However,
there are two main issues that have not been resolved.

We sincerely appreciate the time and effort that you have dedicated to providing
insightful feedback on our work. We have carefully considered all the comments and
have made significant revisions to the manuscript to address them. Below, we have
provided our point-by-point responses to the reviewers' comments. We have worked
hard to address the concerns raised and hope that the revisions in the manuscript and

our responses here will be found satisfactory. Your comments are presented in black
text, while our responses are in blue text, with any revisions or additions to the
manuscript highlighted in blue. In the revised manuscript, the modifications and
additions are displayed in blue font. Thank you once again for your thoughtful review.

The primary issue relates to the authors conclusion that the developmental
phenotype in *Fgf7*^{-/-} tendons is fibrotic. While the response to the prior comment does
provide a more nuanced explanation for the rationale for calling this a fibrotic
phenotype the data do not support this. For example, the new data demonstrating
decreased diameters across *Fgf7*^{-/-} tendon suggest this not a true fibrotic state (e.g.,
excess and disorganized matrix). Similar comment for the picrosirius/ PLM data,
decreased Col1 expression, and the lack of aSMA+ stress fiber + cells. There is clearly
a collagen assembly, cross-linking etc phenotype but not fibrosis.

**Response:**

We sincerely appreciate your insightful comments and valuable feedback. We fully
agree with the reviewers' perspective that the observed phenotypes in *Fgf7*^{-/-} tendons
involve disturbances in collagen assembly and cross-linking. To avoid conceptual
confusion, we have ceased using the term "fibrosis" to describe the *Fgf7*^{-/-} mice tendon
developmental phenotype throughout the manuscript. We have adopted your core
insight and will refer to this phenotype as a "tendon matrix assembly disorder"
characterized by structural defects in the matrix that prevent collagen assembly from
forming structures capable of withstanding the mechanical strength of the tendon. A
key feature of this phenotype is the microstructural uniformity of collagen, presenting
with a smaller, consistent diameter, in contrast to the bimodal distribution of collagen
diameters typically observed in normal tendons. We have made the modifications
regarding the tendon developmental phenotype of *Fgf7*^{-/-} mice, which are highlighted
in blue throughout the revised manuscript.

Second, the authors use the conclusion of fibrosis to justify the identification of the
*Uts2r+* *Lox+* cells as a driver of fibrosis. However, there is no direct evidence to
demonstrate a causal role for these cells in fibrosis. The evidence of greater Tgfb
signaling within/to/from this population is not very compelling, given the strong Tgfb
signaling signature in other populations (e.g., the *Prg4+* population). Moreover,
changes in the presence of this population were not assessed in the GelMA
experiments. It would be helpful to use publicly available mouse scSeq data to
demonstrate evidence of this population in either fibrotic tendon healing or tendon
degeneration.

**Response:**

We thank you for highlighting the insufficient evidence for the causal relationship.
We have rephrased the *Uts2r⁺Lox⁺* subpopulation as a group associated with tendon
matrix assembly disorders. In accordance with your invaluable suggestions, we
analyzed publicly available single-cell RNA sequencing data from mouse tendons
(PMID: 40230536)[1]. After performing UMAP analysis on the cell populations (Fig.
R2A), we found that the number of cells in cluster 0, 1, and 2 was significantly
increased in the injured tendon tissue compared to healthy tendon tissue. Importantly,
the *Uts2r⁺Lox⁺* double-positive cells, which are the focus of our study, were primarily
distributed within subpopulations 0, 1, and 2, and their numbers also notably increased
(as indicated by the black circles highlighting the increased *Uts2r⁺Lox⁺* double-positive
cells in subpopulations 0, 1, and 2 in Fig. R2B; the red box highlighting the increased
rate in subpopulations 0, 1, and 2 in Fig. R2C). This provides independent evidence
for the relevance of this cell population in a true pathological context.

**Fig. R2** The expansion of *Uts2r*⁺*Lox*⁺ cells observed in the post-injury fibrotic healing
 process. **(A)** UMAP show single cell RNA sequencing data from injured and healthy
 mouse tendons. **(B)** A comparison of UMAP plots reveals that *Uts2r*⁺*Lox*⁺ cells are
 increased in the injured tendon, with these cells primarily expressed in subpopulations
 0, 1, and 2 in the injured tendons (as indicated by the black circles). **(C)** Representative
 399 bar plots showing the rate of cells belonging to each subcluster in Achilles tendon
 tissues from from injured and healthy mouse tendons. **(D)** The heatmap illustrates the
 roles and relative importance of the eight tendon subclusters within the *TGF-β* signaling
 network, with the red box highlighting the active signaling in subclusters 0 and 1.

Regarding the changes in this population in the GelMA model that you mentioned,

we have confirmed through supplementary experiments that an increase in the
 *Uts2r*⁺*Lox*⁺ cell population was also observed in our GelMA model (Fig. S20). This
 finding also further confirms the genuine presence of the *Uts2r*⁺*Lox*⁺ cell population in
 the context of tendon injury in rats.

**Fig. S20. Expression of *Uts2r*⁺ *Lox*⁺ cells during the repair process of injured rat**
 **patellar tendons treated with GelMA-c-rFGF7 hydrogel, GelMA, and the**
 **untreated injury group. (A) Confocal fluorescence imaging showing the co-**
 **expression of *Lox* and *Uts2r* at 2 weeks post-surgery for each group (Injury, GelMA,**
 **GelMA-c-rFGF7), *n*=6 independent experiments, bars represent mean ± SD;**
 ******P*<0.001, two-tailed Student's *t* test. Scale bars, 100 μm. (B) Confocal fluorescence**
 **imaging showing the co-expression of *Lox* and *Uts2r* at 4 weeks post-surgery for each**
 **group (Injury, GelMA, GelMA-c-rFGF7), *n*=6 independent experiments, bars represent**
 **mean ± SD; ****P*<0.001, two-tailed Student's *t* test. Scale bars, 100 μm. (C) *Uts2r*⁺*Lox*⁺**
 **cells gradually increase during the tendon injury process in rats from 2 weeks to 4 weeks**
 **post-injury. *n*=6 independent experiments, bars represent mean ± SD; ****P*<0.001, two-**
 **tailed Student's *t* test. Scale bars, 100 μm.**

Additionally, we have revisited our single-cell sequencing data from both FGF7

KO and WT mice to conduct a transcriptomic analysis of the *Uts2r⁺Lox⁺* cell population.
 We found that this population is characterized by the high expression of genes that
 promote fibrotic collagen fiber formation and cross-linking (Fig. S11), including *Mfap4*
 (a marker for hepatic and organ fibrosis [2, 3]), *Loxl2* (associated with hepatic and
 cardiac fibrosis [4, 5]), *P4ha1* (involved in organ fibrosis, particularly hepatic fibrosis [6,
 7]), *Cthrc1* (related to bronchial and cardiac fibrosis [8, 9]), *Ccn4* (linked to hepatic
 fibrosis and fibrotic diseases in humans [10, 11]), and *Fkbp10* (associated with knee
 and pulmonary fibrosis [12, 13]). Our findings indicate that the levels of promoting
 collagen fiber formation and cross-linking genes in this subpopulation are significantly
 elevated compared to those in other subpopulations. This suggests that the *Uts2r⁺Lox⁺*
 cell population may be associated with the pathological processes of collagen
 formation and cross-linking that occur following injury. We will continue to conduct in-
 depth studies on this subpopulation in future research.

 **Fig. S11.** The Dot plot of the fibrosis-related genes in 9 subclusters from 8-week *Fgf7*
 439 ^{-/-} and 8-week wildtype mice Achilles tendon samples. **(A)** Dot plot showing selected
 promoting collagen fiber formation and cross-linking genes (*Mfap4*, *Loxl2*, *P4ha1*,
 *Cthrc1*, *Ccn4*, *Fkbp10*) enriched in the *Lox⁺Uts2r⁺* subpopulation.

Finally, regarding the statement that "The evidence of greater *TGFβ* signaling

within/to/from this population is not very compelling," we initially observed that in the
*Uts2r⁺Lox⁺* cells (subpopulation 0), the *TGFβ* signaling pathway was the most
prominent compared to other pathways within this subset. Therefore, we first focused
our investigation on this pathway (Fig. R3). In an independent analysis, following your
insightful suggestion, we utilized publicly available single-cell RNA sequencing data
from mouse tendons (PMID: 40230536) and identified the *Uts2r⁺Lox⁺* cell population
(characterized by widespread expression in subpopulations 0 and 1), which increases
during tendon injury and also exhibits enhanced active *TGFβ* signaling (Fig. R2D).

Thank you once again for your assistance in enhancing the quality of our manuscript.

**Fig. R3.** Gene Set Variation Analysis in the *Uts2r⁺Lox⁺* cells (A) Gene Set Variation
Analysis (GSVA) using the Hallmark gene sets reveals elevated activity of the TGF-β
signaling pathway in the *Uts2r⁺Lox⁺* cells.

**References:**

- [1] Huang Z, Li Z, Ruan D, et al. Dynamic changes of molecular pattern and cellular subpopulation in
puncture-induced tendon injury model. *iScience*. 2025. 28(4): 112034.
- [2] Mölleken C, Ahrens M, Schlosser A, et al. Direct-acting antivirals-based therapy decreases hepatic fibrosis
serum biomarker microfibrillar-associated protein 4 in hepatitis C patients. *Clin Mol Hepatol*. 2019. 25(1):
42-51.
- [3] Wozny MR, Nelea V, Siddiqui I, et al. Microfibril-associated glycoprotein 4 forms octamers that mediate
interactions with elastogenic proteins and cells. *Nat Commun*. 2024. 15(1): 4015.
- [4] Ikenaga N, Peng ZW, Vaid KA, et al. Selective targeting of lysyl oxidase-like 2 (LOXL2) suppresses
hepatic fibrosis progression and accelerates its reversal. *Gut*. 2017. 66(9): 1697-1708.
- [5] Yang J, Savvatis K, Kang JS, et al. Targeting LOXL2 for cardiac interstitial fibrosis and heart failure
treatment. *Nat Commun*. 2016. 7: 13710.
- [6] Li J, Ghazwani M, Zhang Y, et al. miR-122 regulates collagen production via targeting hepatic stellate cells
and suppressing P4HA1 expression. *J Hepatol*. 2013. 58(3): 522-8.
- [7] Yang X, Zhang D, Li M, Shao Y, Zhang X, Xue Y. P4HA1: an important target for treating fibrosis related
diseases and cancer. *Front Pharmacol*. 2024. 15: 1493420.
- [8] Ruiz-Villalba A, Romero JP, Hernández SC, et al. Single-Cell RNA Sequencing Analysis Reveals a Crucial
Role for CTHRC1 (Collagen Triple Helix Repeat Containing 1) Cardiac Fibroblasts After Myocardial
Infarction. *Circulation*. 2020. 142(19): 1831-1847.
- [9] Fang Y, Chung S, Xu L, et al. RUNX2 promotes fibrosis via an alveolar-to-pathological fibroblast
transition. *Nature*. 2025. 640(8057): 221-230.
- [10] Singh K, Oladipupo SS. An overview of CCN4 (WISP1) role in human diseases. *J Transl Med*. 2024.
22(1): 601.
- [11] Xi Y, LaCanna R, Ma HY, et al. A WISP1 antibody inhibits MRTF signaling to prevent the progression of
established liver fibrosis. *Cell Metab*. 2022. 34(9): 1377-1393.e8.
- [12] Lim J, Lietman C, Grol MW, et al. Localized chondro-ossification underlies joint dysfunction and motor
deficits in the Fkbp10 mouse model of osteogenesis imperfecta. *Proc Natl Acad Sci U S A*. 2021. 118(25):
e2100690118.
- [13] Staab-Weijnitz CA, Fernandez IE, Knüppel L, et al. FK506-Binding Protein 10, a Potential Novel Drug
Target for Idiopathic Pulmonary Fibrosis. *Am J Respir Crit Care Med*. 2015. 192(4): 455-67.

There seem to be far more Utsr2+ Lox+ cells in 5B, vs D, presumably because some
are in the 21 day time point? It is important to clarify whether the lack of this population

is maturation state dependent.

**Response:**

We sincerely appreciate your insightful comments and valuable feedback. Your
observations regarding Figures 5B and 5D are particularly pertinent, as they highlight
an important implication for our research. We confirmed the presence of the *Uts2r⁺Lox⁺*
cell population at 21 days. In normal wild-type mice, this population diminishes with
tendon maturation by 8 weeks, whereas in adult *Fgf7^{-/-}* mice, it persists abnormally. To
present our findings more clearly, we have moved the 21-day and 8-week results to
the supplementary material in the revised manuscript (Fig. S8). We have highlighted
the supplementary material in blue in the revised manuscript (Page 77, Line 1430-
1433). A detailed description of the results can be found in the revised manuscript
(Page 14, Line 305-309) as follows:

“Single-cell RNA sequencing analysis of tendon tissues from 21-day-old and 8-week-
old *Fgf7^{-/-}* and wild-type mice showed that the *Uts2r⁺Lox⁺* cell population is present in
immature tendons at postnatal day 21, but disappears in normally developed wild-type
tendons by 8 weeks. In contrast, this population persists in *Fgf7^{-/-}* tendons at 8 weeks
(Fig. S8).”

**Fig. S8.** *Uts2r⁺Lox⁺* cell persistence in *Fgf7^{-/-}* adult tendon. (A) UMAP plots showing
 the distribution of the 9 subclusters of Achilles tendon samples from 21 days *Fgf7^{-/-}*
 mice, 8-week *Fgf7^{-/-}* mice, 21 days wildtype mice, 8-week wildtype mice.

Several pieces of data including the in response to reviewers should be included in the
 manuscript or supplementary data:

1. Which populations from 5A do the *Uts2r⁺Lox⁺* cells map to? Figure R11B

**Response:**

We sincerely appreciate your insightful comments and valuable feedback. The
 *Uts2r⁺Lox⁺* cells in Figure R11B correspond to Tenocytes 2 from Figure 5A. We have
 included the relevant supplementary data from the previous round of the response to
 reviewers in the manuscript. The results of this experiment have been added to the
 supplementary data as follows (Fig. S9A) and are highlighted in blue in the revised
 manuscript (Page 78, Line 1443-1449). A detailed description of the results can be
 found in the revised manuscript as follows (Page 14, Line 309-312):

“To better illustrate the distribution of the *Lox⁺Uts2r⁺* subpopulation among the
 overall cells in tendon tissue prior to subclustering, we highlighted the *Lox⁺Uts2r⁺*

subpopulation within the grey circle and noted that it is primarily expressed in the
Tenocytes 2 subgroup of the initial clustering (Fig. S9A-B)”

**Fig. S9A.** Uniform Manifold Approximation and Projection (UMAP) plot displaying
31,756 cells divided into 17 distinct subpopulations. The circle subpopulation includes
subgroups 0, 1, 2, 3, and 4, which exhibit tendon characteristics. **B.** UMAP plots
indicating the expression of Lox^+Uts2r^+ subcluster is primarily concentrated within the
tendon characteristics subpopulation. The grey circle highlights Tenocytes 2, where the
Lox^+Uts2r^+ fibrotic subcluster is predominantly expressed.

2. Fig R12

**Response:**

We sincerely appreciate your insightful comments and valuable feedback. We have
added the relevant supplementary data from the previous round of the response to
reviewers in the manuscript. The results of this experiment have been added to the
supplementary materials as follows (Fig. S12) and are highlighted in blue in the revised
manuscript (Page 82, Line 1481-1486). A detailed description of the results can be
found in the revised manuscript as follows (Page 15, Line 324-330):

“Analysis of single-cell RNA sequencing data from both fibrotic tendon repair and
healthy mouse tendon samples revealed dynamic changes in the Lox^+Uts2r^+
subcluster during tendon fibrosis repair in wild-type (WT) mice. In the UMAP
visualization (Fig. S12), the circled regions highlight cell populations that increased
following injury. Specifically, the Lox^+Uts2r^+ subcluster (marked as Cluster 0) was
significantly expanded during tendon injury compared with normal tendons (Fig. S12A–

**Fig. S12A.** A Uniform Manifold Approximation and Projection (UMAP) plot is
 presented divided into 9 distinct subpopulations. **B.** UMAP plots illustrating the
 expression of fibrotic marker genes *Lox*, *Uts2r*, *Acta2*. The subpopulation within the
 black circle represents the increased number of cells following injury, where the
 Lox^+Uts2r^+ cell subpopulation shows elevated expression.

3. Fig R9A. Related, in Figure 6 F -H the authors should be very clear about the
tendons that these samples were isolated from.

**Response:**

We sincerely appreciate your insightful comments and valuable feedback. We have
included the relevant supplementary data from the previous round of the response to
reviewers in the manuscript. We have clearly described the source of the tendons from
which these samples were isolated. The results of this experiment have been added
to the supplementary materials as follows (Fig. S13) and are highlighted in blue in the
revised manuscript (Page 82, Lines 1489-1492). A detailed description of the results
can be found in the revised manuscript as follows (Page 16, Lines 348-354; Page 30,
Lines 631-636):

“Obtaining genuinely healthy rotator cuff tendon samples from human donors
presents considerable practical and ethical difficulties, as such tissues are seldom
accessible through surgical procedures. Therefore, we utilized normal hamstring
tendons as an alternative model for our investigation. The proportions of tendon cell
subpopulations in both normal hamstring tendons and normal rotator cuff tendons were
observed to be largely similar (Fig. S13).”

“Healthy tendon samples were obtained from hamstring tendon tissues used as
autologous grafts during anterior cruciate ligament (ACL) reconstruction surgeries,
while the pathological tendon samples were derived from pathological rotator cuff
tendon tissue. Because obtaining truly healthy rotator cuff tendon samples from human
donors poses significant practical and ethical challenges. Healthy rotator cuff tendons
are rarely available from surgical procedures.”

**Fig. S13A.** UMAP plots of single-cell sequencing data from normal rotator cuff and
 normal hamstring tendon tissues. **B-C.** Statistical data on the proportions of each cell
 subpopulation, along with specific numerical values for each proportion.

4. The nomenclature in Figure 7 is very confusing. Are the WT mice sham controls with
 no injury? The difference between this group and the 'control' is not immediately clear
 from the text.

**Response:**

We sincerely appreciate your insightful comments and valuable feedback. We did
 not find the WT mice sham group in Figure 7. After consideration, we think your
 confusion may stem from the nomenclature used in Figure 9. This was indeed a
 misleading on our part, which could lead to reader confusion, and we apologize for that.
 In Figure 9, the WT rats represent the sham group, which did not undergo the tendon
 injury surgical procedure, while the Control group represents the injury rats, which
 underwent the tendon injury surgical procedure but received no treatment. To enhance
 clarity for our readers, we have revised the nomenclature in the text and in Figures 9
 and 10, changing "WT" to "Sham" and "Control" to "Injury". We have revised the
 relevant results descriptions and the labels for Figures 9 and 10 for improved clarity.

These changes are also highlighted in blue in the revised manuscript.

Minor:

1. Intro- the sentence on lines 80-81 have no relevance, and lack clarity, thus should
be removed.

**Response:**

We sincerely appreciate your insightful comments and valuable feedback. The
content of the sentence on lines 80-81 in the previous manuscript has been removed.

2. The introduction is excessively long and unfocused

**Response:**

We sincerely appreciate your insightful comments and valuable feedback. We have
streamlined the introduction to enhance its focus and made corresponding
modifications, which are highlighted in blue in the revised manuscript (Page 3, Line 48-
117) as follows:

[revised manuscript text omitted]

- and drug delivery strategies. *Adv Drug Deliv Rev.* 2021. 173: 504-519.
- 2. V. Hernandez-Gea, S. L. Friedman, Pathogenesis of liver fibrosis. *Annu Rev Pathol* 6, 425-456 (2011).
- 3. M. Wijsenbeek, V. Cottin, Spectrum of Fibrotic Lung Diseases. *N. Engl. J. Med.* 383, 958-968 (2020).
- 4. T. Harvey, S. Flamenco, C. M. Fan, A Tppp3(+)Pdgfra(+) tendon stem cell population contributes to
regeneration and reveals a shared role for PDGF signalling in regeneration and fibrosis. *Nat. Cell Biol.* 21,
1490-1503 (2019).
- 5. M. A. Karsdal, S. H. Nielsen, D. J. Leeming, L. L. Langholm, M. J. Nielsen, T. Manon-Jensen, A. Siebuhr,
693 N. S. Gudmann, S. Rønnow, J. M. Sand, S. J. Daniels, J. H. Mortensen, D. Schuppan, The good and the bad
collagens of fibrosis - Their role in signaling and organ function. *Adv. Drug Deliv. Rev.* 121, 43-56 (2017).
- 6. E. Yelin, S. Weinstein, T. King, The burden of musculoskeletal diseases in the United States. *Semin.*
*Arthritis Rheum.* 46, 259-260 (2016).
- 7. E. Havis, M. A. Bonnin, J. Esteves de Lima, B. Charvet, C. Milet, D. Duprez, TGF β and FGF promote
tendon progenitor fate and act downstream of muscle contraction to regulate tendon differentiation during
chick limb development. *Development* 143, 3839-3851 (2016).
- 8. E. Havis, M. A. Bonnin, I. Olivera-Martinez, N. Nazaret, M. Ruggiu, J. Weibel, C. Durand, M. J. Guerquin,
C. Bonod-Bidaud, F. Ruggiero, R. Schweitzer, D. Duprez, Transcriptomic analysis of mouse limb tendon
cells during development. *Development* 141, 3683-3696 (2014).
- 9. C. Liu, N. Zhou, N. Li, T. Xu, X. Chen, H. Zhou, A. Xie, H. Liu, L. Zhu, S. Wang, J. Xiao, Disrupted
tenogenesis in masseter as a potential cause of micrognathia. *Int J Oral Sci* 14, 50 (2022).
- 10. R. Rana *et al.*, Impaired 1,25-dihydroxyvitamin D3 action underlies enthesopathy development in the Hyp
mouse model of X-linked hypophosphatemia. *JCI Insight* 8, e163259 (2023).
- 11. T. Tokunaga, C. Shukunami, N. Okamoto, T. Taniwaki, K. Oka, H. Sakamoto, J. Ide, H. Mizuta, Y. Hiraki,
FGF-2 Stimulates the Growth of Tenogenic Progenitor Cells to Facilitate the Generation of Tenomodulin-
Positive Tenocytes in a Rat Rotator Cuff Healing Model. *Am J Sports Med* 43, 2411-2422 (2015).
- 12. S. Liu *et al.*, Tendon healing and anti-adhesion properties of electrospun fibrous membranes containing
bFGF loaded nanoparticles. *Biomaterials* 34, 4690-4701 (2013).
- 13. M. J. Livingston, S. Shu, Y. Fan, Z. Li, Q. Jiao, X. M. Yin, M. A. Venkatachalam, Z. Dong, Tubular cells
produce FGF2 via autophagy after acute kidney injury leading to fibroblast activation and renal fibrosis.
*Autophagy* 19, 256-277 (2023).
- 14. Y. S. Kim, J. S. Lee, M. Y. Jeong, J. W. Jang, M. S. Kim, Recombinant human fibroblast growth factor 7
obtained from stable Chinese hamster ovary cells enhances wound healing. *Biotechnol J* 19, e2300596
(2024).

- 15. M. Mongiat, K. Taylor, J. Otto, S. Aho, J. Uitto, J. M. Whitelock, R. V. Iozzo, The protein core of the
proteoglycan perlecan binds specifically to fibroblast growth factor-7. JOURNAL OF BIOLOGICAL
CHEMISTRY 275, 7095-7100 (2000).
- 16. A. J. Hayes, B. L. Farrugia, I. J. Biose, G. J. Bix, J. Melrose, Perlecan, A Multi-Functional, Cell-Instructive,
Matrix-Stabilizing Proteoglycan With Roles in Tissue Development Has Relevance to Connective Tissue
Repair and Regeneration. Front Cell Dev Biol 10, 856261 (2022).
- 17. E. Geervliet, L. Terstappen, R. Bansal, Hepatocyte survival and proliferation by fibroblast growth factor 7
attenuates liver inflammation, and fibrogenesis during acute liver injury via paracrine mechanisms. Biomed
Pharmacother 167, 115612 (2023).
- 18. H. Zhang *et al.*, Cell-subpopulation alteration and FGF7 activation regulate the function of tendon
stem/progenitor cells in 3D microenvironment revealed by single-cell analysis. Biomaterials 280, 121238
(2022).
- 19. Y. Ren *et al.*, Gene expression of Postn and FGF7 in canine chordae tendineae and their effects on flexor
tenocyte biology. J. Orthop. Res. 42, 961-972 (2024).

Reviewer #3 (Remarks to the Author):

The authors have made comprehensive revisions in accordance with our comments,
and we have no further concerns.

**Response:**

We are truly grateful for your positive feedback and for recognizing the originality
and significant impact of our study. Your insights have been invaluable in enhancing
the quality of our work. We appreciate your encouragement and support, which
motivate us to continue exploring this important area of research. Thank you once
again for your thoughtful review.

Reviewer #4 (Remarks to the Author):

I believe the authors have addressed the issues. The revised manuscript could be
published in this journal now.

**Response:**

We are honored by your positive assessment of our work and grateful for your
recognition of its novelty and potential impact. Your insightful feedback has been
invaluable in strengthening the manuscript. We deeply appreciate your encouragement,
which serves as a great motivation for our continued research in this field. Thank you
for your valuable contributions to our work.